# Adversarial Counterfactual Environment Model Learning

**Xiong-Hui Chen**[1,2,4,*], **Yang Yu**[1,2,4,*,†], **Zheng-Mao Zhu**[1,2], **Zhihua Yu**[1,2,3], **Zhenjun Chen**[1,2,3],
**Chenghe Wang**[1,2], **Yinan Wu**[3], **Hongqiu Wu**[1,2], **Rong-Jun Qin**[1,2,4],**Ruijin Ding**[5], **Fangsheng Huang**[3]

[1] National Key Laboratory for Novel Software Technology, Nanjing University
[2] School of Artificial Intelligence, Nanjing University, [3] Meituan, [4] Polixir.ai, [5] Tsinghua University
{chenxh, yuy, zhuzm}@lamda.nju.edu.cn, {yuzh,chenzj}@smail.nju.edu.cn
wangch@lamda.nju.edu.cn, wuyinan02@meituan.com, {wuhq,qinrj}@lamda.nju.edu.cn
drj17@mails.tsinghua.edu.cn, huangfangsheng@meituan.com

## Abstract

An accurate environment dynamics model is crucial for various downstream tasks in sequential decision-making, such as counterfactual prediction, off-policy evaluation, and offline reinforcement learning. Currently, these models were learned through empirical risk minimization (ERM) by step-wise fitting of historical transition data. This way was previously believed unreliable over long-horizon rollouts because of the compounding errors, which can lead to uncontrollable inaccuracies in predictions. In this paper, we find that the challenge extends beyond just long-term prediction errors: we reveal that even when planning with one step, learned dynamics models can also perform poorly due to the selection bias of behavior policies during data collection. This issue will significantly mislead the policy optimization process even in identifying single-step optimal actions, further leading to a greater risk in sequential decision-making scenarios. To tackle this problem, we introduce a novel model-learning objective called adversarial weighted empirical risk minimization (AWRM). AWRM incorporates an adversarial policy that exploits the model to generate a data distribution that weakens the model's prediction accuracy, and subsequently, the model is learned under this adversarial data distribution. We implement a practical algorithm, GALILEO, for AWRM and evaluate it on two synthetic tasks, three continuous-control tasks, and *a real-world application*. The experiments demonstrate that GALILEO can accurately predict counterfactual actions and improve various downstream tasks, including offline policy evaluation and improvement, as well as online decision-making.

## 1 Introduction

A good environment dynamics model for action-effect prediction is essential for many downstream tasks. For example, humans or agents can leverage this model to conduct simulations to understand future outcomes, evaluate other policies' performance, and discover better policies. With environment models, costly real-world trial-and-error processes can be avoided. These tasks are vital research problems in counterfactual predictions [52, 1], off-policy evaluation (OPE) [32, 35], and offline reinforcement learning (Offline RL) [31, 59, 12, 13, 10]. In these problems, the core role of the models is to answer queries on counterfactual data unbiasedly, that is, given states, correctly answer *what might happen* if we were to carry out actions *unseen* in the training data. However, addressing counterfactual queries differentiates environment model learning from standard supervised learning (SL), which directly fits the offline dataset for empirical risk minimization (ERM). In essence,

---

[*]The authors contribute equally
[†]Yang Yu is the corresponding author.

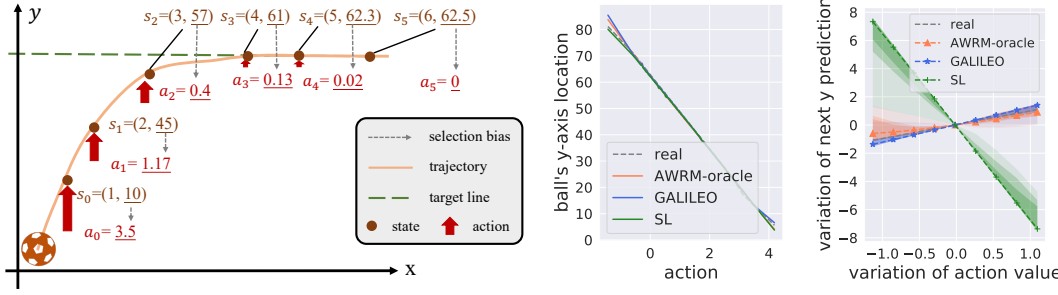

(a) a selection bias example

(b) responses to actions in the training data

(c) responses to actions in the action space

Figure 1: An example of selection bias and predictions under counterfactual queries. Suppose a ball locates in a 2D plane whose position is $s_t = (x_t, y_t)$ at time $t$. The ball will move to $s_{t+1} = (x_{t+1}, y_{t+1})$ according to $x_{t+1} = x_t + 1$ and $y_{t+1} \sim \mathcal{N}(y_t + a_t, 2)$. Here, $a_t$ is chosen by a control policy $a_t \sim \pi_\phi(a|s_t) = \mathcal{N}((\phi - y_t)/15, 0.05)$ parameterized by $\phi$, which tries to keep the ball near the line $y = \phi$. In Fig. 1(a), the behavior policy $\mu$ is $\pi_{62.5}$. Fig. 1(b) shows the collected training data and the learned models' prediction of the next position of $y$. Besides, the dataset superficially presents the relation that the corresponding next $y$ will be smaller with a larger action. However, the truth is not because the larger $a_t$ causes a smaller $y_{t+1}$, but the policy selects a small $a_t$ when $y_t$ is close to the target line. **Mistakenly exploiting the "association" will lead to local optima with serious factual errors**, e.g., believed that $y_{t+1} \propto \pi_\phi^{-1}(y_t|a) + a_t \propto \phi - 14a_t$, where $\pi_\phi^{-1}$ is the inverse function of $\pi_\phi$. When we estimate the response curves by fixing $y_t$ and reassigning action $a_t$ with other actions $a_t + \Delta a$, where $\Delta a \in [-1, 1]$ is a variation of action value, we found that the model of SL indeed exploit the association and give opposite responses, while in AWRM and its practical implementation GALILEO, the predictions are closer to the ground truths ($y_{t+1} \propto y_t + a_t$). The result is in Fig. 1(c), where the darker a region is, the more samples are fallen in. AWRM injects data collected by adversarial policies for model learning to eliminate the unidentifiability between $y_{t+1} \propto \pi_\phi^{-1}(y_t|a) + a_t$ and w$y_{t+1} \propto y_t + a_t$ in offline data.

the problem involves training the model on one dataset and testing it on another with a shifted distribution, specifically, the dataset generated by counterfactual queries. This challenge surpasses the SL's capability as it violates the independent and identically distributed (*i.i.d.*) assumption [31].

In this paper, we concentrate on faithful dynamics model learning in sequential decision-making settings like RL. Specifically, we first highlight a distinct situation of distribution shift that can easily lead to *catastrophic failures in the model's predictions*: In many real-world applications, offline data is often gathered using a single policy with exhibiting selection bias, meaning that, for each state, actions are chosen unfairly. As illustrated in Fig. 1(a), to maintain the ball's trajectory along a target line, a behavior policy applies a smaller force when the ball's location is nearer to the target line. When a dataset is collected with such selection bias, the association between the states (location) and actions (force) will make SL hard to identify the correct causal relationship of the states and actions to the next states respectively (see Fig. 1(c)). Then when we query the model with counterfactual data, the predictions might be catastrophic failures. *Finally, offline policy optimization based on this SL model, even for just seeking one-step optimal actions, will select also a totally opposite direction of policy improvement, making the offline policy learning system fail.* The selection bias can be regarded as an instance of the distributional-shift problem in offline model-based RL, which has also received great attention [31, 59, 14, 34]. However, previous methods employing *naive supervised learning* for environment model learning tend to overlook this issue during the learning process,

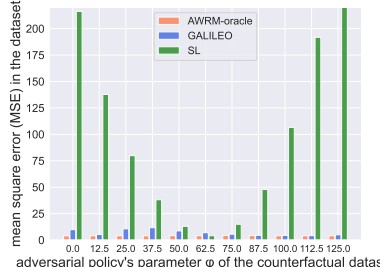

Figure 2: An illustration of the prediction error in counterfactual datasets. The error of SL is small only in training data ($\phi = 62.5$) but becomes much larger in the dataset "far away from" the training data. AWRM-oracle selects the oracle worst counterfactual dataset for training for each iteration (pseudocode is in Alg. 2) which reaches small MSE in all datasets and gives correct response curves (Fig. 1(c)). GALILEO approximates the optimal adversarial counterfactual data distribution based on the training data and model. Although the MSE of GALILEO is a bit larger than SL in the training data, in the counterfactual datasets, the MSE is on the same scale as AWRM-oracle.

addressing it instead by limiting policy exploration and learning in high-risk regions. So far, how to learn a faithful environment model that can alleviate the problem directly has rarely been discussed.

In this work, we focus on faithful environment model learning techniques. The work is first inspired by weighted empirical risk minimization (WERM), which is a typical solution to solve the selection bias problem in causal inference for individual treatment effects (ITEs) estimation in many scenarios like patients' treatment selection [26, 1, 41]. ITEs measure the effects of treatments on individuals by administering treatments uniformly and evaluating the differences in outcomes. To estimate ITEs from offline datasets with selection bias, they estimate an inverse propensity score (IPS) to reweight the training data, approximating the data distribution under a uniform policy, and train the model under this reweighted distribution. Compared with ITEs estimation, the extra challenge of faithful model learning in sequential decision-making settings include: (1) the model needs to answer queries on numerous different policies, resulting in various and unknown target data distributions for reweighting, and (2) the IPS should account for the cumulative effects of behavior policies on state distribution rather than solely focusing on bias of actions. To address these issues, we propose an objective called adversarial weighted empirical risk minimization (AWRM). For each iteration, AWRM employs adversarial policies to construct an adversarial counterfactual dataset that maximizes the model's prediction error, and drive the model to reduce the prediction risks under the adversarial counterfactual data distribution. However, obtaining the adversarial counterfactual data distribution is infeasible in the offline setting. Therefore, we derive an approximation of the counterfactual data distribution queried by the optimal adversarial policy and provide a tractable solution to learn a model from the approximated data distribution. As a result, we propose a practical approach named **G**enerative **A**dversarial off**LI**ne counterfactua**L** **E**nvironment m**O**del learning (GALILEO) for AWRM. Fig. 2 illustrates the difference in the prediction errors learned by these algorithms.

Experiments are conducted in two synthetic tasks, three continuous-control tasks, and *a real-world application*. We first verify that GALILEO can make accurate predictions on counterfactual data queried by other policies compared with baselines. We then demonstrate that the model learned by GALILEO is helpful to several downstream tasks including offline policy evaluation and improvement, and online decision-making in a large-scale production environment.

## 2 Preliminaries

We first introduce weighted empirical risk minimization (WERM) through inverse propensity scoring (IPS), which is commenly used in individualized treatment effects (ITEs) estimation [43]. It can be regarded as a scenario of single-step model learning . We define $M^*(y|x, a)$ as the one-step environment, where $x$ denotes the state vector containing pre-treatment covariates (such as age and weight), $a$ denotes the treatment variable which is the action intervening with the state $x$, and $y$ is the feedback of the environment. When the offline dataset is collected with a behavior policy $\mu(a|x)$ which has selection bias, a classical solution to handle the above problem is WERM through IPS $\omega$ [48, 3, 29]:

**Definition 2.1.** The learning objective of WERM through IPS is formulated as

$$\min_{M \in \mathcal{M}} \mathbb{E}_{x,a,y \sim p^\mu_{M^*}}[\omega(x, a)\ell(M(y|x, a), y)], \tag{1}$$

where $\omega(x, a) := \frac{\beta(a|x)}{\mu(a|x)}$, $\beta$ and $\mu$ denote the policies in testing and training domains, and the joint probability $p^\mu_{M^*}(x, a, y) := \rho_0(x)\mu(a|x)M^*(y|x, a)$ in which $\rho_0(x)$ is the distribution of state. $\mathcal{M}$ is the model space. $\ell$ is a loss function.

The $\omega$ is also known as importance sampling (IS) weight, which corrects the sampling bias by aligning the training data distribution with the testing data. By selecting different $\hat{\omega}$ to approximate $\omega$ to learn the model $M$, current environment model learning algorithms employing reweighting are fallen into the framework. For standard ITEs estimation, $\omega = \frac{1}{\hat{\mu}}$ (i.e., $\beta$ is a uniform policy) for balancing treatments, where $\hat{\mu}$ is an approximated behavior policy $\mu$. Note that it is a reasonable weight in ITEs estimation: ITEs are defined to evaluate the differences of effect on each state under a uniform policy.

In sequential decision-making setting, decision-making processes in a sequential environment are often formulated into Markov Decision Process (MDP) [51]. MDP depicts an agent interacting with the environment through actions. In the first step, states are sampled from an initial state distribution $x_0 \sim \rho_0(x)$. Then at each time-step $t \in \{0, 1, 2, ...\}$, the agent takes an action $a_t \in \mathcal{A}$ through a policy $\pi(a_t|x_t) \in \Pi$ based on the state $x_t \in \mathcal{X}$, then the agent receives a reward $r_t$ from

a reward function $r(x_t, a_t) \in \mathbb{R}$ and transits to the next state $x_{t+1}$ given by a transition function $M^*(x_{t+1}|x_t, a_t)$ built in the environment. $\Pi$, $\mathcal{X}$, and $\mathcal{A}$ denote the policy, state, and action spaces.

# 3 Related Work

We give related adversarial algorithms for model learning in the following and leave other related work in Appx. F. In particular, ITEs in Rubin causal model [48] and causal effect estimation in structural causal model [38] attracted widespread attention in recent years [56, 55, 58, 7]. GANTIE [58] uses a generator to fill counterfactual outcomes for each data pair and a discriminator to judge the source (treatment group or control group) of the filled data pair. The generator is trained to minimize the output of the discriminator. [7] propose SCIGAN to extend GANITE to continuous ITEs estimation via a hierarchical discriminator architecture. In real-world applications, environment model learning based on Generative Adversarial Imitation Learning (GAIL) has also been adopted for sequential decision-making problems [25]. GAIL is first proposed for policy imitation [25], which uses the imitated policy to generate trajectories by interacting with the environment. The policy is learned with the trajectories through RL which maximizes the cumulative rewards given by the discriminator. [47, 11] use GAIL for environment model learning by regarding the environment model as the generator and the behavior policy as the "environment" in standard GAIL. [16] further inject the technique into a unified objective for model-based RL, which joints model and policy optimization. Our study reveals the connection between adversarial model learning and the WERM through IPS, where previous adversarial model learning methods can be regarded as partial implementations of GALILEO, explaining the effectiveness of the former.

# 4 Adversarial Counterfactual Environment Model Learning

In this section, we first propose a new offline model-learning objective for sequential decision-making setting in Sec. 4.1; In Sec. 4.2, we give a surrogate objective to the proposed objective; Finally, we give a practical solution in Sec. 4.3.

## 4.1 Problem Formulation

In scenarios like offline policy evaluation and improvement, it is crucial for the environment model to have generalization ability in counterfactual queries, as we need to query accurate feedback from $M$ for numerous different policies. Referring to the formulation of WERM through IPS in Def. 2.1, these requirements necessitate minimizing counterfactual-query risks for $M$ under multiple unknown policies, rather than focusing on *a specific target policy $\beta$*. More specifically, the question is: If $\beta$ is unknown and can be varied, how can we generally reduce the risks in counterfactual queries? In this article, we call the model learning problem in this setting "counterfactual environment model learning" and propose a new objective to address the issue. To be compatible with multi-step environment model learning, we first define a generalized WERM through IPS based on Def. 2.1:

**Definition 4.1.** In an MDP, given a transition function $M^*$ that satisfies $M^*(x'|x, a) > 0, \forall x \in \mathcal{X}, \forall a \in \mathcal{A}, \forall x' \in \mathcal{X}$ and $\mu$ satisfies $\mu(a|x) > 0, \forall a \in \mathcal{A}, \forall x \in \mathcal{X}$, the learning objective of generalized WERM through IPS is:

$$\min_{M \in \mathcal{M}} \mathbb{E}_{x,a,x' \sim \rho_{M^*}^{\mu}}[\omega(x, a, x')\ell_M(x, a, x')], \tag{2}$$

where $\omega(x, a, x') = \frac{\rho_{M^*}^{\beta}(x,a,x')}{\rho_{M^*}^{\mu}(x,a,x')}$, $\rho_{M^*}^{\mu}$ and $\rho_{M^*}^{\beta}$ the training and testing data distributions collected by policy $\mu$ and $\beta$ respectively. We define $\ell_M(x, a, x') := \ell(M(x'|x, a), x')$ for brevity.

In an MDP, for any given policy $\pi$, we have $\rho_{M^*}^{\pi}(x, a, x') = \rho_{M^*}^{\pi}(x)\pi(a|x)M^*(x'|x, a)$ where $\rho_{M^*}^{\pi}(x)$ denotes the occupancy measure of $x$ for policy $\pi$. This measure can be defined as $(1 - \gamma)\mathbb{E}_{x_0 \sim \rho_0}[\sum_{t=0}^{\infty} \gamma^t \Pr(x_t = x|x_0, M^*)]$ [51, 25] where $\Pr^{\pi}[x_t = x|x_0, M^*]$ is the discounted state visitation probability that the policy $\pi$ visits $x$ at time-step $t$ by executing in the environment $M^*$ and starting at the state $x_0$. Here $\gamma \in [0, 1]$ is the discount factor. We also define $\rho_{M^*}^{\pi}(x, a) := \rho_{M^*}^{\pi}(x)\pi(a|x)$ for simplicity.

With this definition, $\omega$ can be rewritten: $\omega = \frac{\rho_{M^*}^{\beta}(x)\beta(a|x)M^*(x'|x,a)}{\rho_{M^*}^{\mu}(x)\mu(a|x)M^*(x'|x,a)} = \frac{\rho_{M^*}^{\beta}(x,a)}{\rho_{M^*}^{\mu}(x,a)}$. In single-step environments, for any policy $\pi$, $\rho_{M^*}^{\pi}(x) = \rho_0(x)$. Consequently, we obtain $\omega = \frac{\rho_0(x)\beta(a|x)}{\rho_0(x)\mu(a|x)} = \frac{\beta(a|x)}{\mu(a|x)}$, and the objective degrade to Eq. (1). Therefore, Def. 2.1 is a special case of this generalized form.

**Remark 4.2.** $\omega$ is referred to as density ratio and is commonly used to correct the weighting of rewards in off-policy datasets to estimate the value of a specific target policy in off-policy evaluation

[35, 60]. Recent studies in offline RL also provide similar evidence through upper bound analysis, suggesting that offline model learning should be corrected to specific target policies' distribution using $\omega$ [57]. We derive the objective from the perspective of selection bias correlation, further demonstrate the necessity and effects of this term.

In contrast to previous studies, in this article, we would like to propose an objective for faithful model learning which can generally reduce the risks in counterfactual queries in scenarios where $\beta$ *is unknown and can be varied*. To address the problem, we introduce adversarial policies that can *iteratively* induce the worst prediction performance of the current model and propose to optimize WERM under the adversarial policies. In particular, we propose **A**dversarial **W**eighted empirical **R**isk **M**inimization (AWRM) based on Def. 4.1 to handle this problem.

**Definition 4.3.** Given the MDP transition function $M^*$, the learning objective of adversarial weighted empirical risk minimization through IPS is formulated as

$$\min_{M \in \mathcal{M}} \max_{\beta \in \Pi} L(\rho_{M^*}^{\beta}, M) = \min_{M \in \mathcal{M}} \max_{\beta \in \Pi} \mathbb{E}_{x,a,x' \sim \rho_{M^*}^{\mu}} [\omega(x,a|\rho_{M^*}^{\beta}) \ell_M(x,a,x')], \tag{3}$$

where $\omega(x,a|\rho_{M^*}^{\beta}) = \frac{\rho_{M^*}^{\beta}(x,a)}{\rho_{M^*}^{\mu}(x,a)}$, and the re-weighting term $\omega(x,a|\rho_{M^*}^{\beta})$ is conditioned on the distribution $\rho_{M^*}^{\beta}$ of the adversarial policy $\beta$. In the following, we will ignore $\rho_{M^*}^{\beta}$ and use $\omega(x,a)$ for brevity.

In a nutshell, Eq. (3) minimizes the maximum model loss under all counterfactual data distributions $\rho_{M^*}^{\beta}, \beta \in \Pi$ to guarantee the generalization ability for counterfactual queried by policies in $\Pi$.

## 4.2 Surrogate AWRM through Optimal Adversarial Data Distribution Approximation

The main challenge of solving AWRM is constructing the data distribution $\rho_{M^*}^{\beta^*}$ of the best-response policy $\beta^*$ in $M^*$ since obtaining additional data from $M^*$ can be expensive in real-world applications. In this paper, instead of deriving the optimal $\beta^*$, our solution is to offline estimate the optimal adversarial distribution $\rho_{M^*}^{\beta^*}(x,a,x')$ with respect to $M$, enabling the construction of a surrogate objective to optimize $M$ without directly querying the real environment $M^*$.

In the following, we select $\ell_M$ as the negative log-likelihood loss for our full derivation, instantiating $L(\rho_{M^*}^{\beta}, M)$ in Eq. (3) as: $\mathbb{E}_{x,a \sim \rho_{M^*}^{\mu}}[\omega(x,a|\rho_{M^*}^{\beta}) \mathbb{E}_{M^*}(-\log M(x'|x,a))]$, where $\mathbb{E}_{M^*}[\cdot]$ denotes $\mathbb{E}_{x' \sim M^*(x'|x,a)}[\cdot]$. Ideally, for any given $M$, it is obvious that the optimal $\beta$ is the one that makes $\rho_{M^*}^{\beta}(x,a)$ assign all densities to the point with the largest negative log-likelihood. However, this maximization process is impractical, particularly in continuous spaces. To provide a tractable yet relaxed solution, we introduce an $L_2$ regularizer to the original objective in Eq. (3).

$$\min_{M \in \mathcal{M}} \max_{\beta \in \Pi} \bar{L}(\rho_{M^*}^{\beta}, M) = \min_{M \in \mathcal{M}} \max_{\beta \in \Pi} L(\rho_{M^*}^{\beta}, M) - \frac{\alpha}{2} \|\rho_{M^*}^{\beta}(\cdot,\cdot)\|_2^2, \tag{4}$$

where $\alpha$ denotes the regularization coefficient of $\rho_{M^*}^{\beta}$ and $\|\rho_{M^*}^{\beta}(\cdot,\cdot)\|_2^2 = \int_{\mathcal{X},\mathcal{A}} (\rho_{M^*}^{\beta}(x,a))^2 \mathrm{d}a \mathrm{d}x$.

Now we present the final results and the intuitions behind them, while providing a full derivation in Appx.A. Supposing we have $\bar{\rho}_{M^*}^{\bar{\beta}^*}$ representing the approximated data distribution of the approximated best-response policy $\bar{\beta}^*$ under model $M_\theta$ parameterized by $\theta$, we can find the optimal $\theta^*$ of $\min_\theta \max_{\beta \in \Pi} \bar{L}(\rho_{M^*}^{\beta}, M_\theta)$ (Eq. (4)) through iterative optimization of the objective $\theta_{t+1} = \min_\theta \bar{L}(\bar{\rho}_{M^*}^{\bar{\beta}^*}, M_\theta)$. To this end, we approximate $\bar{\rho}_{M^*}^{\bar{\beta}^*}$ via the last-iteration model $M_{\theta_t}$ and derive an upper bound objective for $\min_\theta \bar{L}(\bar{\rho}_{M^*}^{\bar{\beta}^*}, M_\theta)$:

$$\theta_{t+1} = \min_\theta \mathbb{E}_{\rho_{M^*}^{\mu}} \left[ \frac{-1}{\alpha_0(x,a)} \log M_\theta(x'|x,a) \underbrace{\left( \underbrace{f\left( \frac{\rho_{M_{\theta_t}}^{\mu}(x,a,x')}{\rho_{M^*}^{\mu}(x,a,x')} \right)}_{\text{discrepancy}} - \underbrace{f\left( \frac{\rho_{M_{\theta_t}}^{\mu}(x,a)}{\rho_{M^*}^{\mu}(x,a)} \right)}_{\text{density-ratio baseline}} + \underbrace{H_{M^*}(x,a)}_{\text{stochasticity}} \right)}_{W(x,a,x')} \right], \tag{5}$$

where $\mathbb{E}_{\rho_{M^*}^{\mu}}[\cdot]$ denotes $\mathbb{E}_{x,a,x' \sim \rho_{M^*}^{\mu}}[\cdot]$, $f$ is a convex and lower semi-continuous (l.s.c.) function satisfying $f'(x) \le 0, \forall x \in \mathcal{X}$, which is also called $f$ function in $f$-divergence [2], $\alpha_0(x,a) = \alpha_{M_{\theta_t}} \rho_{M^*}^{\mu}(x,a)$, and $H_{M^*}(x,a)$ denotes the entropy of $M^*(\cdot|x,a)$.

**Remark 4.4** (Intuition of $W$). After derivation, we found that the optimal adversarial data distribution can be approximated by $\bar{\rho}_{M^*}^{\bar{\beta}^*}(x,a) = \int_{\mathcal{X}} \rho_{M^*}^{\mu}(x,a,x')W(x,a,x')\mathrm{d}x'$ (see Appx. A), leading to the upper-bound objective Eq. (5), which is a WERM dynamically weighting by $W$. Intuitively, $W$ assigns more learning propensities for data points with (1) larger discrepancy between $\rho_{M_{\theta_t}}^{\mu}$ (generated by model) and $\rho_{M^*}^{\mu}$ (real-data distribution), or (2) larger stochasticity of the real model $M^*$. The latter is contributed by the entropy $H_{M^*}$, while the former is contributed by the first two terms combined. In particular, through the definition of $f$-divergence, we known that the discrepancy of two distribution $P$ and $Q$ can be measured by $\int_{\mathcal{X}} Q(x)f(P(x)/Q(x))\mathrm{d}x$, thus the terms $f(\rho_{M_{\theta_t}}^{\mu}(x,a,x')/\rho_{M^*}^{\mu}(x,a,x'))$ can be interpreted as the discrepancy measure unit between $\rho_{M_{\theta_t}}^{\mu}(x,a,x')$ and $\rho_{M^*}^{\mu}(x,a,x')$, while $f(\rho_{M_{\theta_t}}^{\mu}(x,a)/\rho_{M^*}^{\mu}(x,a))$ serves as a baseline on $x$ and $a$ measured by $f$ to balance the discrepancy contributed by $x$ and $a$, making $M$ focus on errors on $x'$.

In summary, by adjusting the weights $W$, the learning process will iteratively exploit subtle errors of the current model in any data point, *regardless of how many proportions it contributes in the original data distribution, to eliminate potential unidentifiability on counterfactual data caused by selection bias.*

### 4.3 Tractable Solution

In Eq. (5), the terms $f(\rho_{M_{\theta_t}}^{\mu}(x,a,x')/\rho_{M^*}^{\mu}(x,a,x')) - f(\rho_{M_{\theta_t}}^{\mu}(x,a)/\rho_{M^*}^{\mu}(x,a))$ are still intractable. Thanks to previous successful practices in GAN [19] and GAIL [25], we achieve the objective via a generator-discriminator-paradigm objective through similar derivation. We show the results as follows and leave the complete derivation in Appx. A.4. In particular, by introducing two discriminators $D_{\varphi_0^*}(x,a,x')$ and $D_{\varphi_0^*}(x,a)$, we can optimize the surrogate objective Eq. (5) via:

$$\theta_{t+1} = \max_{\theta} \left( \mathbb{E}_{\rho_{M_{\theta_t}}^{\hat{\mu}}} \left[ A_{\varphi_0^*,\varphi_1^*}(x,a,x') \log M_\theta(x'|x,a) \right] + \mathbb{E}_{\rho_{M^*}^{\mu}} \left[ (H_{M^*}(x,a) - A_{\varphi_0^*,\varphi_1^*}(x,a,x')) \log M_\theta(x'|x,a) \right] \right)$$

$$s.t. \quad \varphi_0^* = \arg\max_{\varphi_0} \left( \mathbb{E}_{\rho_{M^*}^{\mu}} \left[ \log D_{\varphi_0}(x,a,x') \right] + \mathbb{E}_{\rho_{M_{\theta_t}}^{\hat{\mu}}} \left[ \log(1 - D_{\varphi_0}(x,a,x')) \right] \right)$$

$$\varphi_1^* = \arg\max_{\varphi_1} \left( \mathbb{E}_{\rho_{M^*}^{\mu}} \left[ \log D_{\varphi_1}(x,a) \right] + \mathbb{E}_{\rho_{M_{\theta_t}}^{\hat{\mu}}} \left[ \log(1 - D_{\varphi_1}(x,a)) \right] \right), \tag{6}$$

where $\mathbb{E}_{\rho_M^{\mu}}[\cdot]$ is a simplification of $\mathbb{E}_{x,a,x' \sim \rho_M^{\mu}}[\cdot]$, $A_{\varphi_0^*,\varphi_1^*}(x,a,x') = \log D_{\varphi_0^*}(x,a,x') - \log D_{\varphi_1^*}(x,a)$, and $\varphi_0$ and $\varphi_1$ are the parameters of $D_{\varphi_0}$ and $D_{\varphi_1}$ respectively. We learn a policy $\hat{\mu} \approx \mu$ via imitation learning based on the offline dataset $\mathcal{D}_{\text{real}}$ [40, 25]. Note that in the process, we ignore the term $\alpha_0(x,a)$ for simplifying the objective. The discussion on the impacts of removing $\alpha_0(x,a)$ is left in App. B.

The overall optimization pipeline is illustration in Fig. 3. In Eq. (6), the reweighting term $W$ from Eq. (5) is split into two terms in the RHS of the equation: the first term is a GAIL-style objective [25], treating $M_\theta$ as the policy generator, $\hat{\mu}$ as the environment, and $A$ as the advantage function, while the second term is WERM through $H_{M^*} - A_{\varphi_0^*,\varphi_1^*}$. The first term resembles the previous adversarial model learning objectives [46, 47, 11]. These two terms have intuitive explanations: *the first term* assigns learning weights on data generated by the model $M_{\theta_t}$. If the predictions of the model appear realistic, mainly assessed by $D_{\varphi_0^*}$, the propensity weights would be increased, encouraging the model to generate more

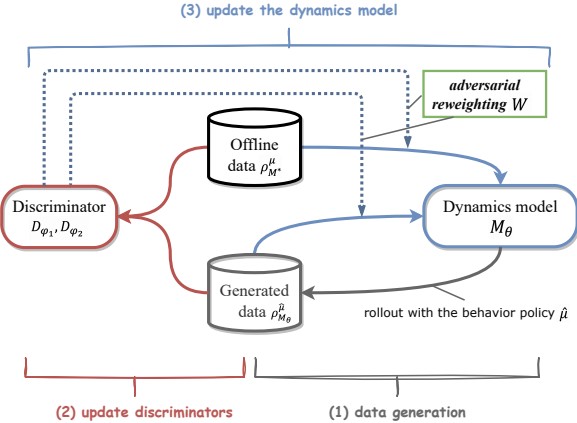

Figure 3: Illustration of the GALILEO workflow.

such kind of data; Conversely, *the second term* assigns weights on real data generated by $M^*$. If the model's predictions seem unrealistic (mainly assessed by $-D_{\varphi_0^*}$) or stochastic (evaluated by $H_{M^*}$), the propensity weights will be increased, encouraging the model to pay more attention to these real data points when improving the likelihoods.

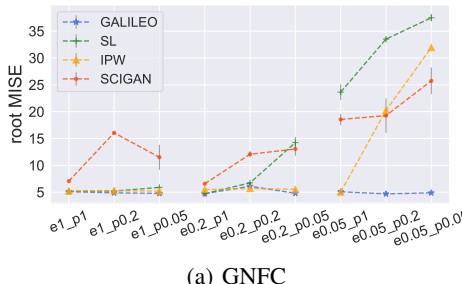
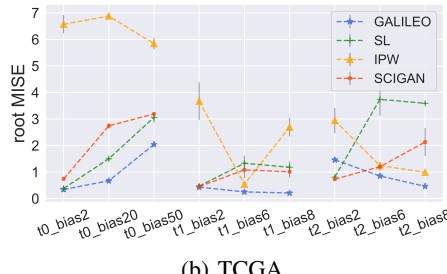

(a) GNFC

(b) TCGA

Figure 4: Illustration of the performance in GNFC and TCGA. The grey bar denotes the standard error ($\times 0.3$ for brevity) of 3 random seeds.

Based on the above implementations, we propose **G**enerative **A**dversarial off**LI**ne counterfactua**L** **E**nvironment m**O**del learning (GALILEO) for environment model learning. We list a brief of GALILEO in Alg. 1 and the details in Appx. E.

---

**Algorithm 1** Pseudocode for GALILEO

---

**Input**:
$\mathcal{D}_{\text{real}}$: offline dataset sampled from $\rho_{M^*}^\mu$ where $\mu$ is the behavior policy; $N$: total iterations;
**Process**:

1: Approximate a behavior policy $\hat{\mu}$ via behavior cloning through offline dataset $\mathcal{D}_{\text{real}}$
2: Initialize an environment model $M_{\theta_1}$
3: **for** $t = 1 : N$ **do**
4:     Use $\hat{\mu}$ to generate a dataset $\mathcal{D}_{\text{gen}}$ with the model $M_{\theta_t}$
5:     Update the discriminators $D_{\varphi_0}$ and $D_{\varphi_1}$ through the second and third equations in Eq. (6) where $\rho_{M_{\theta_t}}^{\hat{\mu}}$ is estimated by $\mathcal{D}_{\text{gen}}$ and $\rho_{M^*}^\mu$ is estimated by $\mathcal{D}_{\text{real}}$
6:     Generative adversarial training for $M_{\theta_t}$ by regarding $A_{\varphi_0^*,\varphi_1^*}$ as the advantage function and computing the gradient to $M_{\theta_t}$, named $g_{\text{pg}}$, with a standard policy gradient method like TRPO [44] or PPO [45] based on $\mathcal{D}_{\text{gen}}$.
7:     Regard $H_{M^*} - A_{\varphi_0^*,\varphi_1^*}$ as the reweighting term for WERM and compute the gradient to $M_{\theta_t}$ based on $\mathcal{D}_{\text{real}}$. Record it as $g_{\text{sl}}$.
8:     Update the model $\theta_{t+1} \leftarrow \theta_t + g_{\text{pg}} + g_{\text{sl}}$.
9: **end for**

---

## 5  Experiments

In this section, we first conduct experiments in two synthetic environments to quantify the performance of GALILEO on counterfactual queries [3]. Then we deploy GALILEO in two complex environments: MuJoCo in Gym [53] and a real-world food-delivery platform to test the performance of GALILEO in difficult tasks. The results are in Sec. 5.2. Finally, to further verify the abiliy GALILEO, in Sec. 5.3, we apply models learned by GALILEO to several downstream tasks including off-policy evaluation, offline policy improvement, and online decision-making in production environment. The algorithms compared are: (1) **SL**: using standard empirical risk minimization for model learning; (2) **IPW** [50]: a standard implementation of WERM based IPS; (3) **SCIGAN** [7]: an adversarial algorithms for model learning used for causal effect estimation, which can be roughly regarded as a partial implementation of GALILEO (Refer to Appx. E.2). We give a detailed description in Appx. G.2.

### 5.1  Environment Settings

**Synthetic Environments**    Previous experiments on counterfactual environment model learning are based on single-step semi-synthetic data simulation [7]. As GALILEO is compatible with *single-step* environment model learning, we first benchmark GALILEO in the same task named TCGA as previous studies do [7]. Based on the three synthetic response functions, we construct 9 tasks by choosing different parameters of selection bias on $\mu$ which is constructed with beta distribution, and

---

[3]code https://github.com/xionghuichen/galileo.

design a coefficient $c$ to control the selection bias. We name the tasks with the format of "t?_bias?". For example, `t1_bias2` is the task with the *first* response functions and $c = 2$. The details of TCGA is in Appx. G.1.2. Besides, for *sequential* environment model learning under selection bias, we construct a new synthetic environment, general negative feedback control (GNFC), which can represent a classic type of task with policies having selection bias, where Fig. 1(a) is also an instance of GNFC. We construct 9 tasks on GNFC by adding behavior policies $\mu$ with different scales of uniform noise $U(-e, e)$ with probabilities $p$. Similarly, we name them with the format "e?_p?".

**Continuous-control Environments**   We select 3 MuJoCo environments from D4RL [17] to construct our model learning tasks. We compare it with a standard transition model learning algorithm used in the previous offline model-based RL algorithms [59, 30], which is a standard supervised learning. We name the method OFF-SL. Besides, we also implement IPW and SCIGAN as the baselines. In D4RL benchmark, only the "medium" tasks is collected with a fixed policy, i.e., the behavior policy is with 1/3 performance to the expert policy), which is most matching to our proposed problem. So we train models in datasets HalfCheetah-medium, Walker2d-medium, and Hopper-medium.

**A Real-world Large-scale Food-delivery Platform**   We finally deploy GALILEO in a real-world large-scale food-delivery platform. We focus on a Budget Allocation task to the Time period (BAT) in the platform (see Appx. G.1.3 for details). The goal of the BAT task is to handle the imbalance problem between the demanded orders from customers and the supply of delivery clerks in different time periods by allocating reasonable allowances to those time periods. The challenge of the environment model learning in BAT tasks is similar to the challenge in Fig. 1: the behavior policy is a human-expert policy, which tends to increase the budget of allowance in the time periods with a lower supply of delivery clerks, otherwise tends to decrease the budget (We gives a real-data instance in Appx. G.1.3).

## 5.2   Prediction Accuracy on Shifted Data Distributions

**Test in Synthetic Environments**   For all of the tasks, we select mean-integrated-square error $\text{MISE} = \mathbb{E}\left[\int_{\mathcal{A}} \left(M^*(x'|x, a) - M(x'|x, a)\right)^2 \mathrm{d}a\right]$ as the metric, which is a metric to measure the accuracy in counterfactual queries by considering the prediction errors in the whole action space. The results are summarized in Fig. 4 and the detailed results can be found in Appx. H. The results show that the property of the behavior policy (i.e., $e$ and $p$) dominates the generalization ability of the baseline algorithms. When $e = 0.05$, almost all of the baselines fail and give a completely opposite response curve, while GALILEO gives the correct response. (see Fig. 5). IPW still performs well when $0.2 \leq e \leq 1.0$ but fails when $e = 0.05, p <= 0.2$. We also found that SCIGAN can reach a better performance than other baselines when $e = 0.05, p <= 0.2$, but the results in other tasks are unstable. GALILEO is the only algorithm that is robust to the selection bias and outputs correct response curves in all of the tasks. Based on the experiment, we also indicate that the commonly used overlap assumption is unreasonable to a certain extent especially in real-world applications since it is impractical to inject noises into the whole action space.

The problem of overlap assumption being violated, e.g., $e < 1$ in our setting, should be taken into consideration otherwise the algorithm will be hard to use in practice if it is sensitive to the noise range. On the other hand, we found the phenomenon in TCGA experiment is similar to the one in GNFC, which demonstrates the compatibility of GALILEO to single-step environments.

We also found that the results of IPW are unstable in TCGA experiment. It might be because the behavior policy is modeled with beta distribution while the propensity score $\hat{\mu}$ is modeled with Gaussian distribution. Since IPW directly reweight loss with $\frac{1}{\hat{\mu}}$, the results are sensitive to the error of $\hat{\mu}$.

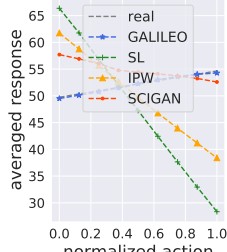

Figure 5: Illustration of the averaged response curves in task `e0.05_p0.2`.

Finally, we plot the averaged response curves which are constructed by equidistantly sampling action from the action space and averaging the feedback of the states in the dataset as the averaged response. One result is in Fig. 5 (all curves can be seen in Appx. H). For those tasks where baselines fail in reconstructing response curves, GALILEO not only reaches a better MISE score but reconstructs almost exact responses, while the baselines might give completely opposite responses.

**Test in MuJoCo Benchmarks**   We test the prediction error of the learned model in corresponding unseen "expert" and "medium-replay" datasets. Fig. 6 illustrates the results in halfcheetah. We can see that all algorithms perform well in the training datasets. OFF-SL can even reach a bit lower error. However, when we verify the models through "expert" and "medium-replay" datasets, which

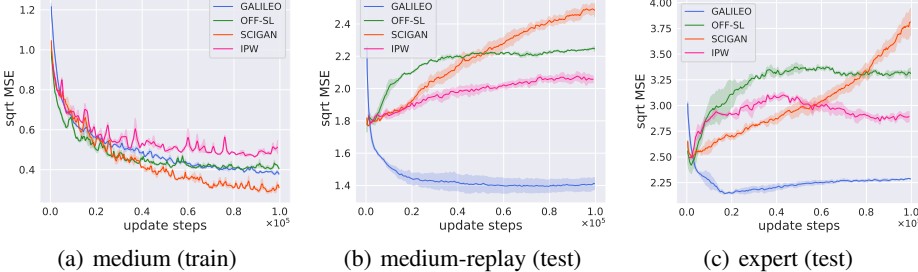

(a) medium (train)  (b) medium-replay (test)  (c) expert (test)

Figure 6: Illustration of learning curves of the halfcheetah tasks (full results are in Appx. H.5). The figures with titles ending in "(train)" means the dataset is used for training while the titles ending in "(test)" means the dataset is **just used for testing**. The X-axis records the steps of the environment model update, and the Y-axis is the prediction errors in the corresponding steps evaluated by the datasets. The solid curves are the mean reward and the shadow is the standard error of three seeds.

are collected by other policies, the performance of GALILEO is more stable and better than all other algorithms. As the training continues, the baseline algorithms even gets worse and worse. The phenomenon are similar among three datasets, and we leave the full results in Appx. H.5.

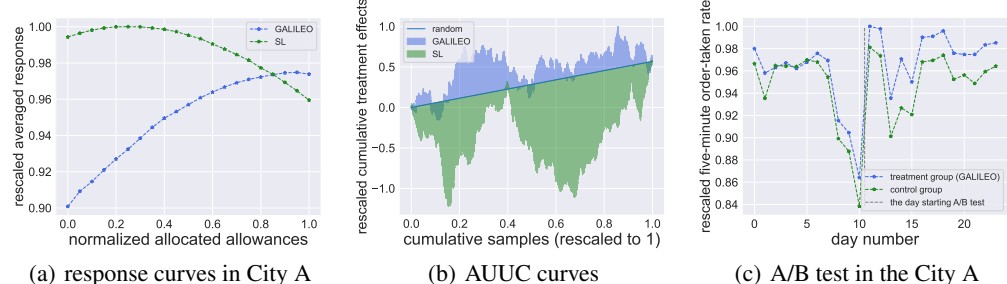

(a) response curves in City A  (b) AUUC curves  (c) A/B test in the City A

Figure 7: Parts of the results in BAT tasks. Fig. 7(a) demonstrate the averaged response curves of the SL and GALILEO model in City A. It is expected to be monotonically increasing through our prior knowledge. In Fig. 7(b) show the AUCC curves, where the model with larger areas above the "random" line makes better predictions in randomized-controlled-trials data [61].

**Test in a Real-world Dataset**  We first learn a model to predict the supply of delivery clerks (measured by fulfilled order amount) on given allowances. Although the SL model can efficiently fit the offline data, the tendency of the response curve is easily to be incorrect. As can be seen in Fig. 7(a), with a larger budget of allowance, the prediction of the supply is decreased in SL, which obviously goes against our prior knowledge. This is because, in the offline dataset, the corresponding supply will be smaller when the allowance is larger. It is conceivable that if we learn a policy through the model of SL, the optimal solution is canceling all of the allowances, which is obviously incorrect in practice. On the other hand, the tendency of GALILEO's response is correct. In Appx. H.7, we plot all the results in 6 cities. We further collect some randomized controlled trials data, and the Area Under the Uplift Curve (AUUC) [6] curve in Fig. 7(b) verify that GALILEO gives a reasonable sort order on the supply prediction while the standard SL technique fails to achieve this task.

## 5.3 Apply GALILEO to Downstream Tasks

**Off-policy Evaluation (OPE)**  We first verify the ability of the models in MuJoCo environments by adopting them into off-policy evaluation tasks. We use 10 unseen policies constructed by DOPE benchmark [18] to conduct our experiments.  We select three common-used metrics: value gap, regret@1, and rank correlation and averaged the results among three tasks in Tab. 1.  The baselines and the corre-

Table 1: Results of OPE on DOPE benchmark. We list the **averaged** performances on three tasks. The detailed results are in Appx. H.6. ± is the standard deviation among the tasks. We bold the best scores for each metric.

| Algorithm | Norm. value gap | Rank corr. | Regret@1 |
|---|---|---|---|
| GALILEO | **0.37 ± 0.24** | **0.44 ± 0.10** | **0.09 ± 0.02** |
| Best DICE | 0.48 ± 0.19 | 0.15 ± 0.04 | 0.42 ± 0.28 |
| VPM | 0.71 ± 0.04 | 0.29 ± 0.15 | 0.17 ± 0.11 |
| FQE (L2) | 0.54 ± 0.09 | -0.19 ± 0.10 | 0.34 ± 0.03 |
| IS | 0.67 ± 0.01 | -0.40 ± 0.15 | 0.36 ± 0.27 |
| Doubly Rubost | 0.57 ± 0.07 | -0.14 ± 0.17 | 0.33 ± 0.06 |

Table 2: Results of policy performance directly optimized through SAC [20] using the learned dynamics models and deployed in MuJoCo environments. MAX-RETURN is the policy performance of SAC in the MuJoCo environments, and "avg. norm." is the averaged normalized return of the policies in the 9 tasks, where the returns are normalized to lie between 0 and 100, where a score of 0 corresponds to the worst policy, and 100 corresponds to MAX-RETURN.

| Task | Hopper | | | Walker2d | | | HalfCheetah | | | avg. norm. |
|---|---|---|---|---|---|---|---|---|---|---|
| Horizon | H=10 | H=20 | H=40 | H=10 | H=20 | H=40 | H=10 | H=20 | H=40 | / |
| GALILEO | **13.0 ± 0.1** | **33.2 ± 0.1** | **53.5 ± 1.2** | **11.7 ± 0.2** | **29.9 ± 0.3** | **61.2 ± 3.4** | 0.7 ± 0.2 | -1.1 ± 0.2 | -14.2 ± 1.4 | **51.1** |
| OFF-SL | 4.8 ± 0.5 | 3.0 ± 0.2 | 4.6 ± 0.2 | 10.7 ± 0.2 | 20.1 ± 0.8 | 37.5 ± 6.7 | 0.4 ± 0.5 | -1.1 ± 0.6 | -13.2 ± 0.3 | 21.1 |
| IPW | 5.9 ± 0.7 | 4.1 ± 0.6 | 5.9 ± 0.2 | 4.7 ± 1.1 | 2.8 ± 3.9 | 14.5 ± 1.4 | **1.6 ± 0.2** | **0.5 ± 0.8** | **-11.3 ± 0.9** | 19.7 |
| SCIGAN | 12.7 ± 0.1 | 29.2 ± 0.6 | 46.2 ± 5.2 | 8.4 ± 0.5 | 9.1 ± 1.7 | 1.0 ± 5.8 | 1.2 ± 0.3 | -0.3 ± 1.0 | -11.4 ± 0.3 | 41.8 |
| MAX-RETURN | 13.2 ± 0.0 | 33.3 ± 0.2 | 71.0 ± 0.5 | 14.9 ± 1.3 | 60.7 ± 11.1 | 221.1 ± 8.9 | 2.6 ± 0.1 | 13.3 ± 1.1 | 49.1 ± 2.3 | 100.0 |

sponding results we used are the same as the one proposed by [18]. As seen in Tab. 1, compared with all the baselines, OPE by GALILEO always reach the better performance with *a large margin (at least 23%, 193% and 47% respectively)*, which verifies that GALILEO can eliminate the effect of selection bias and give correct evaluations on unseen policies.

**Offline Policy Improvement**  We then verify the generalization ability of the models in MuJoCo environments by adopting them into offline model-based RL. To strictly verify the ability of the models, we abandon all tricks to suppress policy exploration and learning in risky regions as current offline model-based RL algorithms [59] do, and we just use the standard SAC algorithm [20] to fully exploit the models to search an optimal policy. Unfortunately, we found that the compounding error will still be inevitably large in the 1,000-step rollout, which is the standard horizon in MuJoCo tasks, leading all models to fail to derive a reasonable policy. To better verify the effects of models on policy improvement, we learn and evaluate the policies with three smaller horizons: $H \in \{10, 20, 40\}$. The results are listed in Tab. 2. We first averaged the normalized return (refer to "avg. norm.") under each task, and we can see that the policy obtained by GALILEO is significantly higher than other models (the improvements are 24% to 161%). But in HalfCheetah, IPW works slightly better. However, compared with MAX-RETURN, it can be found that all methods fail to derive reasonable policies because their policies' performances are far away from the optimal policy. By further checking the trajectories, we found that all the learned policies just keep the cheetah standing in the same place or even going backward. This phenomenon is also similar to the results in MOPO [59]. In MOPO's experiment in the medium datasets, the truncated-rollout horizon used in Walker and Hopper for policy training is set to 5, while HalfCheetah has to be set to *the minimal value: 1*. These phenomena indicate that HalfCheetah may still have unknown problems, resulting in the generalization bottleneck of the models.

**Online Decision-making in a Production Environment**  Finally, we search for the optimal policy via model-predict control (MPC) using cross-entropy planner [21] based on the learned model and deploy the policy in a real-world platform. The results of A/B test in City A is shown in Fig. 7(c). It can be seen that after the day of the A/B test, the treatment group (deploying our policy) significant improve the five-minute order-taken rate than the baseline policy (the same as the behavior policy). In summary, *the policy improves the supply from 0.14 to 1.63 percentage points to the behavior policies in the 6 cities*. The details of these results are in Appx. H.7.

## 6  Discussion and Future Work

In this work, we propose AWRM which handles the generalization challenges of the counterfactual environment model learning. By theoretical modeling, we give a tractable solution to handle AWRM and propose GALILEO. GALILEO is verified in synthetic environments, complex robot control tasks, and a real-world platform, and shows great generalization ability on counterfactual queries.

Giving correct answers to counterfactual queries is important for policy learning. We hope the work can inspire researchers to develop more powerful tools for counterfactual environment model learning. The current limitation lies in: There are several simplifications in the theoretical modeling process (further discussion is in Appx. B), which can be modeled more elaborately. Besides, experiments on MuJoCo indicate that these tasks are still challenging to give correct predictions on counterfactual data. These should also be further investigated in future work.

## Acknowledgements and Disclosure of Funding

This work is supported by the National Key Research and Development Program of China (2020AAA0107200) and the National Science Foundation of China (61921006).

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

# Appendix

## Table of Contents

# A Proof of Theoretical Results

In the proof section, we replace the notation of $\mathbb{E}$ with an integral for brevity. Now we rewrite the original objective $\bar{L}(\rho_{M^*}^\beta, M)$ as:

$$\min_{M \in \mathcal{M}} \max_{\beta \in \Pi} \int_{\mathcal{X},\mathcal{A}} \rho_{M^*}^\mu(x,a)\omega(x,a) \int_{\mathcal{X}} M^*(x'|x,a)\left(-\log M(x'|x,a)\right) \mathrm{d}x'\mathrm{d}a\mathrm{d}x - \frac{\alpha}{2}\|\rho_{M^*}^\beta(\cdot,\cdot)\|_2^2, \tag{7}$$

where $\omega(x,a) = \frac{\rho_{M^*}^\beta(x,a)}{\rho_{M^*}^\mu(x,a)}$ and $\|\rho_{M^*}^\beta(\cdot,\cdot)\|_2^2 = \int_{\mathcal{X},\mathcal{A}} \rho_{M^*}^\beta(x,a)^2 \mathrm{d}a\mathrm{d}x$, which is the squared $l_2$-norm. In an MDP, given any policy $\pi$, $\rho_{M^*}^\pi(x,a,x') = \rho_{M^*}^\pi(x)\pi(a|x)M^*(x'|x,a)$ where $\rho_{M^*}^\pi(x)$ denotes the occupancy measure of $x$ for policy $\pi$, which can be defined [51, 25] as $\rho_{M^*}^\pi(x) := (1-\gamma)\mathbb{E}_{x_0 \sim \rho_0}\left[\sum_{t=0}^{\infty} \gamma^t \Pr(x_t = x|x_0, M^*)\right]$ where $\Pr^\pi[x_t = x|x_0, M^*]$ is the state visitation probability that $\pi$ starts at state $x_0$ in model $M^*$ and receive $x$ at timestep $t$ and $\gamma \in [0,1]$ is the discount factor.

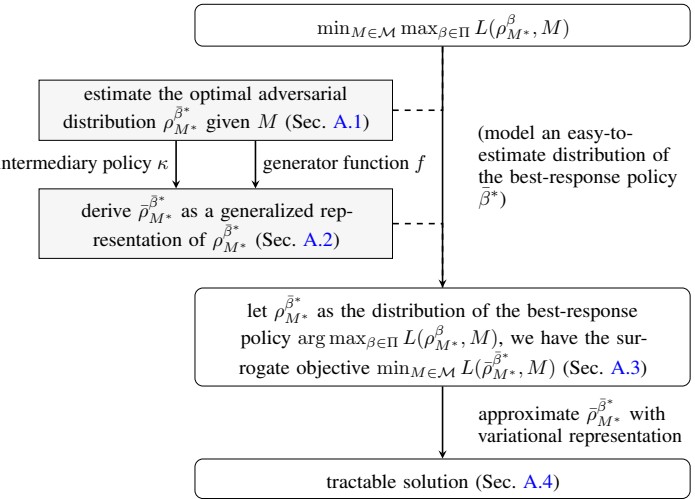

Figure 8: The overall pipeline to model the tractable solution to AWRM. $f$ is a generator function defined by $f$-divergence [37]. $\kappa$ is an intermediary policy introduced in the estimation.

The overall pipeline to model the tractable solution to AWRM is given in Fig. 8. In the following, we will summarize the modelling process based on Fig. 8. We first approximate the optimal distribution $\rho_{M^*}^{\bar{\beta}^*}$ via Lemma. A.1.

**Lemma A.1.** *Given any $M$ in $\bar{L}(\rho_{M^*}^\beta, M)$, the distribution $\rho_{M^*}^{\bar{\beta}^*}(x,a)$ of the ideal best-response policy $\bar{\beta}^*$ satisfies:*

$$\frac{1}{\alpha_M}(D_{KL}(M^*(\cdot|x,a), M(\cdot|x,a)) + H_{M^*}(x,a)), \tag{8}$$

*where $D_{KL}(M^*(\cdot|x,a), M(\cdot|x,a))$ is the Kullback-Leibler (KL) divergence between $M^*(\cdot|x,a)$ and $M(\cdot|x,a)$, $H_{M^*}(x,a)$ denotes the entropy of $M^*(\cdot|x,a)$, and $\alpha_M$ is the regularization coefficient $\alpha$ in Eq. (4) and also as a normalizer of Eq. (8).*

Note that the ideal best-response policy $\bar{\beta}^*$ is not the real best-response policy $\beta^*$. The distribution $\rho_{M^*}^{\bar{\beta}^*}$ is an approximation of the real optimal adversarial distribution. We give a discussion of the rationality of the ideal best-response policy $\bar{\beta}^*$ as a replacement of the real best-response policy $\beta^*$ in Remark A.4. Intuitively, $\rho_{M^*}^{\bar{\beta}^*}$ has larger densities on the data where the divergence between the approximation model and the real model (i.e., $D_{KL}(M^*(\cdot|x,a), M(\cdot|x,a))$) is larger or the stochasticity of the real model (i.e., $H_{M^*}$) is larger.

However, the integral process of $D_{KL}$ in Eq. (8) is intractable in the offline setting as it explicitly requires the conditional probability function of $M^*$. Our solution to solve the problem is utilizing the offline dataset $\mathcal{D}_{\mathrm{real}}$ as the empirical *joint* distribution $\rho_{M^*}^\mu(x,a,x')$ and adopting practical

techniques for distance estimation on two joint distributions, like GAN [19, 37], to approximate Eq. (8). To adopt that solution, we should first transform Eq. (8) into a form under joint distributions. Without loss of generality, we introduce an intermediary policy $\kappa$, of which $\mu$ can be regarded as a specific instance. Then we have $M(x'|x,a) = \rho_M^\kappa(x,a,x')/\rho_M^\kappa(x,a)$ for any $M$ if $\rho_M^\kappa(x,a) > 0$. Assuming $\forall x \in \mathcal{X}, \forall a \in \mathcal{A}, \rho_{M^*}^\kappa(x,a) > 0$ if $\rho_{M^*}^{\bar{\beta}^*}(x,a) > 0$, which will hold when $\kappa$ overlaps with $\mu$, then Eq. (8) can transform to:

$$\frac{1}{\alpha_0(x,a)}\left(\int_{\mathcal{X}} \rho_{M^*}^\kappa(x,a,x')\log\frac{\rho_{M^*}^\kappa(x,a,x')}{\rho_M^\kappa(x,a,x')}\mathrm{d}x' - \rho_{M^*}^\kappa(x,a)\left(\log\frac{\rho_{M^*}^\kappa(x,a)}{\rho_M^\kappa(x,a)} - H_{M^*}(x,a)\right)\right),$$

where $\alpha_0(x,a) = \alpha_M \rho_{M^*}^\kappa(x,a)$. We notice that the form $\rho_{M^*}^\kappa \log\frac{\rho_{M^*}^\kappa}{\rho_M^\kappa}$ is the integrated function in reverse KL divergence, which is an instance of $f$ function in $f$-divergence [2]. Replacing that form with $f$ function, we obtain a generalized representation of $\rho_{M^*}^{\bar{\beta}^*}$:

$$\bar{\rho}_{M^*}^{\bar{\beta}^*} := \frac{1}{\alpha_0(x,a)}\left(\int_{\mathcal{X}} \rho_{M^*}^\kappa(x,a,x')f\left(\frac{\rho_M^\kappa(x,a,x')}{\rho_{M^*}^\kappa(x,a,x')}\right)\mathrm{d}x' - \rho_{M^*}^\kappa(x,a)\left(f\left(\frac{\rho_M^\kappa(x,a)}{\rho_{M^*}^\kappa(x,a)}\right) - H_{M^*}(x,a)\right)\right),$$

(9)

where $f : \mathbb{R}_+ \to \mathbb{R}$ is a convex and lower semi-continuous (l.s.c.) function. $\bar{\rho}_{M^*}^{\bar{\beta}^*}$ gives a generalized representation of the optimal adversarial distribution to maximize the error of the model. Based on Eq. (9), we have a surrogate objective of AWRM which can avoid querying $M^*$ to construct $\rho_{M^*}^{\beta^*}$:

**Theorem A.2.** *Let $\bar{\rho}_{M^*}^{\bar{\beta}^*}$ as the data distribution of the best-response policy $\bar{\beta}^*$ in Eq. (4) under model $M_\theta$ parameterized by $\theta$, then we can find the optimal $\theta^*$ of $\min_\theta \max_{\beta \in \Pi} \bar{L}(\rho_{M^*}^\beta, M_\theta)$ (Eq. (4)) via iteratively optimizing the objective $\theta_{t+1} = \min_\theta \bar{L}(\bar{\rho}_{M^*}^{\bar{\beta}^*}, M_\theta)$, where $\bar{\rho}_{M^*}^{\bar{\beta}^*}$ is approximated via the last-iteration model $M_{\theta_t}$. Based on Corollary A.9, we derive an upper bound objective for $\min_\theta \bar{L}(\bar{\rho}_{M^*}^{\bar{\beta}^*}, M_\theta)$:*

$$\theta_{t+1} = \min_\theta \mathbb{E}_{\rho_{M^*}^\kappa}\left[\frac{-1}{\alpha_0(x,a)}\log M_\theta(x'|x,a)\underbrace{\left(f\left(\frac{\rho_{M_{\theta_t}}^\kappa(x,a,x')}{\rho_{M^*}^\kappa(x,a,x')}\right) - f\left(\frac{\rho_{M_{\theta_t}}^\kappa(x,a)}{\rho_{M^*}^\kappa(x,a)}\right) + H_{M^*}(x,a)\right)}_{W(x,a,x')}\right],$$

*where $\mathbb{E}_{\rho_{M^*}^\kappa}[\cdot]$ denotes $\mathbb{E}_{x,a,x'\sim\rho_{M^*}^\kappa}[\cdot]$, $f$ is a l.s.c function satisfying $f'(x) \leq 0, \forall x \in \mathcal{X}$, and $\alpha_0(x,a) = \alpha_{M_{\theta_t}}\rho_{M^*}^\kappa(x,a)$.*

Thm. A.2 approximately achieve AWRM by using $\kappa$ and a pseudo-reweighting module $W$. $W$ assigns learning propensities for data points with larger differences between distributions $\rho_{M_{\theta_t}}^\kappa$ and $\rho_{M^*}^\kappa$. By adjusting the weights, the learning process will exploit subtle errors in any data point, whatever how many proportions it contributes, to correct potential generalization errors on counterfactual data.

**Remark A.3.** In practice, we need to use real-world data to construct the distribution $\rho_{M^*}^\kappa$. In the offline model-learning setting, we only have a real-world dataset $\mathcal{D}_{\text{real}}$ collected by the behavior policy $\mu$, which is the empirical distribution of $\rho_{M^*}^\mu$. Let $\kappa = \mu$, we have

$$\theta_{t+1} = \min_\theta \mathbb{E}_{\rho_{M^*}^\mu}\left[\frac{-1}{\alpha_0(x,a)}\log M_\theta(x'|x,a)\underbrace{\left(f\left(\frac{\rho_{M_{\theta_t}}^\mu(x,a,x')}{\rho_{M^*}^\mu(x,a,x')}\right) - f\left(\frac{\rho_{M_{\theta_t}}^\mu(x,a)}{\rho_{M^*}^\mu(x,a)}\right) + H_{M^*}(x,a)\right)}_{W(x,a,x')}\right],$$

*which is Eq. (5) in the main body.*

In Thm. A.2, the terms $f(\rho_{M_{\theta_t}}^\kappa(x,a,x')/\rho_{M^*}^\kappa(x,a,x')) - f(\rho_{M_{\theta_t}}^\kappa(x,a)/\rho_{M^*}^\kappa(x,a))$ are still intractable. Thanks to previous successful practices in GAN [19] and GAIL [25], we achieve the objective via a generator-discriminator-paradigm objective through similar derivation. We show the results as follows and leave the complete derivation in Appx. A.4. In particular, by introducing two discriminators $D_{\varphi_0^*}(x,a,x')$ and $D_{\varphi_0^*}(x,a)$, letting $\kappa = \mu$, we can optimize the surrogate objective Eq. (5) via:

$$\theta_{t+1} = \max_\theta \left(\mathbb{E}_{\rho_{M_{\theta_t}}^{\hat{\mu}}}\left[A_{\varphi_0^*,\varphi_1^*}(x,a,x')\log M_\theta(x'|x,a)\right] + \mathbb{E}_{\rho_{M^*}^\mu}\left[(H_{M^*}(x,a) - A_{\varphi_0^*,\varphi_1^*}(x,a,x'))\log M_\theta(x'|x,a)\right]\right)$$

$$s.t. \quad \varphi_0^* = \arg\max_{\varphi_0}\left(\mathbb{E}_{\rho_{M^*}^\mu}\left[\log D_{\varphi_0}(x,a,x')\right] + \mathbb{E}_{\rho_{M_{\theta_t}}^{\hat{\mu}}}\left[\log(1 - D_{\varphi_0}(x,a,x'))\right]\right)$$

$$\varphi_1^* = \arg\max_{\varphi_1}\left(\mathbb{E}_{\rho_{M^*}^\mu}\left[\log D_{\varphi_1}(x,a)\right] + \mathbb{E}_{\rho_{M_{\theta_t}}^{\hat{\mu}}}\left[\log(1 - D_{\varphi_1}(x,a))\right]\right),$$

where $\mathbb{E}_{\rho_M^\mu}[\cdot]$ is a simplification of $\mathbb{E}_{x,a,x'\sim\rho_M^\mu}[\cdot]$, $A_{\varphi_0^*,\varphi_1^*}(x,a,x') = \log D_{\varphi_0^*}(x,a,x') - \log D_{\varphi_1^*}(x,a)$, and $\varphi_0$ and $\varphi_1$ are the parameters of $D_{\varphi_0}$ and $D_{\varphi_1}$ respectively. We learn a policy $\hat\mu \approx \mu$ via imitation learning based on the offline dataset $\mathcal{D}_{\text{real}}$ [40, 25]. Note that in the process, we ignore the term $\alpha_0(x,a)$ for simplifying the objective. The discussion on the impacts of removing $\alpha_0(x,a)$ is left in App. B.

## A.1 Proof of Lemma A.1

*Proof.* Given a transition function $M$ of an MDP, the distribution of the best-response policy $\beta^*$ satisfies:

$$
\begin{aligned}
\rho_{M^*}^{\beta^*} &= \arg\max_{\rho_{M^*}^\beta} \int_{\mathcal{X},\mathcal{A}} \rho_{M^*}^\mu(x,a)\omega(x,a)\int_{\mathcal{X}} M^*(x'|x,a)\left(-\log M(x'|x,a)\right)\mathrm{d}x'\mathrm{d}a\mathrm{d}x - \frac{\alpha}{2}\|\rho_{M^*}^\beta(\cdot,\cdot)\|_2^2 \\
&= \arg\max_{\rho_{M^*}^\beta} \int_{\mathcal{X},\mathcal{A}} \rho_{M^*}^\beta(x,a)\underbrace{\int_{\mathcal{X}} M^*(x'|x,a)\left(-\log M(x'|x,a)\right)\mathrm{d}x'}_{g(x,a)}\mathrm{d}a\mathrm{d}x - \frac{\alpha}{2}\|\rho_{M^*}^\beta(\cdot,\cdot)\|_2^2 \\
&= \arg\max_{\rho_{M^*}^\beta} \frac{2}{\alpha}\int_{\mathcal{X},\mathcal{A}} \rho_{M^*}^\beta(x,a)g(x,a)\mathrm{d}a\mathrm{d}x - \|\rho_{M^*}^\beta(\cdot,\cdot)\|_2^2 \\
&= \arg\max_{\rho_{M^*}^\beta} \frac{2}{\alpha}\int_{\mathcal{X},\mathcal{A}} \rho_{M^*}^\beta(x,a)g(x,a)\mathrm{d}a\mathrm{d}x - \|\rho_{M^*}^\beta(\cdot,\cdot)\|_2^2 - \frac{\|g(\cdot,\cdot)\|_2^2}{\alpha^2} \\
&= \arg\max_{\rho_{M^*}^\beta} -\left(-2\int_{\mathcal{X},\mathcal{A}} \rho_{M^*}^\beta(x,a)\frac{g(x,a)}{\alpha}\mathrm{d}a\mathrm{d}x + \|\rho_{M^*}^\beta(\cdot,\cdot)\|_2^2 + \frac{\|g(\cdot,\cdot)\|_2^2}{\alpha^2}\right) \\
&= \arg\max_{\rho_{M^*}^\beta} -\|\rho_{M^*}^\beta(\cdot,\cdot) - \frac{g(\cdot,\cdot)}{\alpha}\|_2^2.
\end{aligned}
$$

We know that the occupancy measure $\rho_{M^*}^\beta$ is a density function with a constraint $\int_{\mathcal{X}}\int_{\mathcal{A}}\rho_{M^*}^\beta(x,a)\mathrm{d}a\mathrm{d}x = 1$. Assuming the occupancy measure $\rho_{M^*}^\beta$ has an upper bound $c$, that is $0 \le \rho_{M^*}^\beta(x,a) \le c, \forall a \in \mathcal{A}, \forall x \in \mathcal{X}$, constructing a regularization coefficient $\alpha_M = \int_{\mathcal{X}}\int_{\mathcal{A}}(D_{KL}(M^*(\cdot|x,a),M(\cdot|x,a)) + H_{M^*}(x,a))\mathrm{d}x\mathrm{d}a$ as a constant value given any $M$, then we have

$$
\begin{aligned}
\rho_{M^*}^{\beta^*}(x,a) &= \frac{g(x,a)}{\alpha_M} \\
&= \frac{\int_{\mathcal{X}} M^*(x'|x,a)\log\frac{M^*(x'|x,a)}{M(x'|x,a)}\mathrm{d}x - \int_{\mathcal{X}} M^*(x'|x,a)\log M^*(x'|x,a)\mathrm{d}x}{\alpha_M} \\
&= \frac{D_{KL}(M^*(\cdot|x,a),M(\cdot|x,a)) + H_{M^*}(x,a)}{\alpha_M} \\
&\propto \left(D_{KL}(M^*(\cdot|x,a),M(\cdot|x,a)) + H_{M^*}(x,a)\right),
\end{aligned}
$$

which is the optimal density function of Eq. (7) with $\alpha = \alpha_M$.

Note that in some particular $M^*$, we still cannot construct a $\beta$ that can generate an occupancy specified by $g(x,a)/\alpha_M$ for any $M$. We can only claim the distribution of the ideal best-response policy $\bar\beta^*$ satisfies:

$$
\rho_{M^*}^{\bar\beta^*}(x,a) = \frac{1}{\alpha_M}(D_{KL}(M^*(\cdot|x,a),M(\cdot|x,a)) + H_{M^*}(x,a)), \tag{10}
$$

where $\alpha_M$ is a normalizer that $\alpha_M = \int_{\mathcal{X}}\int_{\mathcal{A}}(D_{KL}(M^*(\cdot|x,a),M(\cdot|x,a)) + H_{M^*}(x,a))\mathrm{d}x\mathrm{d}a$. We give a discussion of the rationality of the ideal best-response policy $\bar\beta^*$ as a replacement of the real best-response policy $\beta^*$ in Remark A.4.

$\square$

**Remark A.4.** The optimal solution Eq. (10) relies on $g(x, a)$. In some particular $M^*$, it is intractable to derive a $\beta$ that can generate an occupancy specified by $g(x, a)/\alpha_M$. Consider the following case: a state $x_1$ in $M^*$ might be harder to reach than another state $x_2$, e.g., $M^*(x_1|x, a) < M^*(x_2|x, a), \forall x \in \mathcal{X}, \forall a \in \mathcal{A}$, then it is impossible to find a $\beta$ that the occupancy satisfies $\rho_{M^*}^\beta(x_1, a) > \rho_{M^*}^\beta(x_2, a)$. In this case, Eq. (10) can be a sub-optimal solution. Since this work focuses on task-agnostic solution derivation while the solution to the above problem should rely on the specific description of $M^*$, we leave it as future work. However, we point out that Eq. (10) is a reasonable re-weighting term even as a sub-optimum: $\rho_{M^*}^{\bar\beta^*}$ gives larger densities on the data where the distribution distance between the approximation model and the real model (i.e., $D_{KL}(M^*, M)$) is larger or the stochasticity of the real model (i.e., $H_{M^*}$) is larger.

## A.2  Proof of Eq. (9)

The integral process of $D_{KL}$ in Eq. (8) is intractable in the offline setting as it explicitly requires the conditional probability function of $M^*$. Our motivation for the tractable solution is utilizing the offline dataset $\mathcal{D}_\text{real}$ as the empirical *joint* distribution $\rho_{M^*}^\mu(x, a, x')$ and adopting practical techniques for distance estimation on two joint distributions, like GAN [19, 37], to approximate Eq. (8). To adopt that solution, we should first transform Eq. (8) into a form under joint distributions. Without loss of generality, we introduce an intermediary policy $\kappa$, of which $\mu$ can be regarded as a specific instance. Then we have $M(x'|x, a) = \rho_M^\kappa(x, a, x')/\rho_M^\kappa(x, a)$ for any $M$ if $\rho_M^\kappa(x, a) > 0$. Assuming $\forall x \in \mathcal{X}, \forall a \in \mathcal{A}, \rho_{M^*}^\kappa(x, a) > 0$ if $\rho_{M^*}^{\bar\beta^*}(x, a) > 0$, which will hold when $\kappa$ overlaps with $\mu$, then Eq. (8) can transform to:

$$
\begin{aligned}
\rho_{M^*}^{\bar\beta^*}(x, a) =& \frac{D_{KL}(M^*(\cdot|x, a), M(\cdot|x, a)) + H_{M^*}(x, a)}{\alpha_M} \\
=& \frac{1}{\alpha_M} \int_{\mathcal{X}} M^*(x'|x, a) \left( \log \frac{M^*(x'|x, a)}{M(x'|x, a)} - \log M^*(x'|x, a) \right) \mathrm{d}x' \\
=& \frac{1}{\alpha_M \rho_{M^*}^\kappa(x, a)} \int_{\mathcal{X}} \rho_{M^*}^\kappa(x, a) M^*(x'|x, a) \left( \log \frac{M^*(x'|x, a)}{M(x'|x, a)} - \log M^*(x'|x, a) \right) \mathrm{d}x'
\end{aligned}
$$
(11)

$$
\begin{aligned}
=& \frac{1}{\alpha_M \rho_{M^*}^\kappa(x, a)} \int_{\mathcal{X}} \rho_{M^*}^\kappa(x, a, x') \left( \log \frac{\rho_{M^*}^\kappa(x, a, x')}{\rho_M^\kappa(x, a, x')} + \log \frac{\rho_M^\kappa(x, a)}{\rho_{M^*}^\kappa(x, a)} - \log M^*(x'|x, a) \right) \mathrm{d}x' \\
=& \frac{1}{\alpha_M \rho_{M^*}^\kappa(x, a)} \left( \int_{\mathcal{X}} \rho_{M^*}^\kappa(x, a, x') \log \frac{\rho_{M^*}^\kappa(x, a, x')}{\rho_M^\kappa(x, a, x')} \mathrm{d}x' - \right. \\
& \left. \rho_{M^*}^\kappa(x, a) \log \frac{\rho_{M^*}^\kappa(x, a)}{\rho_M^\kappa(x, a)} \underbrace{\int_{\mathcal{X}} M^*(x'|x, a) \mathrm{d}x'}_{=1} - \rho_{M^*}^\kappa(x, a) \int_{\mathcal{X}} M^*(x'|x, a) \log M^*(x'|x, a) \mathrm{d}x' \right) \\
=& \frac{1}{\alpha_0(x, a)} \left( \int_{\mathcal{X}} \rho_{M^*}^\kappa(x, a, x') \log \frac{\rho_{M^*}^\kappa(x, a, x')}{\rho_M^\kappa(x, a, x')} \mathrm{d}x' - \rho_{M^*}^\kappa(x, a) \log \frac{\rho_{M^*}^\kappa(x, a)}{\rho_M^\kappa(x, a)} + \rho_{M^*}^\kappa(x, a) H_{M^*}(x, a) \right)
\end{aligned}
$$
(12)

where $\alpha_0(x, a) = \alpha_M \rho_{M^*}^\kappa(x, a)$.

**Definition A.5** ($f$-divergence). Given two distributions $P$ and $Q$, two absolutely continuous density functions $p$ and $q$ with respect to a base measure $\mathrm{d}x$ defined on the domain $\mathcal{X}$, we define the $f$-divergence [37],

$$
D_f(P\|Q) = \int_{\mathcal{X}} q(x) f\left( \frac{p(x)}{q(x)} \right) \mathrm{d}x,
$$
(13)

where the generator function $f : \mathbb{R}_+ \to \mathbb{R}$ is a convex, lower-semicontinuous function.

We notice that the terms $\rho_{M^*}^\kappa(x, a, x') \log \frac{\rho_{M^*}^\kappa(x, a, x')}{\rho_M^\kappa(x, a, x')}$ and $\rho_{M^*}^\kappa(x, a) \log \frac{\rho_{M^*}^\kappa(x, a)}{\rho_M^\kappa(x, a)}$ are the integrated functions in reverse KL divergence, which is an instance of $f$ function in $f$-divergence (See Reverse-

KL divergence of Tab.1 in [37] for more details). Replacing that form $q \log \frac{q}{p}$ with $qf(\frac{p}{q})$, we obtain a generalized representation of $\bar{\rho}_{M^*}^{\bar{\beta}^*}$:

$$\bar{\rho}_{M^*}^{\bar{\beta}^*} := \frac{1}{\alpha_0(x,a)} \left( \int_{\mathcal{X}} \rho_{M^*}^{\kappa}(x,a,x') f\left( \frac{\rho_M^{\kappa}(x,a,x')}{\rho_{M^*}^{\kappa}(x,a,x')} \right) \mathrm{d}x' - \rho_{M^*}^{\kappa}(x,a) \left( f\left( \frac{\rho_M^{\kappa}(x,a)}{\rho_{M^*}^{\kappa}(x,a)} \right) - H_{M^*}(x,a) \right) \right),$$
$$(14)$$

### A.3 Proof of Thm. A.2

We first introduce several useful lemmas for the proof.

**Lemma A.6.** *Rearrangement inequality* *The rearrangement inequality states that, for two sequences* $a_1 \geq a_2 \geq \ldots \geq a_n$ *and* $b_1 \geq b_2 \geq \ldots \geq b_n$, *the inequalities*

$$a_1 b_1 + a_2 b_2 + \cdots + a_n b_n \geq a_1 b_{\pi(1)} + a_2 b_{\pi(2)} + \cdots + a_n b_{\pi(n)} \geq a_1 b_n + a_2 b_{n-1} + \cdots + a_n b_1$$

*hold, where* $\pi(1), \pi(2), \ldots, \pi(n)$ *is any permutation of* $1, 2, \ldots, n$.

**Lemma A.7.** *For two sequences* $a_1 \geq a_2 \geq \ldots \geq a_n$ *and* $b_1 \geq b_2 \geq \ldots \geq b_n$, *the inequalities*

$$\sum_{i=1}^{n} \frac{1}{n} a_i b_i \geq \sum_{i=1}^{n} \frac{1}{n} a_i \sum \frac{1}{n} b_i$$

*hold.*

*Proof.* By rearrangement inequality, we have

$$\sum_{i=1}^{n} a_i b_i \geq a_1 b_1 + a_2 b_2 + \cdots + a_n b_n$$

$$\sum_{i=1}^{n} a_i b_i \geq a_1 b_2 + a_2 b_3 + \cdots + a_n b_1$$

$$\sum_{i=1}^{n} a_i b_i \geq a_1 b_3 + a_2 b_4 + \cdots + a_n b_2$$

$$\vdots$$

$$\sum_{i=1}^{n} a_i b_i \geq a_1 b_n + a_2 b_1 + \cdots + a_n b_{n-1}$$

Then we have

$$n \sum_{i=1}^{n} a_i b_i \geq \sum_{i=1}^{n} a_i \sum_{i=1}^{n} b_i$$

$$\sum_{i=1}^{n} \frac{1}{n} a_i b_i \geq \sum_{i=1}^{n} \frac{1}{n} a_i \sum \frac{1}{n} b_i$$

$\square$

Now we extend Lemma A.7 into the continuous integral scenario:

**Lemma A.8.** *Given* $\mathcal{X} \subset \mathbb{R}$, *for two functions* $f : \mathcal{X} \to \mathbb{R}$ *and* $g : \mathcal{X} \to \mathbb{R}$ *that* $f(x) \geq f(y)$ *if and only if* $g(x) \geq g(y)$, $\forall x, y \in \mathcal{X}$, *the inequality*

$$\int_{\mathcal{X}} p(x) f(x) g(x) \mathrm{d}x \geq \int_{\mathcal{X}} p(x) f(x) \mathrm{d}x \int_{\mathcal{X}} p(x) g(x) \mathrm{d}x$$

*holds, where* $p : \mathcal{X} \to \mathbb{R}$ *and* $p(x) > 0, \forall x \in \mathcal{X}$ *and* $\int_{\mathcal{X}} p(x) \mathrm{d}x = 1$.

*Proof.* Since $(f(x) - f(y))(g(x) - g(y)) \geq 0, \forall x, y \in \mathcal{X}$, we have

$$\int_{x\in\mathcal{X}}\int_{y\in\mathcal{X}}p(x)p(y)(f(x)-f(y))(g(x)-g(y))\mathrm{d}y\mathrm{d}x\geq 0$$

$$\int_{x\in\mathcal{X}}\int_{y\in\mathcal{X}}p(x)p(y)f(x)g(x)+p(x)p(y)f(y)g(y)-p(x)p(y)f(x)g(y)-p(x)p(y)f(y)g(x)\mathrm{d}y\mathrm{d}x\geq 0$$

$$\int_{x\in\mathcal{X}}\int_{y\in\mathcal{X}}p(x)p(y)f(x)g(x)+p(x)p(y)f(y)g(y)\mathrm{d}y\mathrm{d}x\geq\int_{x\in\mathcal{X}}\int_{y\in\mathcal{X}}p(x)p(y)f(x)g(y)+p(x)p(y)f(y)g(x)\mathrm{d}y\mathrm{d}x$$

$$\int_{x\in\mathcal{X}}\left(\int_{y\in\mathcal{X}}p(x)p(y)f(x)g(x)\mathrm{d}y+\int_{y\in\mathcal{X}}p(x)p(y)f(y)g(y)\mathrm{d}y\right)\mathrm{d}x\geq\int_{x\in\mathcal{X}}\int_{y\in\mathcal{X}}p(x)p(y)f(x)g(y)+p(x)p(y)f(y)g(x)\mathrm{d}y\mathrm{d}x$$

$$\int_{x\in\mathcal{X}}\left(p(x)f(x)g(x)+\int_{y\in\mathcal{X}}p(x)p(y)f(y)g(y)\mathrm{d}y\right)\mathrm{d}x\geq\int_{x\in\mathcal{X}}\int_{y\in\mathcal{X}}p(x)p(y)f(x)g(y)+p(x)p(y)f(y)g(x)\mathrm{d}y\mathrm{d}x$$

$$\int_{x\in\mathcal{X}}p(x)f(x)g(x)\mathrm{d}x+\int_{x\in\mathcal{X}}\int_{y\in\mathcal{X}}p(x)p(y)f(y)g(y)\mathrm{d}y\mathrm{d}x\geq\int_{x\in\mathcal{X}}\int_{y\in\mathcal{X}}p(x)p(y)f(x)g(y)+p(x)p(y)f(y)g(x)\mathrm{d}y\mathrm{d}x$$

$$\int_{x\in\mathcal{X}}p(x)f(x)g(x)\mathrm{d}x+\int_{y\in\mathcal{X}}p(y)f(y)g(y)\mathrm{d}y\geq\int_{x\in\mathcal{X}}\int_{y\in\mathcal{X}}p(x)p(y)f(x)g(y)+p(x)p(y)f(y)g(x)\mathrm{d}y\mathrm{d}x$$

$$2\int_{x\in\mathcal{X}}p(x)f(x)g(x)\mathrm{d}x\geq 2\int_{y\in\mathcal{X}}\int_{x\in\mathcal{X}}p(x)p(y)f(x)g(y)\mathrm{d}y\mathrm{d}x$$

$$2\int_{x\in\mathcal{X}}p(x)f(x)g(x)\mathrm{d}x\geq 2\int_{x\in\mathcal{X}}p(x)f(x)\mathrm{d}x\int_{x\in\mathcal{X}}p(x)g(x)\mathrm{d}x$$

$$\int_{x\in\mathcal{X}}p(x)f(x)g(x)\mathrm{d}x\geq\int_{x\in\mathcal{X}}p(x)f(x)\mathrm{d}x\int_{x\in\mathcal{X}}p(x)g(x)\mathrm{d}x$$

$\square$

**Corollary A.9.** *Let* $g(\frac{p(x)}{q(x)})=-\log\frac{p(x)}{q(x)}$ *where* $p(x)>0,\forall x\in\mathcal{X}$ *and* $q(x)>0,\forall x\in\mathcal{X}$, *for* $\upsilon>0$, *the inequality*

$$\int_{\mathcal{X}}q(x)f(\upsilon\frac{p(x)}{q(x)})g(\frac{p(x)}{q(x)})\mathrm{d}x\geq\int_{\mathcal{X}}q(x)f(\upsilon\frac{p(x)}{q(x)})\mathrm{d}x\int_{\mathcal{X}}q(x)g(\frac{p(x)}{q(x)})\mathrm{d}x,$$

*holds if* $f'(x)\leq 0,\forall x\in\mathcal{X}$. *It is not always satisfied for* $f$ *functions of* $f$-*divergence. We list a comparison of* $f$ *on that condition in Tab. 3.*

*Proof.* $g'(x)=-\log x=-\frac{1}{x}<0,\forall x\in\mathcal{X}$. Suppose $f'(x)\leq 0,\forall x\in\mathcal{X}$, we have $f(x)\geq f(y)$ if and only if $g(x)\geq g(y)$, $\forall x,y\in\mathcal{X}$ holds. Thus $f(\upsilon\frac{p(x)}{q(x)})\geq f(\upsilon\frac{p(y)}{q(y)})$ if and only if $g(\frac{p(x)}{q(x)})\geq g(\frac{p(y)}{q(y)})$, $\forall x,y\in\mathcal{X}$ holds for all $\upsilon>0$. By defining $F(x)=f(\upsilon\frac{p(x)}{q(x)}))$ and $G(x)=g(\frac{p(x)}{q(x)})$ and using Lemma A.8, we have:

$$\int_{\mathcal{X}}q(x)F(x)G(x)\mathrm{d}x\geq\int_{\mathcal{X}}q(x)F(x)\mathrm{d}x\int_{\mathcal{X}}q(x)G(x)\mathrm{d}x.$$

Then we know

$$\int_{\mathcal{X}}q(x)f(\upsilon\frac{p(x)}{q(x)})g(\frac{p(x)}{q(x)})\mathrm{d}x\geq\int_{\mathcal{X}}q(x)f(\upsilon\frac{p(x)}{q(x)})\mathrm{d}x\int_{\mathcal{X}}q(x)g(\frac{p(x)}{q(x)})\mathrm{d}x$$

holds. $\square$

Now, we prove Thm. A.2. For better readability, we first rewrite Thm. A.2 as follows:

**Theorem A.10.** *Let* $\bar{\rho}_{M^*}^{\bar{\beta}^*}$ *as the data distribution of the best-response policy* $\bar{\beta}^*$ *in Eq. (4) under model* $M_\theta$ *parameterized by* $\theta$, *then we can find the optimal* $\theta^*$ *of* $\min_\theta\max_{\beta\in\Pi}\bar{L}(\rho_{M^*}^\beta,M_\theta)$ *(Eq. (4)) via iteratively optimizing the objective* $\theta_{t+1}=\min_\theta\bar{L}(\bar{\rho}_{M^*}^{\bar{\beta}^*},M_\theta)$, *where* $\bar{\rho}_{M^*}^{\bar{\beta}^*}$ *is approximated via the last-iteration model* $M_{\theta_t}$. *Based on Corollary A.9, we have an upper bound objective for*

Table 3: Properties of $f'(x) \leq 0, \forall x \in \mathcal{X}$ for $f$-divergences.

| Name | Generator function $f(x)$ | If $f'(x) \leq 0, \forall x \in \mathcal{X}$ |
|---|---|---|
| Kullback-Leibler | $x \log x$ | False |
| Reverse KL | $-\log x$ | True |
| Pearson $\chi^2$ | $(x-1)^2$ | False |
| Squared Hellinger | $(\sqrt{x}-1)^2$ | False |
| Jensen-Shannon | $-(x+1)\log\frac{1+x}{2} + x \log x$ | False |
| GAN | $x \log x - (x+1)\log(x+1)$ | True |

$\min_\theta \bar{L}(\bar{\rho}_{M^*}^{\bar{\beta}^*}, M_\theta)$ *and derive the following objective*

$$\theta_{t+1} = \arg\max_\theta \mathbb{E}_{\rho_{M^*}^\kappa} \left[ \frac{1}{\alpha_0(x,a)} \log M_\theta(x'|x,a) \underbrace{\left( f\left( \frac{\rho_{M_{\theta_t}}^\kappa(x,a,x')}{\rho_{M^*}^\kappa(x,a,x')} \right) - f\left( \frac{\rho_{M_{\theta_t}}^\kappa(x,a)}{\rho_{M^*}^\kappa(x,a)} \right) + H_{M^*}(x,a) \right)}_{W(x,a,x')} \right],$$

*where $\alpha_0(x,a) = \alpha_{M_{\theta_t}} \rho_{M^*}^\kappa(x,a)$, $\mathbb{E}_{\rho_{M^*}^\kappa}[\cdot]$ denotes $\mathbb{E}_{x,a,x' \sim \rho_{M^*}^\kappa}[\cdot]$, $f$ is the generator function in $f$-divergence which satisfies $f'(x) \leq 0, \forall x \in \mathcal{X}$, and $\theta$ is the parameters of $M$. $M_{\theta_t}$ denotes a probability function with the same parameters as the learned model (i.e., $\bar{\theta} = \theta$) but the parameter is fixed and only used for sampling.*

*Proof.* Let $\bar{\rho}_{M^*}^{\bar{\beta}^*}$ as the data distribution of the best-response policy $\bar{\beta}^*$ in Eq. (4) under model $M_\theta$ parameterized by $\theta$, then we can find the optimal $\theta_{t+1}$ of $\min_\theta \max_{\beta \in \Pi} \bar{L}(\rho_{M^*}^\beta, M_\theta)$ (Eq. (4)) via iteratively optimizing the objective $\theta_{t+1} = \min_\theta \bar{L}(\bar{\rho}_{M^*}^{\bar{\beta}^*}, M_\theta)$, where $\bar{\rho}_{M^*}^{\bar{\beta}^*}$ is approximated via the last-iteration model $M_{\theta_t}$:

$$\theta_{t+1} = \min_\theta \int_{\mathcal{X},\mathcal{A}} \bar{\rho}_{M^*}^{\bar{\beta}^*}(x,a) \int_{\mathcal{X}} M^*(x'|x,a) \left(-\log M_\theta(x'|x,a)\right) \mathrm{d}x' \mathrm{d}a \mathrm{d}x \tag{15}$$

$$= \min_\theta \int_{\mathcal{X},\mathcal{A}} \frac{1}{\alpha_0(x,a)} \left( \int_{\mathcal{X}} \rho_{M^*}^\kappa(x,a,x') f\left( \frac{\rho_{M_{\theta_t}}^\kappa(x,a,x')}{\rho_{M^*}^\kappa(x,a,x')} \right) \mathrm{d}x' \int_{\mathcal{X}} M^*(x'|x,a)(-\log M_\theta(x'|x,a))\mathrm{d}x' \right.$$
$$\left. - \rho_{M^*}^\kappa(x,a) \left( f\left( \frac{\rho_{M_{\theta_t}}^\kappa(x,a)}{\rho_{M^*}^\kappa(x,a)} \right) - H_{M^*}(x,a) \right) \int_{\mathcal{X}} M^*(x'|x,a)(-\log M_\theta(x'|x,a))\mathrm{d}x' \right) \mathrm{d}a \mathrm{d}x$$

$$= \min_\theta \int_{\mathcal{X},\mathcal{A}} \frac{1}{\alpha_0(x,a)} \left( \int_{\mathcal{X}} \rho_{M^*}^\kappa(x,a,x') f\left( \frac{\rho_{M_{\theta_t}}^\kappa(x,a,x')}{\rho_{M^*}^\kappa(x,a,x')} \right) \mathrm{d}x' \left( \int_{\mathcal{X}} M^*(x'|x,a)(-\log \frac{M_\theta(x'|x,a)}{M^*(x'|x,a)})\mathrm{d}x' + H_{M^*}(x,a) \right) \right.$$
$$\left. - \rho_{M^*}^\kappa(x,a) \left( f\left( \frac{\rho_{M_{\theta_t}}^\kappa(x,a)}{\rho_{M^*}^\kappa(x,a)} \right) - H_{M^*}(x,a) \right) \int_{\mathcal{X}} M^*(x'|x,a)(-\log M_\theta(x'|x,a))\mathrm{d}x' \right) \mathrm{d}a \mathrm{d}x$$

$$\leq \min_\theta \int_{\mathcal{X},\mathcal{A}} \frac{1}{\alpha_0(x,a)} \left( \underbrace{\rho_{M^*}^\kappa(x,a) \int_{\mathcal{X}} M^*(x'|x,a) f\left( \frac{\rho_{M_{\theta_t}}^\kappa(x,a,x')}{\rho_{M^*}^\kappa(x,a,x')} \right) (-\log \frac{M_\theta(x'|x,a)}{M^*(x'|x,a)})\mathrm{d}x'}_{\text{based on Corollary } A.9} \right.$$
$$\left. - \rho_{M^*}^\kappa(x,a) \left( f\left( \frac{\rho_{M_{\theta_t}}^\kappa(x,a)}{\rho_{M^*}^\kappa(x,a)} \right) - H_{M^*}(x,a) \right) \int_{\mathcal{X}} M^*(x'|x,a)(-\log M_\theta(x'|x,a))\mathrm{d}x' \right) \mathrm{d}a \mathrm{d}x$$

$$= \min_\theta \int_{\mathcal{X},\mathcal{A}} \frac{1}{\alpha_0(x,a)} \left( \rho_{M^*}^\kappa(x,a) \int_{\mathcal{X}} \left( M^*(x'|x,a) f\left( \frac{\rho_{M_{\theta_t}}^\kappa(x,a,x')}{\rho_{M^*}^\kappa(x,a,x')} \right) (-\log M_\theta(x'|x,a)) \right) \mathrm{d}x' \right.$$
$$\left. - \rho_{M^*}^\kappa(x,a) \left( f\left( \frac{\rho_{M_{\theta_t}}^\kappa(x,a)}{\rho_{M^*}^\kappa(x,a)} \right) - H_{M^*}(x,a) \right) \int_{\mathcal{X}} M^*(x'|x,a)(-\log M_\theta(x'|x,a))\mathrm{d}x' \right) \mathrm{d}a \mathrm{d}x$$

$$= \max_\theta \int_{\mathcal{X},\mathcal{A},\mathcal{X}} \frac{1}{\alpha_0(x,a)} \rho_{M^*}^\kappa(x,a,x') \log M_\theta(x'|x,a) \left( f\left( \frac{\rho_{M_{\theta_t}}^\kappa(x,a,x')}{\rho_{M^*}^\kappa(x,a,x')} \right) - f\left( \frac{\rho_{M_{\theta_t}}^\kappa(x,a)}{\rho_{M^*}^\kappa(x,a)} \right) + H_{M^*}(x,a) \right) \mathrm{d}x' \mathrm{d}a \mathrm{d}x,$$

$$\tag{16}$$

where $M_{\theta_t}$ is introduced to approximate the term $\bar{\rho}_{M^*}^{\bar{\beta}^*}$ and fixed when optimizing $\theta$. In Eq. (15), $\|\rho_{M^*}^{\beta}(\cdot,\cdot)\|_2^2$ for Eq. (7) is eliminated as it does not contribute to the gradient of $\theta$. Assume $f'(x) \leq 0, \forall x \in \mathcal{X}$, let $\upsilon(x,a) := \frac{\rho_{M_{\theta_t}}^{\kappa}(x,a)}{\rho_{M^*}^{\kappa}(x,a)} > 0$, $p(x'|x,a) = M_\theta(x'|x,a)$, and $q(x'|x,a) = M^*(x'|x,a)$, the first inequality can be derived by adopting Corollary A.9 and eliminating the first $H_{M^*}$ since it does not contribute to the gradient of $\theta$.

$\square$

## A.4 Proof of the Tractable Solution

Now we are ready to prove the tractable solution:

*Proof.* The core challenge is that the term $f(\frac{\rho_{M_{\theta_t}}^{\kappa}(x,a,x')}{\rho_{M^*}^{\kappa}(x,a,x')}) - f(\frac{\rho_{M_{\theta_t}}^{\kappa}(x,a)}{\rho_{M^*}^{\kappa}(x,a)})$ is still intractable. In the following, we give a tractable solution to Thm. A.2. First, we resort to the first-order approximation. Given some $u \in (1-\xi, 1+\xi), \xi > 0$, we have

$$f(u) \approx f(1) + f'(u)(u-1), \tag{17}$$

where $f'$ is the first-order derivative of $f$. By Taylor's formula and the fact that $f'(u)$ of the generator function $f$ is bounded in $(1-\xi, 1+\xi)$, the approximation error is no more than $\mathcal{O}(\xi^2)$. Substituting $u$ with $\frac{p(x)}{q(x)}$ in Eq. (17), the pattern $f(\frac{p(x)}{q(x)})$ in Eq. (16) can be converted to $\frac{p(x)}{q(x)}f'(\frac{p(x)}{q(x)}) - f'(\frac{p(x)}{q(x)}) + f(1)$, then we have:

$$
\begin{aligned}
\theta_{t+1} = \arg\max_\theta \frac{1}{\alpha_0(x,a)} \int_{\mathcal{X},\mathcal{A}} & \left( \rho_{M^*}^{\kappa}(x,a) \int_{\mathcal{X}} M^*(x'|x,a) f\left( \frac{\rho_{M_{\theta_t}}^{\kappa}(x,a,x')}{\rho_{M^*}^{\kappa}(x,a,x')} \right) \log M_\theta(x'|x,a) \mathrm{d}x' - \right. \\
& \rho_{M^*}^{\kappa}(x,a) f\left( \frac{\rho_{M_{\theta_t}}^{\kappa}(x,a)}{\rho_{M^*}^{\kappa}(x,a)} \right) \int_{\mathcal{X}} M^*(x'|x,a) \log M_\theta(x'|x,a) \mathrm{d}x' + \\
& \left. \rho_{M^*}^{\kappa}(x,a) H_{M^*}(x,a) \int_{\mathcal{X}} M^*(x'|x,a) \log M_\theta(x'|x,a) \mathrm{d}x' \right) \mathrm{d}a \mathrm{d}x \\
\approx \arg\max_\theta \int_{\mathcal{X},\mathcal{A}} & \left( \rho_{M_{\theta_t}}^{\kappa}(x,a) \int_{\mathcal{X}} M_{\theta_t}(x'|x,a) f'\left( \frac{\rho_{M_{\theta_t}}^{\kappa}(x,a,x')}{\rho_{M^*}^{\kappa}(x,a,x')} \right) \log M_\theta(x'|x,a) \mathrm{d}x' - \right. \\
& \rho_{M^*}^{\kappa}(x,a) \int_{\mathcal{X}} M^*(x'|x,a) \left( f'\left( \frac{\rho_{M_{\theta_t}}^{\kappa}(x,a,x')}{\rho_{M^*}^{\kappa}(x,a,x')} \right) - f(1) \right) \log M_\theta(x'|x,a) \mathrm{d}x' - \\
& \rho_{M_{\theta_t}}^{\kappa}(x,a) f'\left( \frac{\rho_{M_{\theta_t}}^{\kappa}(x,a)}{\rho_{M^*}^{\kappa}(x,a)} \right) \int_{\mathcal{X}} M^*(x'|x,a) \log M_\theta(x'|x,a) \mathrm{d}x' + \\
& \rho_{M^*}^{\kappa}(x,a) \left( f'\left( \frac{\rho_{M_{\theta_t}}^{\kappa}(x,a)}{\rho_{M^*}^{\kappa}(x,a)} \right) - f(1) \right) \int_{\mathcal{X}} M^*(x'|x,a) \log M_\theta(x'|x,a) \mathrm{d}x' + \\
& \left. \rho_{M^*}^{\kappa}(x,a) H_{M^*}(x,a) \int_{\mathcal{X}} M^*(x'|x,a) \log M_\theta(x'|x,a) \mathrm{d}x' \right) \mathrm{d}a \mathrm{d}x \\
= \arg\max_\theta \int_{\mathcal{X},\mathcal{A},\mathcal{X}} & \frac{1}{\alpha_0(x,a)} \rho_{M_{\theta_t}}^{\kappa}(x,a,x) \left( f'\left( \frac{\rho_{M_{\theta_t}}^{\kappa}(x,a,x')}{\rho_{M^*}^{\kappa}(x,a,x')} \right) - f'\left( \frac{\rho_{M_{\theta_t}}^{\kappa}(x,a)}{\rho_{M^*}^{\kappa}(x,a)} \right) \right) \log M_\theta(x'|x,a) \mathrm{d}x' \mathrm{d}a \mathrm{d}x + \\
& \int_{\mathcal{X},\mathcal{A},\mathcal{X}} \frac{1}{\alpha_0(x,a)} \rho_{M^*}^{\kappa}(x,a,x') \left( f'\left( \frac{\rho_{M_{\theta_t}}^{\kappa}(x,a)}{\rho_{M^*}^{\kappa}(x,a)} \right) - f'\left( \frac{\rho_{M_{\theta_t}}^{\kappa}(x,a,x')}{\rho_{M^*}^{\kappa}(x,a,x')} \right) + H_{M^*}(x,a) \right) \log M_\theta(x'|x,a) \mathrm{d}x' \mathrm{d}a \mathrm{d}x.
\end{aligned}
$$

Note that the part $\rho_{M^*}^{\kappa}(x,a)$ in $\rho_{M^*}^{\kappa}(x,a,x')$ can be canceled because of $\alpha_0(x,a) = \alpha_{M_{\theta_t}} \rho_{M^*}^{\kappa}(x,a)$, but we choose to keep it and ignore $\alpha_0(x,a)$. The benefit is that we can estimate $\rho_{M^*}^{\kappa}(x,a,x')$ from an empirical data distribution through data collected by $\kappa$ in $M^*$ directly, rather than from a uniform distribution which is harder to be generated. Although keeping $\rho_{M^*}^{\kappa}(x,a)$ incurs extra bias in theory, the results in our experiments show that it has not made significant negative effects in practice. We

leave this part of modeling in future work. In particular, by ignoring $\alpha_0(x,a)$, we have:

$$\theta_{t+1} = \arg\max_\theta \int_{\mathcal{X},\mathcal{A},\mathcal{X}} \rho^\kappa_{M_{\theta_t}}(x,a,x) \left( f'\left(\frac{\rho^\kappa_{M_{\theta_t}}(x,a,x')}{\rho^\kappa_{M^*}(x,a,x')}\right) - f'\left(\frac{\rho^\kappa_{M_{\theta_t}}(x,a)}{\rho^\kappa_{M^*}(x,a)}\right) \right) \log M_\theta(x'|x,a)\mathrm{d}x'\mathrm{d}a\mathrm{d}x+ \tag{18}$$

$$\int_{\mathcal{X},\mathcal{A},\mathcal{X}} \rho^\kappa_{M^*}(x,a,x') \left( f'\left(\frac{\rho^\kappa_{M_{\theta_t}}(x,a)}{\rho^\kappa_{M^*}(x,a)}\right) - f'\left(\frac{\rho^\kappa_{M_{\theta_t}}(x,a,x')}{\rho^\kappa_{M^*}(x,a,x')}\right) + H_{M^*}(x,a)\right) \log M_\theta(x'|x,a)\mathrm{d}x'\mathrm{d}a\mathrm{d}x. \tag{19}$$

We can estimate $f'\left(\frac{\rho^\kappa_{M_{\theta_t}}(x,a)}{\rho^\kappa_{M^*}(x,a)}\right)$ and $f'\left(\frac{\rho^\kappa_{M_{\theta_t}}(x,a,x')}{\rho^\kappa_{M^*}(x,a,x')}\right)$ through Lemma A.11.

**Lemma A.11** ($f'(\frac{p}{q})$ estimation [36]). *Given a function $T_\varphi : \mathcal{X} \to \mathbb{R}$ parameterized by $\varphi \in \Phi$, if $f$ is convex and lower semi-continuous, by finding the maximum point of $\varphi$ in the following objective:*

$$\varphi^* = \arg\max_\varphi \mathbb{E}_{x\sim p(x)}\left[T_\varphi(x)\right] - \mathbb{E}_{x\sim q(x)}\left[f^*(T_\varphi(x))\right],$$

*we have $f'(\frac{p(x)}{q(x)}) = T_{\varphi^*}(x)$. $f^*$ is Fenchel conjugate of $f$ [23].*

In particular,

$$\varphi_0^* = \arg\max_{\varphi_0} \mathbb{E}_{x,a,x'\sim\rho^\kappa_{M^*}}\left[T_{\varphi_0}(x,a,x')\right] - \mathbb{E}_{x,a,x'\sim\rho^\kappa_{M_{\theta_t}}}\left[f^*(T_{\varphi_0}(x,a,x'))\right]$$

$$\varphi_1^* = \arg\max_{\varphi_1} \mathbb{E}_{x,a\sim\rho^\kappa_{M^*}}\left[T_{\varphi_1}(x,a)\right] - \mathbb{E}_{x,a\sim\rho^\kappa_{M_{\theta_t}}}\left[f^*(T_{\varphi_1}(x,a))\right],$$

then we have $f'\left(\frac{\rho^\kappa_{M_{\theta_t}}(x,a,x')}{\rho^\kappa_{M^*}(x,a,x')}\right) \approx T_{\varphi_0^*}(x,a,x')$ and $f'\left(\frac{\rho^\kappa_{M_{\theta_t}}(x,a)}{\rho^\kappa_{M^*}(x,a)}\right) \approx T_{\varphi_1^*}(x,a)$. Given $\varphi_0^*$ and $\varphi_1^*$, let $A_{\varphi_0^*,\varphi_1^*}(x,a,x') = T_{\varphi_0^*}(x,a,x') - T_{\varphi_1^*}(x,a)$, then we can optimize $\theta$ via:

$$\theta_{t+1} = \arg\max_\theta \int_{\mathcal{X},\mathcal{A},\mathcal{X}} \rho^\kappa_{M_{\theta_t}}(x,a,x) \left(T_{\varphi_0^*}(x,a,x') - T_{\varphi_1^*}(x,a)\right) \log M_\theta(x'|x,a)\mathrm{d}x'\mathrm{d}a\mathrm{d}x+$$

$$\int_{\mathcal{X},\mathcal{A},\mathcal{X}} \rho^\kappa_{M^*}(x,a,x') \left(T_{\varphi_1^*}(x,a) - T_{\varphi_0^*}(x,a,x') + H_{M^*}(x,a)\right) \log M_\theta(x'|x,a)\mathrm{d}x'\mathrm{d}a\mathrm{d}x$$

$$= \arg\max_\theta \int_{\mathcal{X},\mathcal{A},\mathcal{X}} \rho^\kappa_{M_{\theta_t}}(x,a,x) A_{\varphi_0^*,\varphi_1^*}(x,a,x') \log M_\theta(x'|x,a)\mathrm{d}x'\mathrm{d}a\mathrm{d}x+$$

$$\int_{\mathcal{X},\mathcal{A},\mathcal{X}} \rho^\kappa_{M^*}(x,a,x')(-A_{\varphi_0^*,\varphi_1^*}(x,a,x') + H_{M^*}(x,a)) \log M_\theta(x'|x,a)\mathrm{d}x'\mathrm{d}a\mathrm{d}x.$$

Based on the specific $f$-divergence, we can represent $T$ and $f^*(T)$ with a discriminator $D_\varphi$. It can be verified that $f(u) = u\log u - (u+1)\log(u+1)$, $T_\varphi(u) = \log D_\varphi(u)$, and $f^*(T_\varphi(u)) = -\log(1 - D_\varphi(u))$ proposed in [37] satisfies the condition $f'(x) \le 0, \forall x \in \mathcal{X}$ (see Tab. 3). We select the former in the implementation and convert the tractable solution to:

$$\theta_{t+1} = \arg\max_\theta \mathbb{E}_{\rho^\kappa_{M_{\theta_t}}}\left[A_{\varphi_0^*,\varphi_1^*}(x,a,x')\log M_\theta(x'|x,a)\right] + \mathbb{E}_{\rho^\kappa_{M^*}}\left[(H_{M^*}(x,a) - A_{\varphi_0^*,\varphi_1^*}(x,a,x'))\log M_\theta(x'|x,a)\right]$$

$$s.t. \quad \varphi_0^* = \arg\max_{\varphi_0} \mathbb{E}_{\rho^\kappa_{M^*}}\left[\log D_{\varphi_0}(x,a,x')\right] + \mathbb{E}_{\rho^\kappa_{M_{\theta_t}}}\left[\log(1 - D_{\varphi_0}(x,a,x'))\right]$$

$$\varphi_1^* = \arg\max_{\varphi_1} \mathbb{E}_{\rho^\kappa_{M^*}}\left[\log D_{\varphi_1}(x,a)\right] + \mathbb{E}_{\rho^\kappa_{M_{\theta_t}}}\left[\log(1 - D_{\varphi_1}(x,a))\right], \tag{20}$$

where $A_{\varphi_0^*,\varphi_1^*}(x,a,x') = \log D_{\varphi_0^*}(x,a,x') - \log D_{\varphi_1^*}(x,a)$, $\mathbb{E}_{\rho^\kappa_{M_{\theta_t}}}[\cdot]$ is a simplification of $\mathbb{E}_{x,a,x'\sim\rho^\kappa_{M_{\theta_t}}}[\cdot]$.

$\square$

**Remark A.12.** In practice, we need to use the real-world data to construct the distribution $\rho^\kappa_{M^*}$ and the generative data to construct $\rho^\kappa_{M_{\theta_t}}$. In the offline model-learning setting, we only have a real-world dataset $\mathcal{D}_{\text{real}}$ collected by the behavior policy $\mu$. We can learn a policy $\hat{\mu} \approx \mu$ via imitation learning based on $\mathcal{D}_{\text{real}}$ [40, 25] and let $\hat{\mu}$ be the policy $\kappa$. Then we can regard $\mathcal{D}_{\text{real}}$ as the empirical

data distribution of $\rho_{M^*}^\kappa$ and the trajectories collected by $\hat\mu$ in the model $M_{\theta_t}$ as the empirical data distribution of $\rho_{M_{\theta_t}}^\kappa$. Based on the above specializations, we have:

$$\theta_{t+1} = \max_\theta \ \Big( \mathbb{E}_{\rho_{M_{\theta_t}}^{\hat\mu}} \left[ A_{\varphi_0^*,\varphi_1^*}(x,a,x') \log M_\theta(x'|x,a) \right] + \mathbb{E}_{\rho_{M^*}^\mu} \left[ (H_{M^*}(x,a) - A_{\varphi_0^*,\varphi_1^*}(x,a,x')) \log M_\theta(x'|x,a) \right] \Big)$$

$$s.t. \quad \varphi_0^* = \arg\max_{\varphi_0} \ \Big( \mathbb{E}_{\rho_{M^*}^\mu} \left[ \log D_{\varphi_0}(x,a,x') \right] + \mathbb{E}_{\rho_{M_{\theta_t}}^{\hat\mu}} \left[ \log(1 - D_{\varphi_0}(x,a,x')) \right] \Big)$$

$$\varphi_1^* = \arg\max_{\varphi_1} \ \Big( \mathbb{E}_{\rho_{M^*}^\mu} \left[ \log D_{\varphi_1}(x,a) \right] + \mathbb{E}_{\rho_{M_{\theta_t}}^{\hat\mu}} \left[ \log(1 - D_{\varphi_1}(x,a)) \right] \Big),$$

*which is Eq. (6) in the main body.*

# B    Discussion and Limitations of the Theoretical Results

We summarize the limitations of current theoretical results and future work as follows:

1. As discussed in Remark A.4, the solution Eq. (10) relies on $\rho_{M^*}^\beta(x,a) \in [0,c], \forall a \in \mathcal{A}, \forall x \in \mathcal{X}$. In some particular $M^*$, it is intractable to derive a $\beta$ that can generate an occupancy specified by $g(x,a)/\alpha_M$. If more knowledge of $M^*$ or $\beta^*$ is provided or some mild assumptions can be made on the properties of $M^*$ or $\beta^*$, we may model $\rho$ in a more sophisticated approach to alleviating the above problem.

2. In the tractable solution derivation, we ignore the term $\alpha_0(x,a) = \alpha_{M_{\theta_t}} \rho_{M^*}^\kappa(x,a)$ (See Eq. (19)). The benefit is that $\rho_{M^*}^\kappa(x,a,x')$ in the tractable solution can be estimated through offline datasets directly. Although the results in our experiments show that it does not produce significant negative effects in these tasks, ignoring $\rho_{M^*}^\kappa(x,a)$ indeed incurs extra bias in theory. In future work, techniques for estimating $\rho_{M^*}^\kappa(x,a)$ [33] can be incorporated to correct the bias. On the other hand, $\alpha_{M_{\theta_t}}$ is also ignored in the process. $\alpha_{M_{\theta_t}}$ can be regarded as a global rescaling term of the final objective Eq. (19). Intuitively, it constructs an adaptive learning rate for Eq. (19), which increases the step size when the model is better fitted and decreases the step size otherwise. It can be considered to further improve the learning process in future work, e.g., cooperating with empirical risk minimization by balancing the weights of the two objectives through $\alpha_{M_{\theta_t}}$.

# C    Societal Impact

This work studies a method toward counterfactual environment model learning. Reconstructing an accurate environment of the real world will promote the wide adoption of decision-making policy optimization methods in real life, enhancing our daily experience. We are aware that decision-making policy in some domains like recommendation systems that interact with customers may have risks of causing price discrimination and misleading customers if inappropriately used. A promising way to reduce the risk is to introduce fairness into policy optimization and rules to constrain the actions (Also see our policy design in Sec. G.1.3). We are involved in and advocating research in such directions. We believe that business organizations would like to embrace fair systems that can ultimately bring long-term financial benefits by providing a better user experience.

# D    AWRM-oracle Pseudocode

We list the pseudocode of AWRM-oracle in Alg. 2.

# E    Implementation

## E.1    Details of the GALILEO Implementation

The approximation of Eq. (17) holds only when $p(x)/q(x)$ is close to 1, which might not be satisfied. To handle the problem, we inject a standard supervised learning loss

$$\arg\max_\theta \mathbb{E}_{\rho_{M^*}^\kappa} \left[ \log M_\theta(x'|x,a) \right] \tag{21}$$

---

**Algorithm 2** AWRM with Oracle Counterfactual Datasets

---

**Input**:
$\Phi$: policy space; $N$: total iterations
**Process**:

1: Generate counterfactual datasets $\{\mathcal{D}_{\pi_\phi}\}$ for all adversarial policies $\pi_\phi, \phi \in \Phi$
2: Initialize an environment model $M_\theta$
3: **for** i = 1:N **do**
4:    Select $\mathcal{D}_{\pi_\phi}$ with worst prediction errors through $M_\theta$ from $\{\mathcal{D}_{\pi_\phi}\}$
5:    Optimize $M_\theta$ with standard supervised learning based on $\mathcal{D}_{\pi_\phi}$
6: **end for**

---

to replace the second term of the above objective when the output probability of $D$ is far away from $0.5$ ($f'(1) = \log 0.5$).

In the offline model-learning setting, we only have a real-world dataset $\mathcal{D}$ collected by the behavior policy $\mu$. We learn a policy $\hat\mu \approx \mu$ via behavior cloning with $\mathcal{D}$ [40, 25] and let $\hat\mu$ be the policy $\kappa$. We regard $\mathcal{D}$ as the empirical data distribution of $\rho_{M^*}^\kappa$ and the trajectories collected by $\hat\mu$ in the model $M_{\theta_t}$ as the empirical data distribution of $\rho_{M_{\theta_t}}^\kappa$. But the assumption $\forall x \in \mathcal{X}, \forall a \in \mathcal{A}, \mu(a|x) > 0$ might not be satisfied. In behavior cloning, we model $\hat\mu$ with a Gaussian distribution and constrain the lower bound of the variance with a small value $\epsilon_\mu > 0$ to keep the assumption holding. Besides, we add small Gaussian noises $\mathbf{u} \sim \mathcal{N}(0, \epsilon_D)$ to the inputs of $D_\varphi$ to handle the mismatch between $\rho_{M^*}^\mu$ and $\rho_{M^*}^{\hat\mu}$ due to $\epsilon_\mu$. In particular, for $\varphi_0$ and $\varphi_1$ learning, we have:

$$\varphi_0^* = \arg\max_{\varphi_0} \mathbb{E}_{\rho_{M^*}^\kappa, \mathbf{u}} \left[\log D_{\varphi_0}(x + u_x, a + u_a, x' + u_{x'})\right] + \mathbb{E}_{\rho_{M_{\theta_t}}^\kappa, \mathbf{u}} \left[\log(1 - D_{\varphi_0}(x + u_x, a + u_a, x' + u_{x'}))\right]$$

$$\varphi_1^* = \arg\max_{\varphi_1} \mathbb{E}_{\rho_{M^*}^\kappa, \mathbf{u}} \left[\log D_{\varphi_1}(x + u_x, a + u_a)\right] + \mathbb{E}_{\rho_{M_{\theta_t}}^\kappa, \mathbf{u}} \left[\log(1 - D_{\varphi_1}(x + u_x, a + u_a))\right],$$

where $\mathbb{E}_{\rho_{M_{\theta_t}}^\kappa, \mathbf{u}}[\cdot]$ is a simplification of $\mathbb{E}_{x,a,x' \sim \rho_{M_{\theta_t}}^\kappa, \mathbf{u} \sim \mathcal{N}(0, \epsilon_D)}[\cdot]$ and $\mathbf{u} = [u_x, u_a, u_{x'}]$.

On the other hand, we notice that the first term in Eq. (20) is similar to the objective of GAIL [25] by regarding $M_\theta$ as the policy to learn and $\kappa$ as the environment to generate data. For better capability in sequential environment model learning, here we introduce some practical tricks inspired by GAIL for model learning [47, 46]: we introduce an MDP for $\kappa$ and $M_\theta$, where the reward is defined by the discriminator $D$, i.e., $r(x, a, x') = \log D(x, a, x')$. $M_\theta$ is learned to maximize the cumulative rewards. With advanced policy gradient methods [44, 45], the objective is converted to $\max_\theta \left[A_{\varphi_0^*, \varphi_1^*}(x, a, x') \log M_\theta(x, a, x')\right]$, where $A = Q_{M_{\theta_t}}^\kappa - V_{M_{\theta_t}}^\kappa$, $Q_{M_{\bar\theta}}^\kappa(x, a, x') = \mathbb{E}\left[\sum_{t=0}^\infty \gamma^t r(x_t, a_t, x_{t+1}) \mid (x_t, a_t, x_{t+1}) = (x, a, x'), \kappa, M_{\theta_t}\right]$, and $V_{M_{\bar\theta}}^\kappa(x, a) = \mathbb{E}_{M_{\bar\theta}}\left[Q_{M_{\bar\theta}}^\kappa(x, a, x')\right]$. $A$ in Eq. (20) can also be constructed similarly. Although it looks unnecessary in theory since the one-step optimal model $M_\theta$ is the global optimal model in this setting, the technique is helpful in practice as it makes $A$ more sensitive to the compounding effect of one-step prediction errors: we would consider the cumulative effects of prediction errors induced by multi-step transitions in environments. In particular, to consider the cumulative effects of prediction errors induced by multi-step of transitions in environments, we overwrite function $A_{\varphi_0^*, \varphi_1^*}$ as $A_{\varphi_0^*, \varphi_1^*} = Q_{M_{\theta_t}}^\kappa - V_{M_{\theta_t}}^\kappa$, where $Q_{M_{\theta_t}}^\kappa(x, a, x') = \mathbb{E}\left[\sum_t^\infty \gamma^t \log D_{\varphi_0^*}(x_t, a_t, x_{t+1})|(x_t, a_t, x_{t+1}) = (x, a, x'), \kappa, M_{\theta_t}\right]$ and $V_{M_{\theta_t}}^\kappa(x, a) = \mathbb{E}\left[\sum_t^\infty \gamma^t \log D_{\varphi_1^*}(x_t, a_t)|(x_t, a_t) = (x, a), \kappa, M_{\theta_t}\right]$. To give an algorithm for single-step environment model learning, we can just set $\gamma$ in $Q$ and $V$ to 0.

---

**Algorithm 3** GALILEO pseudocode

---

**Input**:
$\mathcal{D}_{\text{real}}$: offline dataset sampled from $\rho^{\mu}_{M^*}$ where $\mu$ is the behavior policy;
$N$: total iterations;
**Process**:

1: Approximate a behavior policy $\hat{\mu}$ via behavior cloning
2: Initialize an environment model $M_{\theta_1}$
3: **for** $t = 1 : N$ **do**
4:     Use $\hat{\mu}$ to generate a dataset $\mathcal{D}_{\text{gen}}$ with the model $M_{\theta_t}$
5:     Update the discriminators $D_{\varphi_0}$ and $D_{\varphi_1}$ via Eq. (25) and Eq. (26) respectively, where $\rho^{\hat{\mu}}_{M_{\theta_t}}$
     is estimated by $\mathcal{D}_{\text{gen}}$ and $\rho^{\mu}_{M^*}$ is estimated by $\mathcal{D}_{\text{real}}$
6:     Update $Q$ and $V$ via Eq. (23) and Eq. (24) through $\mathcal{D}_{\text{gen}}$, $D_{\varphi_0}$, and $D_{\varphi_1}$
7:     Update the model $M_{\theta_t}$ via the first term of Eq. (22), which is implemented with a standard
     policy gradient method like TRPO [44] or PPO [45]. Record the policy gradient $g_{\text{pg}}$
8:     **if** $p_0 < \mathbb{E}_{\mathcal{D}_{\text{gen}}}\left[D_{\varphi_0}(x_t, a_t, x_{t+1})\right] < p_1$ **then**
9:         Compute the gradient of $M_{\theta_t}$ via the second term of Eq. (22) and record it as $g_{\text{sl}}$
10:     **else**
11:         Compute the gradient of $M_{\theta_t}$ via Eq. (21) and record it as $g_{\text{sl}}$
12:     **end if**
13:     Rescale $g_{\text{sl}}$ via Eq. (27)
14:     Update the model $M_{\theta_t}$ via the gradient $g_{\text{sl}}$ and obtain $M_{\theta_{t+1}}$
15: **end for**

---

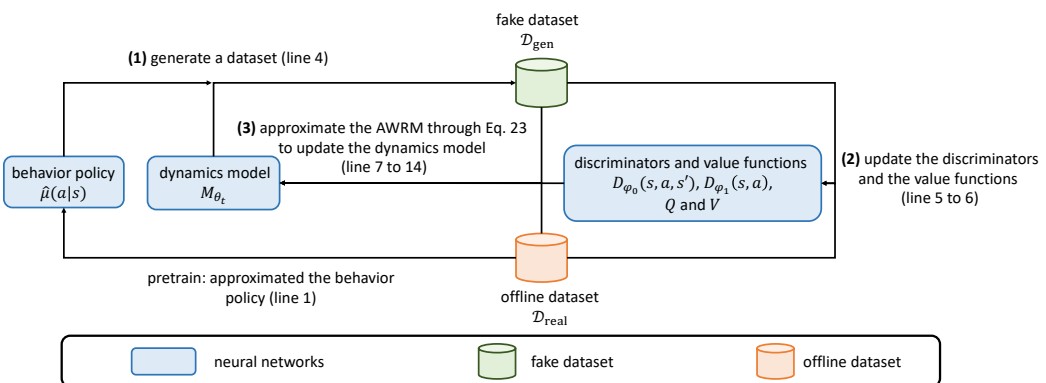

Figure 9: Illustration of the workflow of the GALILEO algorithm.

By adopting the above implementation techniques, we convert the objective into the following formulation

$$\theta_{t+1} = \arg\max_{\theta} \mathbb{E}_{\rho^{\kappa}_{M_{\theta_t}}}\left[A_{\varphi_0^*, \varphi_1^*}(x, a, x')\log M_{\theta}(x'|x, a)\right] + \mathbb{E}_{\rho^{\kappa}_{M^*}}\left[(H_{M^*}(x, a) - A_{\varphi_0^*, \varphi_1^*}(x, a, x'))\log M_{\theta}(x'|x, a)\right]$$

(22)

$$s.t. \qquad Q^{\kappa}_{M_{\theta_t}}(x, a, x') = \mathbb{E}\left[\sum_{t}^{\infty} \gamma^t \log D_{\varphi_0^*}(x_t, a_t, x_{t+1})|(x_t, a_t, x_{t+1}) = (x, a, x'), \kappa, M_{\theta_t}\right]$$

(23)

$$V^{\kappa}_{M_{\theta_t}}(x, a) = \mathbb{E}\left[\sum_{t}^{\infty} \gamma^t \log D_{\varphi_1^*}(x_t, a_t)|(x_t, a_t) = (x, a), \kappa, M_{\theta_t}\right]$$

(24)

$$\varphi_0^* = \arg\max_{\varphi_0} \mathbb{E}_{\rho^{\kappa}_{M^*}, \mathbf{u}}\left[\log D_{\varphi_0}(x + u_x, a + u_a, x' + u_{x'})\right] + \mathbb{E}_{\rho^{\kappa}_{M_{\theta_t}}, \mathbf{u}}\left[\log(1 - D_{\varphi_0}(x + u_x, a + u_a, x' + u_{x'}))\right]$$

(25)

$$\varphi_1^* = \arg\max_{\varphi_1} \mathbb{E}_{\rho^{\kappa}_{M^*}, \mathbf{u}}\left[\log D_{\varphi_1}(x + u_x, a + u_a)\right] + \mathbb{E}_{\rho^{\kappa}_{M_{\theta_t}}, \mathbf{u}}\left[\log(1 - D_{\varphi_1}(x + u_x, a + u_a))\right],$$

(26)

where $A_{\varphi_0^*, \varphi_1^*}(x, a, x') = Q_{M_\theta}^\kappa(x, a, x') - V_{M_\theta}^\kappa(x, a)$. In practice, GALILEO optimizes the first term of Eq. (22) with conservative policy gradient algorithms (e.g., PPO [45] or TRPO [44]) to avoid unreliable gradients for model improvements. Eq. (25) and Eq. (26) are optimized with supervised learning. The second term of Eq. (22) is optimized with supervised learning with a re-weighting term $-A_{\varphi_0^*, \varphi_1^*} + H_{M^*}$. Since $H_{M^*}$ is unknown, we use $H_{M_\theta}$ to estimate it. When the mean output probability of a batch of data is larger than $0.6$ or small than $0.4$, we replace the second term of Eq. (22) with a standard supervised learning in Eq. (21). Besides, unreliable gradients also exist in the process of optimizing the second term of Eq. (22). In our implementation, we use the scale of policy gradients to constrain the gradients of the second term of Eq. (22). In particular, we first compute the $l_2$-norm of the gradient of the first term of Eq. (22) via conservative policy gradient algorithms, named $||g_{\mathrm{pg}}||_2$. Then we compute the $l_2$-norm of the gradient of the second term of Eq. (22), name $||g_{\mathrm{sl}}||_2$. Finally, we rescale the gradients of the second term $g_{\mathrm{sl}}$ by

$$ g_{\mathrm{sl}} \leftarrow g_{\mathrm{sl}} \frac{||g_{\mathrm{pg}}||_2}{\max\{||g_{\mathrm{pg}}||_2, ||g_{\mathrm{sl}}||_2\}}. \tag{27} $$

For each iteration, Eq. (22), Eq. (25), and Eq. (26) are trained with certain steps (See Tab. 6) following the same framework as GAIL. Based on the above techniques, we summarize the pseudocode of GALILEO in Alg. 3, where $p_0$ and $p_1$ are set to $0.4$ and $0.6$ in all of our experiments. The overall architecture is shown in Fig. 9.

### E.2 Connection with Previous Adversarial Algorithms

Standard GAN [19] can be regarded as a partial implementation including the first term of Eq. (22) and Eq. (25) by degrading them into the single-step scenario. In the context of GALILEO, the objective of GAN is

$$ \theta_{t+1} = \arg\max_\theta \mathbb{E}_{\rho_{M_{\theta_t}}^\kappa} \left[ A_{\varphi^*}(x, a, x') \log M_\theta(x'|x, a) \right] $$

$$ s.t. \quad \varphi* = \arg\max_\varphi \mathbb{E}_{\rho_{M^*}^\kappa} \left[ \log D_\varphi(x, a, x') \right] + \mathbb{E}_{\rho_{M_{\theta_t}}^\kappa} \left[ \log(1 - D_\varphi(x, a, x')) \right], $$

where $A_{\varphi^*}(x, a, x') = \log D_{\varphi^*}(x, a, x')$. In the single-step scenario, $\rho_{M_{\theta_t}}^\kappa(x, a, x') = \rho_0(x)\kappa(a|x)M_{\theta_t}(x'|a, x)$. The term $\mathbb{E}_{\rho_{M_{\theta_t}}^\kappa} \left[ A_{\varphi^*}(x, a, x') \log M_\theta(x'|x, a) \right]$ can convert to $\mathbb{E}_{\rho_{M_\theta}^\kappa} \left[ \log D_{\varphi^*}(x, a, x') \right]$ by replacing the gradient of $M_{\theta_t}(x'|x, a)\nabla_\theta \log M_\theta(x'|x, a)$ with $\nabla_\theta M_\theta(x'|x, a)$ [51]. Previous algorithms like GANITE [58] and SCIGAN [7] can be regarded as variants of the above training framework.

The first term of Eq. (22) and Eq. (25) are similar to the objective of GAIL by regarding $M_\theta$ as the "policy" to imitate and $\hat{\mu}$ as the "environment" to collect data. In the context of GALILEO, the objective of GAIL is:

$$ \theta_{t+1} = \arg\max_\theta \mathbb{E}_{\rho_{M_{\theta_t}}^\kappa} \left[ A_{\varphi^*}(x, a, x') \log M_\theta(x'|x, a) \right] $$

$$ s.t. \quad Q_{M_{\theta_t}}^\kappa(x, a, x') = \mathbb{E} \left[ \sum_t^\infty \gamma^t \log D_{\varphi^*}(x_t, a_t, x_{t+1}) | (x_t, a_t, x_{t+1}) = (x, a, x'), \kappa, M_{\theta_t} \right] $$

$$ \varphi^* = \arg\max_\varphi \mathbb{E}_{\rho_{M^*}^\kappa} \left[ \log D_\varphi(x, a, x') \right] + \mathbb{E}_{\rho_{M_{\theta_t}}^\kappa} \left[ \log(1 - D_\varphi(x, a, x')) \right], $$

where $A_{\varphi^*}(x, a, x') = Q_{M_\theta}^\kappa(x, a, x') - V_{M_\theta}^\kappa(x, a)$ and $V_{M_{\theta_t}}^\kappa(x, a) = \mathbb{E}_{M_{\theta_t}(x, a)} \left[ Q^\kappa(x, a, x') \right]$.

## F  Additional Related Work

Our primitive objective is inspired by weighted empirical risk minimization (WERM) based on inverse propensity score (IPS). WERM is originally proposed to solve the generalization problem of domain adaptation in machine learning literature. For instance, we would like to train a predictor $M(y|x)$ in a domain with distribution $P_{\mathrm{train}}(x)$ to minimize the prediction risks in the domain with distribution $P_{\mathrm{test}}(x)$, where $P_{\mathrm{test}} \neq P_{\mathrm{test}}$. To solve the problem, we can train a weighted objective with $\max_M \mathbb{E}_{x \sim P_{\mathrm{train}}}[\frac{P_{\mathrm{test}}(x)}{P_{\mathrm{train}}(x)} \log M(y|x)]$, which is called weighted empirical risk minimization methods [5, 4, 15, 9, 42]. These results have been extended and applied to causal inference, where the predictor is required to be generalized from the data distribution in observational studies (source domain) to the data distribution in randomized controlled trials (target domain) [48, 3, 22, 29, 28].

In this case, the input features include a state $x$ (a.k.a. covariates) and an action $a$ (a.k.a. treatment variable) which is sampled from a policy. We often assume the distribution of $x$, $P(x)$ is consistent between the source domain and the test domain, then we have $\frac{P_{\text{test}}(x)}{P_{\text{train}}(x)} = \frac{P(x)\beta(a|x)}{P(x)\mu(a|x)} = \frac{\beta(a|x)}{\mu(a|x)}$, where $\mu$ and $\beta$ are the policies in source and target domains respectively. In [48, 3, 22], the policy in randomized controlled trials is modeled as a uniform policy, then $\frac{P_{\text{test}}(x)}{P_{\text{train}}(x)} = \frac{P(x)\beta(a|x)}{P(x)\mu(a|x)} = \frac{\beta(a|x)}{\mu(a|x)} \propto \frac{1}{\mu(a|x)} \cdot \frac{1}{\mu(a|x)}$ is also known as inverse propensity score (IPS). In [28], it assumes that the policy in the target domain is predefined as $\beta(a|x)$ before environment model learning, then it uses $\frac{\beta}{\mu}$ as the IPS. The differences between AWRM and previous works are fallen in two aspects: (1) We consider the distribution-shift problem in the sequential decision-making scenario. In this scenario, we not only consider the action distribution mismatching between the behavior policy $\mu$ and the policy to evaluation $\beta$, but also the follow-up effects of policies to the state distribution; (2) For faithful offline policy optimization, we require the environment model to have generalization ability in numerous different policies. The objective of AWRM is proposed to guarantee the generalization ability of $M$ in numerous different policies instead of a specific policy.

On a different thread, there are also studies that bring counterfactual inference techniques of causal inference into model-based RL [8, 39, 49]. These works consider that the transition function is relevant to some hidden noise variables and use Pearl-style structural causal models (SCMs), which is a directed acyclic graphs to define the causality of nodes in an environment, to handle the problem. SCMs can help RL in different ways: [8] approximate the posterior of the noise variables based on the observation of data, and environment models are learned based on the inferred noises. The generalization ability is improved if we can infer the correct value of the noise variables. [39] discover several local causal structural models of a global environment model, then data augmentation strategies by leveraging these local structures to generate counterfactual experiences. [49] proposes a representation learning technique for causal factors, which is an instance of the hidden noise variables, in partially observable Markov decision processes (POMDPs). With the learned representation of causal factors, the performance of policy learning and transfer in downstream tasks will be improved.

Instead of considering the hidden noise variables in the environments, our study considers the environment model learning problem in the fully observed setting and focuses on unbiased causal effect estimation in the offline dataset under behavior policies collected with selection bias.

In offline model-based RL, the problem is called distribution shift [59, 31, 14] which has received great attentions. However, previous algorithms do not handle the model learning challenge directly but propose techniques to suppress policy sampling and learning in risky regions [59, 30]. Although these algorithms have made great progress in offline policy optimization in many tasks, so far, how to learn a better environment model in this scenario has rarely been discussed.

We are not the first article to use the concept of density ratio for weighting. In off-policy estimation, [32, 35, 60] use density ratio to evaluate the value of a given target policy $\beta$. These methods attempt to solve an accurate approximation of $\omega(s, a|\rho^\beta)$. The objective of our work, AWRM, is for the model to provide faithful feedback for different policies, formalized as minimizing the model error of the density function for any $\beta$ in the policy space $\Pi$. The core of this problem is how to obtain an approximation of the density function of the best-response $\beta^*$ corresponding to the current model, and then approximate the corresponding $\omega(s, a|\rho^{\beta^*}))$. It should be emphasized that since $\beta^*$ is unknown in advance and will change as the induction bias of $M$ is changed, the solutions proposed in [32, 35, 60] cannot be applied to AWRM; Recently, [24, 57] use density ratio weighting to learn a model. The purpose of weighting is to make the model adapt to the current policy learning and adjust the model learning according to the current policy, which is the same as the WERM objective proposed in Def. 4.1. [54] also utilizes density ratio weighting to learn a model. Instead of estimating them based on the offline dataset and target policy as previous works do, they propose to design an adversarial model learning objective by constructing two function classes $\mathcal{V}$ and $\mathcal{W}$, satisfying the target policy's value $V_\beta$ and density ratio $\omega_\beta$ are covered, i.e., $V_\beta \in \mathcal{V}$ and $\omega_\beta \in \mathcal{W}$. Different from previous articles, our approach uses adversarial weighting to learn a universal model that provides good feedback for any target policy $\beta \in \Pi$, i.e., AWRM, instead of learning a model suitable to a specific target policy.

# G  Experiment Details

## G.1  Settings

### G.1.1  General Negative Feedback Control (GNFC)

The design of GNFC is inspired by a classic type of scenario that behavior policies $\mu$ have selection bias and easily lead to counterfactual risks: For some internet platforms, we would like to allocate budgets to a set of targets (e.g., customers or cities) to increase the engagement of the targets in the platforms. Our task is to train a model to predict targets' feedback on engagement given targets' features and allocated budgets.

In these tasks, for better benefits, the online working policy (i.e., the behavior policy) will tend to cut down the budgets if targets have better engagement, otherwise, the budgets might be increased. The risk of counterfactual environment model learning in the task is that: the object with better historical engagement will be sent to smaller budgets because of the selection bias of the behavior policies, then the model might exploit this correlation for learning and get a conclusion that: increasing budgets will reduce the targets' engagement, which violates the real causality. We construct an environment and a behavior policy to mimic the above process. In particular, the behavior policy $\mu_{GNFC}$ is

$$\mu_{GNFC}(x) = \frac{(62.5 - \text{mean}(x))}{15} + \epsilon,$$

where $\epsilon$ is a sample noise, which will be discussed later. The environment includes two parts:

(1) response function $M_1(y|x,a)$:

$$M_1(y|x,a) = \mathcal{N}(\text{mean}(x) + a, 2)$$

(2) mapping function $M_2(x'|x,y)$:

$$M_2(x'|x,a,y) = y - \text{mean}(x) + x$$

The transition function $M^*$ is a composite of $M^*(x'|x,a) = M_2(x'|x,a,M_1(y|x,a))$. The behavior policies have selection bias: the actions taken are negatively correlated with the states, as illustrated in Fig. 10(a) and Fig. 10(b). We control the difficulty of distinguishing the correct causality of $x$, $a$, and $y$ by designing different strategies of noise sampling on $\epsilon$. In principle, with a larger number or more pronounced disturbances, there are more samples violating the correlation between $x$ and $a$, then more samples can be used to find the correct causality. Therefore, we can control the difficulty of counterfactual environment model learning by controlling the strength of disturbance. In particular, we sample $\epsilon$ from a uniform distribution $U(-e,e)$ with probability $p$. That is, $\epsilon = 0$ with probability $1 - p$ and $\epsilon \sim U(-e,e)$ with probability $p$. Then with larger $p$, there are more samples in the dataset violating the negative correlation (i.e., $\mu_{GNFC}$), and with larger $e$, the difference of the feedback will be more obvious. By selecting different $e$ and $p$, we can construct different tasks to verify the effectiveness and ability of the counterfactual environment model learning algorithm.

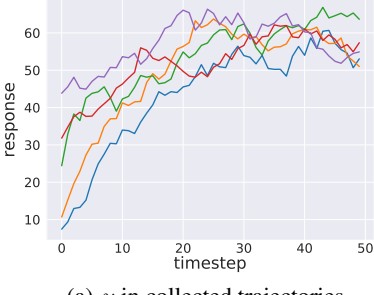
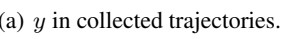

(a) $y$ in collected trajectories.

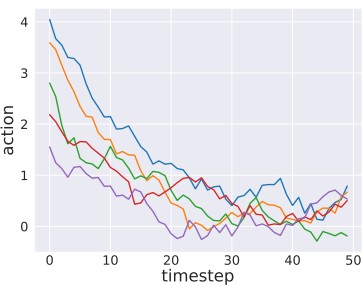

(b) $a$ in collected trajectories.

Figure 10: Illustration of information about the collected dataset in GNFC. Each color of the line denotes one of the collected trajectories. The X-axis denotes the timestep of a trajectory.

### G.1.2 The Cancer Genomic Atlas (TCGA)

The Cancer Genomic Atlas (TCGA) is a project that has profiled and analyzed large numbers of human tumors to discover molecular aberrations at the DNA, RNA, protein, and epigenetic levels. The resulting rich data provide a significant opportunity to accelerate our understanding of the molecular basis of cancer. We obtain features, $\mathbf{x}$, from the TCGA dataset and consider three continuous treatments as done in SCIGAN [7]. Each treatment, $a$, is associated with a set of parameters, $\mathbf{v}_1$, $\mathbf{v}_2$, $\mathbf{v}_3$, that are sampled randomly by sampling a vector from a standard normal distribution and scaling it with its norm. We assign interventions by sampling a treatment, $a$, from a beta distribution, $a \mid \mathbf{x} \sim \text{Beta}(\alpha, \beta)$. $\alpha \geq 1$ controls the sampling bias and $\beta = \frac{\alpha - 1}{a^*} + 2 - \alpha$, where $a^*$ is the optimal treatment. This setting of $\beta$ ensures that the mode of $\text{Beta}(\alpha, \beta)$ is $a^*$.

The calculation of treatment response and optimal treatment are shown in Table 4.

Table 4: Treatment response used to generate semi-synthetic outcomes for patient features $\mathbf{x}$. In the experiments, we set $C = 10$.

| Treatment | Treatment Response | Optimal treatment |
|---|---|---|
| 1 | $f_1(\mathbf{x}, a_1) = C\left(\left(\mathbf{v}_1^1\right)^T \mathbf{x} + 12\left(\mathbf{v}_2^1\right)^T \mathbf{x} a_1 - 12\left(\mathbf{v}_3^1\right)^T \mathbf{x} a_1^2\right)$ | $a_1^* = \frac{\left(\mathbf{v}_2^1\right)^T \mathbf{x}}{2\left(\mathbf{v}_3^1\right)^T \mathbf{x}}$ |
| 2 | $f_2(\mathbf{x}, a_2) = C\left(\left(\mathbf{v}_1^2\right)^T \mathbf{x} + \sin\left(\pi\left(\frac{\mathbf{v}_2^{2T}\mathbf{x}}{\mathbf{v}_3^{2T}\mathbf{x}}\right) a_2\right)\right)$ | $a_2^* = \frac{\left(\mathbf{v}_3^2\right)^T \mathbf{x}}{2\left(\mathbf{v}_2^2\right)^T \mathbf{x}}$ |
| 3 | $f_3(\mathbf{x}, a_3) = C\left(\left(\mathbf{v}_1^3\right)^T \mathbf{x} + 12 a_3(a_3 - b)^2\right)$, where $b = 0.75\frac{\left(\mathbf{v}_2^3\right)^T \mathbf{x}}{\left(\mathbf{v}_3^3\right)^T \mathbf{x}}$ | $\frac{3}{b}$ if $b \geq 0.75$, 1 if $b < 0.75$ |

We conduct experiments on three different treatments separately and change the value of bias $\alpha$ to assess the robustness of different methods to treatment bias. When the bias of treatment is large, which means $\alpha$ is large, the training set contains data with a strong bias on treatment so it would be difficult for models to appropriately predict the treatment responses out of the distribution of training data.

### G.1.3 Budget Allocation task to the Time period (BAT)

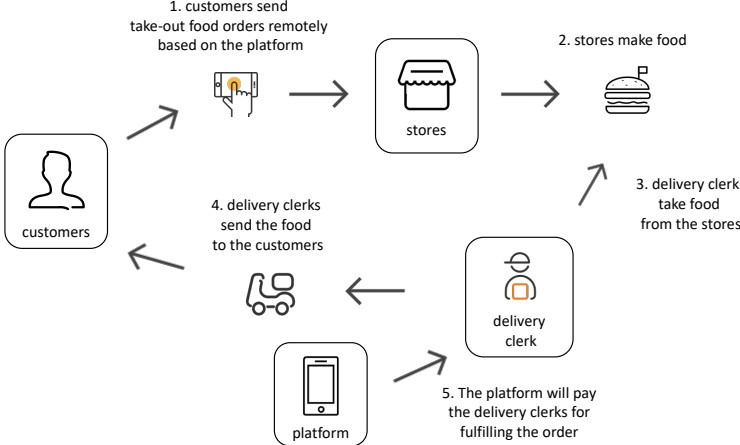

Figure 11: Illustration of the workflow of the food-delivery platform.

We deploy GALILEO in a real-world large-scale food-delivery platform. The platform contains various food stores, and food delivery clerks. The overall workflow is as follows: the platform presents the nearby food stores to the customers and the customers make orders, i.e., purchase take-out foods from some stores on the platform. The food delivery clerks can select orders from the platform to fulfill. After an order is selected to fulfill, the delivery clerks will take the ordered take-out foods from the stores and then send the food to the customers. The platform will pay the delivery clerks (mainly in proportion to the distance between the store and the customers' location) once the orders are fulfilled. An illustration of the workflow can be found in Fig. 11.

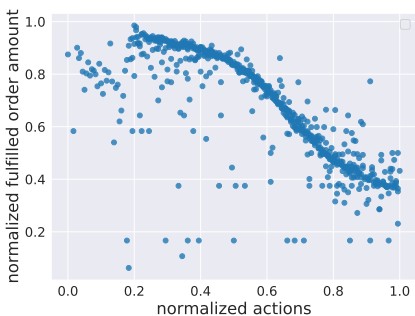

Figure 12: Illustration of relationship between user feedback and the actions of the offline dataset in the real-world food-delivery platform.

However, there is an imbalance problem between the demanded orders from customers and the supply of delivery clerks to fulfill these orders. For example, at peak times like lunchtime, there will be many more demanded orders than at other times, and the existed delivery clerks might not be able to fulfill all of these orders timely. The goal of the Budget Allocation task to the Time period (BAT) is to handle the imbalance problem in time periods by sending reasonable allowances to different time periods. More precisely, the goal of BAT is to make all orders (i.e., the demand) sent in different time periods can be fulfilled (i.e., the supply) timely.

The core challenge of the environment model learning in BAT tasks is similar to the challenge in Fig. 1. Specifically, the behavior policy in BAT tasks is a human-expert policy, which will tend to increase the budget of allowance in the time periods with a lower supply of delivery clerks, otherwise will decrease the budget (Fig. 12 gives an instance of this phenomenon in the real data).

To handle the imbalance problem in different time periods, in the platform, the orders in different time periods $t \in [0, 1, 2..., 23]$ will be allocated with different allowances $c \in \mathcal{N}^+$. For example, at 10 A.M. (i.e., $t = 10$), we add 0.5\$ (i.e., $c = 0.5$) allowances to all of the demanded orders. From 10 A.M. to 11 A.M., the delivery clerks who take orders and send food to customers will receive extra allowances. Specifically, if the platform pays the delivery clerks 2\$ for fulfilling the order, now he/she will receive 2.5\$. For each day, the budget of allowance $C$ is fixed. We should find the best budget allocation policy $\pi^*(c|t)$ of the limited budget $C$ to make as many orders as possible can be taken timely.

To find the policy, we first learn a model to reconstruct the response of allowance for each delivery clerk $\hat{M}(y_{t+1}|s_t, p_t, c_t)$, where $y_{t+1}$ is the taken orders of the delivery clerks in state $s_t$, $c_t$ is the allowances, $p_t$ denotes static features of the time period $t$. In particular, the state $s_t$ includes historical order-taken information of the delivery clerks, current orders information, the feature of weather, city information, and so on. Then we use a rule-based mapping function $f$ to fill the complete next time-period states, i.e., $s_{t+1} = f(s_t, p_t, c_t, y_{t+1})$. Here we define the composition of the above functions $\hat{M}$ and $f$ as $\hat{M}_f$. Finally, we learn a budget allocation policy based on the learned model. For each day, the policy we would like to find is:

$$\max_\pi \mathbb{E}_{s_0 \sim \mathcal{S}} \left[ \sum_{t=0}^{23} y_t | \hat{M}_f, \pi \right],$$

$$\text{s.t.,} \sum_{t,s \in \mathcal{S}} c_t y_t \leq C$$

In our experiment, we evaluate the degree of balancing between demand and supply by computing the averaged five-minute order-taken rate, that is the percentage of orders picked up within five minutes. Note that the behavior policy is fixed for the long term in this application. So we directly use the data replay with a small scale of noise (See Tab. 6) to reconstruct the behavior policy for model learning in GALILEO.

**Also note that although we model the response for each delivery clerk, for fairness, the budget allocation policy is just determining the allowance of each time period $t$ and keeps the allowance to each delivery clerk $s$ the same.**

### G.2 Baseline Algorithms

The algorithm we compared are: (1) Supervised Learning (SL): training a environment model to minimize the expectation of prediction error, without considering the counterfactual risks; (2) inverse propensity weighting (IPW) [50]: a practical way to balance the selection bias by re-weighting. It can be regarded as $\omega = \frac{1}{\hat{\mu}}$, where $\hat{\mu}$ is another model learned to approximate the behavior policy; (3) SCIGAN: a recent proposed adversarial algorithm for model learning for continuous-valued interventions [7]. All of the baselines algorithms are implemented with the same capacity of neural networks (See Tab. 6).

#### G.2.1 Supervised Learning (SL)

As a baseline, we train a multilayer perceptron model to directly predict the response of different treatments, without considering the counterfactual risks. We use mean square error to estimate the performance of our model so that the loss function can be expressed as $MSE = \frac{1}{n} \sum_{i=1}^{n} (y_i - \hat{y}_i)^2$, where $n$ is the number of samples, $y$ is the true value of response and $\hat{y}$ is the predicted response. In practice, we train our SL models using Adam optimizer and the initial learning rate $3e^{-4}$ on both datasets TCGA and GNFC. The architecture of the neural networks is listed in Tab. 6.

#### G.2.2 Inverse Propensity Weighting (IPW)

Inverse propensity weighting [50] is an approach where the treatment outcome model uses sample weights to balance the selection bias by re-weighting. The weights are defined as the inverse propensity of actually getting the treatment, which can be expressed as $\frac{1}{\hat{\mu}(a|x)}$, where $x$ stands for the feature vectors in a dataset, $a$ is the corresponding action and $\hat{\mu}(a|x)$ indicates the action taken probability of $a$ given the features $x$ within the dataset. $\hat{\mu}$ is learned with standard supervised learning. Standard IPW leads to large weights for the points with small sampling probabilities and finally makes the learning process unstable. We solve the problem by clipping the propensity score: $\hat{\mu} \leftarrow \min(\hat{\mu}, 0.05)$, which is common used in existing studies [27]. The loss function can thus be expressed as $\frac{1}{n} \sum_{i=1}^{n} \frac{1}{\hat{\mu}(a_i|x_i)} (y_i - \hat{y}_i)^2$. The architecture of the neural networks is listed in Tab. 6.

#### G.2.3 SCIGAN

SCIGAN [7] is a model that uses generative adversarial networks to learn the data distribution of the counterfactual outcomes and thus generate individualized response curves. SCIGAN does not place any restrictions on the form of the treatment-does response functions and is capable of estimating patient outcomes for multiple treatments, each with an associated parameter. SCIGAN first trains a generator to generate response curves for each sample within the training dataset. The learned generator can then be used to train an inference network using standard supervised methods. For fair comparison, we increase the number of parameters for the open-source version of SCIGAN so that the SCIGAN model can have same order of magnitude of network parameters as GALILEO. In addition, we also finetune the hyperparameters (Tab. 5) of the enlarged SCIGAN to realize its full strength. We set num_dosage_samples 9 and $\lambda = 10$.

Table 5: Table of hyper-parameters for SCIGAN.

| Parameter | Values |
|---|---|
| Number of samples | 3, 5, 7, 9, 11 |
| $\lambda$ | 0.1, 1, 10, 20 |

### G.3 Hyper-parameters

We list the hyper-parameter of GALILEO in Tab. 6.

### G.4 Computation Resources

We use one Tesla V100 PCIe 32GB GPU and a 32-core Intel(R) Xeon(R) Gold 5118 CPU @ 2.30GHz to train all of our models.

Table 6: Table of hyper-parameters for all of the tasks.

| Parameter | GNFC | TAGC | MuJoCo | BAT |
|---|---|---|---|---|
| hidden layers of all neural networks | 4 | 4 | 5 | 5 |
| hidden units of all neural networks | 256 | 256 | 512 | 512 |
| collect samples for each time of model update | 5000 | 5000 | 40000 | 96000 |
| batch size of discriminators | 5000 | 5000 | 40000 | 80000 |
| horizon | 50 | 1 | 100 | 48 (half-hours) |
| $\epsilon_\mu$ (also $\epsilon_D$) | 0.005 | 0.01 | 0.05 (0.1 for walker2d) | 0.05 |
| times for discriminator update | 2 | 2 | 1 | 5 |
| times for model update | 1 | 1 | 2 | 20 |
| times for supervised learning update | 1 | 1 | 4 | 20 |
| learning rate for supervised learning | 1e-5 | 1e-5 | 3e-4 | 1e-5 |
| $\gamma$ | 0.99 | 0.0 | 0.99 | 0.99 |
| clip-ratio | NAN | NAN | NAN | 0.1 |
| max $D_{KL}$ | 0.001 | 0.001 | 0.001 | NAN |
| optimization algorithm (the first term of Eq. (22)) | TRPO | TRPO | TRPO | PPO |

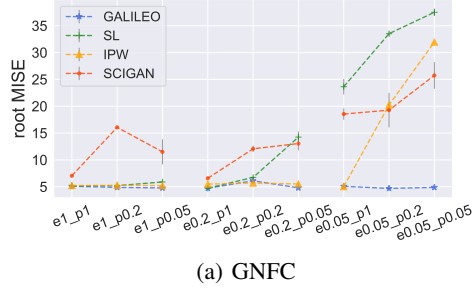

(a) GNFC

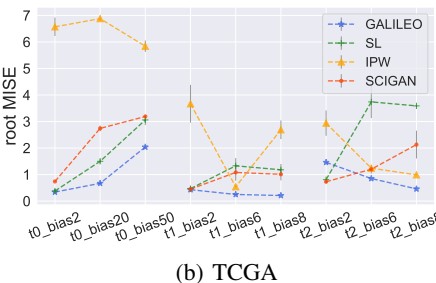

(b) TCGA

Figure 13: Illustration of the performance in GNFC and TCGA. The grey bar denotes the standard error ($\times 0.3$ for brevity) of 3 random seeds.

## H  Additional Results

### H.1  Test in Single-step Environments

The results of GNFC tasks are summarized in Fig. 13(a) and the detailed results can be found in Tab. 11. The results show that the property of the behavior policy (i.e., $e$ and $p$) dominates the generalization ability of the baseline algorithms. When $e = 0.05$, almost all of the baselines fail and give a completely opposite response curve. IPW still perform well when $0.2 \le e \le 1.0$ but fails when $e = 0.05, p <= 0.2$. We also found that SCIGAN can reach a better performance than other baselines when $e = 0.05, p <= 0.2$, but the results in other tasks are unstable. GALILEO is the only algorithm that is robust to the selection bias and outputs correct response curves in all of the tasks. Based on the experiment, we also indicate that the commonly used overlap assumption is unreasonable to a certain extent especially in real-world applications since it is impractical to inject noises into the whole action space. The problem of overlap assumption being violated should be taken into consideration otherwise the algorithm will be hard to use in practice if it is sensitive to the noise range.

The results of TCGA tasks are summarized in Fig. 13(b) and the detailed results can be found in Tab. 12. We found the phenomenon in this experiment is similar to the one in GNFC, which demonstrates the compatibility of GALILEO to single-step environments. We also found that the results of IPW are unstable in this experiment. It might be because the behavior policy is modeled with beta distribution while the propensity score $\hat{\mu}$ is modeled with Gaussian distribution. Since IPW directly reweight loss function with $\frac{1}{\hat{\mu}}$, the results are sensitive to the error on $\hat{\mu}$. GALILEO also models $\hat{\mu}$ with Gaussian distribution but the results are more stable since GALILEO does not re-weight through $\hat{\mu}$ explicitly.

We give the averaged responses for all of the tasks and the algorithms in Fig. 21 to Fig. 28. We randomly select 20% of the states in the dataset and equidistantly sample actions from the action space for each sampled state, and plot the averaged predicted feedback of each action. The real response is slightly different among different figure as the randomly-selected states for testing is different. We sample 9 points in GNFC tasks and 33 points in TAGC tasks for plotting.

## H.2 All of the Result Table

We give the result of CNFC in Tab. 11, TCGA in Tab. 12, BAT in Tab. 9, and MuJoCo in Tab. 10.

## H.3 Ablation Studies

In Appx. E.1, we introduce several techniques to develop a practical GALILEO algorithm. Based on task `e0.2_p0.05` in GNFC, we give the ablation studies to investigate the effects of these techniques. We first compare two variants that do not handle the assumptions violation problems: (1) `NO_INJECT_NOISE`: set $\epsilon_\mu$ and $\epsilon_D$ to zero, which makes the overlap assumption not satisfied;; (2) `SINGLE_SL`: without replacing the second term in Eq. (6) with standard supervised learning even when the output probability of $D$ is far away from $0.5$. Besides, we introduced several tricks inspired by GAIL and give a comparison of these tricks and GAIL: (3) `ONE_STEP`: use one-step reward instead of cumulative rewards (i.e., Q and V; see Eq. (23) and Eq. (24)) for re-weighting, which is implemented by set $\gamma$ to 0; (4) `SINGE_DIS`: remove $T_{\varphi_1^*}(x, a)$ and replace it with $\mathbb{E}_{M_\theta}\left[T_{\varphi_0^*}(x, a, x')\right]$, which is inspired by GAIL that uses a value function as a baseline instead of using another discriminator; (5) `PURE_GAIL`: remove the

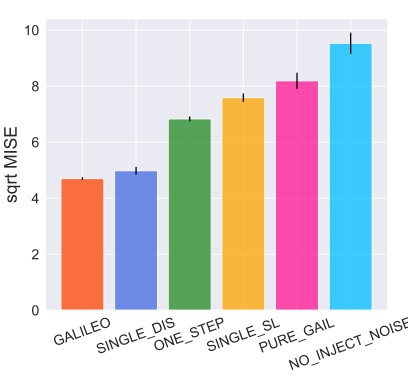

Figure 14: Illustration of the ablation studies. The error bars are the standard error.

second term in Eq. (6). It can be regarded as a naive adoption of GAIL and a partial implementation of GALILEO.

We summarize the results in Fig. 14. Based on the results of `NO_INJECT_NOISE` and `SINGLE_SL`, we can see that handling the assumption violation problems is important and will increase the ability on counterfactual queries. The results of `PURE_GAIL` tell us that the partial implementation of GALILEO is not enough to give stable predictions on counterfactual data; On the other hand, the result of `ONE_STEP` also demonstrates that embedding the cumulative error of one-step prediction is helpful for GALILEO training; Finally, we also found that `SINGLE_DIS` nearly has almost no effect on the results. It suggests that, empirically, we can use $\mathbb{E}_{M_\theta}\left[T_{\varphi_0^*}(x, a, x')\right]$ as a replacement for $T_{\varphi_1^*}(x, a)$, which can reduce the computation costs of the extra discriminator training.

## H.4 Worst-Case Prediction Error

In theory, GALILEO increases the generalization ability by focusing on the worst-case samples' training to achieve AWRM. To demonstrate the property, we propose a new metric named Mean-Max Square Error (MMSE): $\mathbb{E}\left[\max_{a \in \mathcal{A}}\left(M^*(x'|x, a) - M(x'|x, a)\right)^2\right]$ and give the results of MMSE for GNFC in Tab. 13 and for TCGA in Tab. 14.

## H.5 Detailed Results in the MuJoCo Tasks

We select 3 environments from D4RL [17] to construct our model learning tasks. We compare it with a typical transition model learning algorithm used in the previous offline model-based RL algorithms [59, 30], which is a variant of standard supervised learning. We name the method OFF-SL. Besides, we also implement IPW and SCIGAN as the baselines. We train models in datasets HalfCheetah-medium, Walker2d-medium, and Hopper-medium, which are collected by a behavior policy with 1/3 performance to the expert policy, then we test them in the corresponding expert dataset. For better training efficiency, the trajectories in the training and testing datasets are truncated, remaining *the first 100 steps*. We plot the converged results and learning curves of the three MuJoCo tasks in Tab. 10 and Fig. 15 respectively.

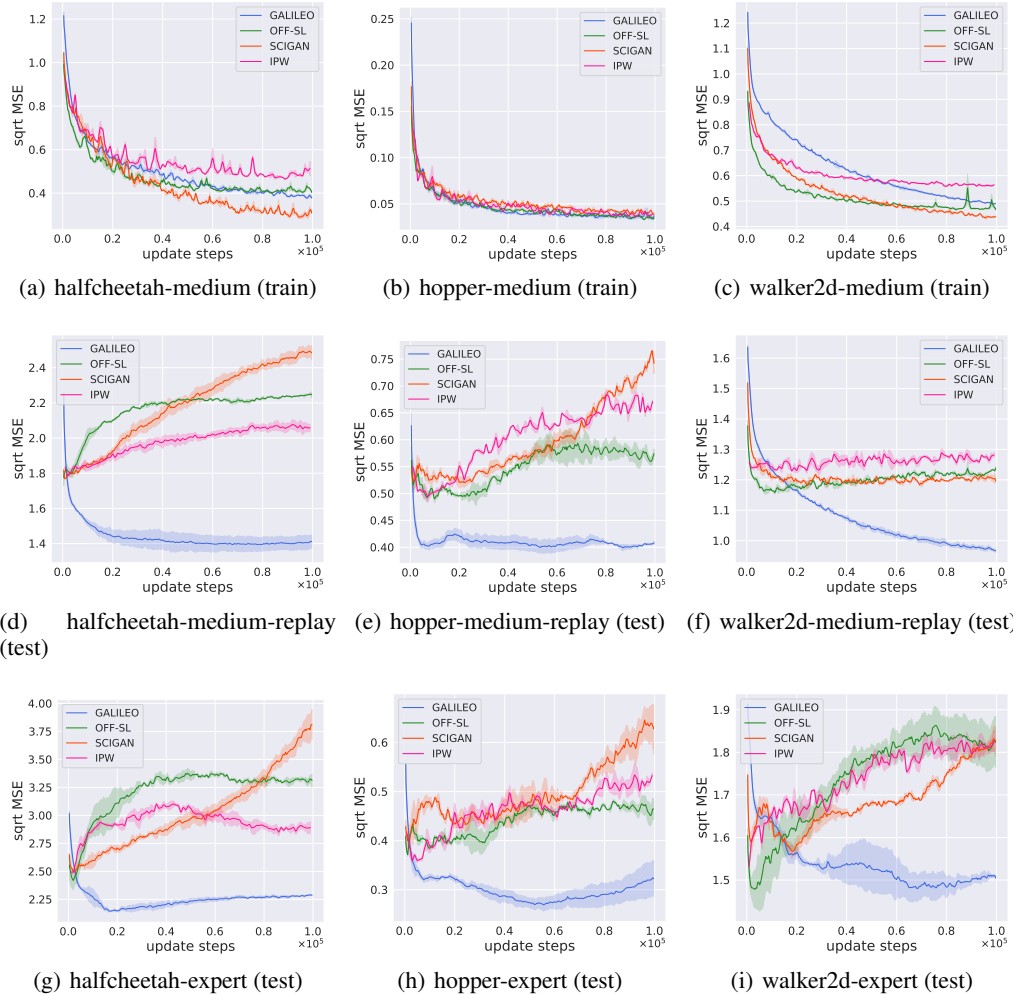

(a) halfcheetah-medium (train)    (b) hopper-medium (train)    (c) walker2d-medium (train)

(d) halfcheetah-medium-replay (test)    (e) hopper-medium-replay (test)    (f) walker2d-medium-replay (test)

(g) halfcheetah-expert (test)    (h) hopper-expert (test)    (i) walker2d-expert (test)

Figure 15: Illustration of learning curves of the MuJoCo Tasks. The X-axis record the steps of the environment model update, and the Y-axis is the corresponding prediction error. The figures with titles ending in "(train)" means the dataset is used for training while the titles ending in "(test)" means the dataset is *just used for testing*. The solid curves are the mean reward and the shadow is the standard error of three seeds.

In Fig. 15, we can see that all algorithms perform well in the training datasets. OFF-SL and SCIGAN can even reach a bit lower error in halfcheetah and walker2d. However, when we verify the models through "expert" and "medium-replay" datasets, which are collected by other policies, the performance of GALILEO is significantly more stable and better than all other algorithms. As the training continues, the baseline algorithms even gets worse and worse. However, whether in GALILEO or other baselines, the performance for testing is at least 2x worse than in the training

dataset, and the error is large especially in halfcheetah. The phenomenon indicates that although GALILEO can make better performances for counterfactual queries, the risks of using the models are still large and still challenging to be further solved.

Table 7: Results of policy performance directly optimized through SAC [20] using the learned dynamics models and deployed in MuJoCo environments. MAX-RETURN is the policy performance of SAC in the MuJoCo environments, and "avg. norm." is the averaged normalized return of the policies in the 9 tasks, where the returns are normalized to lie between 0 and 100, where a score of 0 corresponds to the worst policy, and 100 corresponds to MAX-RETURN.

| Task | Hopper | | | Walker2d | | | HalfCheetah | | | avg. norm. |
|---|---|---|---|---|---|---|---|---|---|---|
| Horizon | H=10 | H=20 | H=40 | H=10 | H=20 | H=40 | H=10 | H=20 | H=40 | / |
| GALILEO | $13.0 \pm 0.1$ | $33.2 \pm 0.1$ | $53.5 \pm 1.2$ | $11.7 \pm 0.2$ | $29.9 \pm 0.3$ | $61.2 \pm 3.4$ | $0.7 \pm 0.2$ | $-1.1 \pm 0.2$ | $-14.2 \pm 1.4$ | **51.1** |
| OFF-SL | $4.8 \pm 0.5$ | $3.0 \pm 0.2$ | $4.6 \pm 0.2$ | $10.7 \pm 0.2$ | $20.1 \pm 0.8$ | $37.5 \pm 6.7$ | $0.4 \pm 0.5$ | $-1.1 \pm 0.6$ | $-13.2 \pm 0.3$ | 21.1 |
| IPW | $5.9 \pm 0.7$ | $4.1 \pm 0.6$ | $5.9 \pm 0.2$ | $4.7 \pm 1.1$ | $2.8 \pm 3.9$ | $14.5 \pm 1.4$ | $1.6 \pm 0.2$ | $0.5 \pm 0.8$ | $-11.3 \pm 0.9$ | 19.7 |
| SCIGAN | $12.7 \pm 0.1$ | $29.2 \pm 0.6$ | $46.2 \pm 5.2$ | $8.4 \pm 0.5$ | $9.1 \pm 1.7$ | $1.0 \pm 5.8$ | $1.2 \pm 0.3$ | $-0.3 \pm 1.0$ | $-11.4 \pm 0.3$ | 41.8 |
| MAX-RETURN | $13.2 \pm 0.0$ | $33.3 \pm 0.2$ | $71.0 \pm 0.5$ | $14.9 \pm 1.3$ | $60.7 \pm 11.1$ | $221.1 \pm 8.9$ | $2.6 \pm 0.1$ | $13.3 \pm 1.1$ | $49.1 \pm 2.3$ | 100.0 |

We then verify the generalization ability of the learned models above by adopting them into offline model-based RL. Instead of designing sophisticated tricks to suppress policy exploration and learning in risky regions as current offline model-based RL algorithms [59, 30] do, we just use the standard SAC algorithm [20] to exploit the models for policy learning to strictly verify the ability of the models. Unfortunately, we found that the compounding error will still be inevitably large in the 1,000-step rollout, which is the standard horizon in MuJoCo tasks, leading all models to fail to derive a reasonable policy. To better verify the effects of models on policy optimization, we learn and evaluate the policies with three smaller horizons: $H \in \{10, 20, 40\}$.

The results have been listed in Tab. 7, where the learning curve in the dynamics models and the real environments is shown in Fig. 16 and Fig. 17. We first averaged the normalized return (refer to "avg. norm.") under each task, and we can see that the policy obtained by GALILEO is significantly higher than other models (the improvements are 24% to 161%). At the same time, we found that SCIGAN performed better in policy learning, while IPW performed similarly to SL. This is in line with our expectations, since IPW only considers the uniform policy as the target policy for debiasing, while policy optimization requires querying a wide variety of policies. Minimizing the prediction risks only under a uniform policy cannot yield a good environment model for policy optimization. On the other hand, SCIGAN, as a partial implementation of GALILEO (refer to Appx. E.2), also roughly achieves AWRM and considers the cumulative effects of policy on the state distribution, so its overall performance is better; In addition, we find that GALILEO achieves significant improvement in 6 of the 9 tasks. But in HalfCheetah, IPW works slightly better. However, compared with MAX-RETURN, it can be found that all methods fail to derive reasonable policies because their policies' performances are far away from the optimal policy. By further checking the trajectories, we found that all the learned policies just keep the cheetah standing in the same place or even going backward and fall down [4].

## H.6 Off-policy Evaluation (OPE) in the MuJoCo Tasks

### H.6.1 Training and Evaluation Settings

We select 3 environments from D4RL [17] to construct our model learning tasks as Appx. H.5. To match the experiment setting in DOPE [18], here we use the whole datasets to the train GALILEO model, instead of truncated dataset in Appx. H.5, for GALILEO model training.

OPE via a learned dynamics model is straightforward, which only needs to compute the return using simulated trajectories generated by the evaluated policy under the learned dynamics model. Due to the stochasticity in the model and the policy, we estimate the return for a policy with Monte-Carlo sampling. See Alg. 4 for pseudocode, where we use $\gamma = 0.995$, $N = 10$, $H = 1000$ for all of the tasks.

### H.6.2 Metrics

The metrics we use in our paper are defined as follows:

---

[4]the videos can be found in https://github.com/xionghuichen/GALILEO

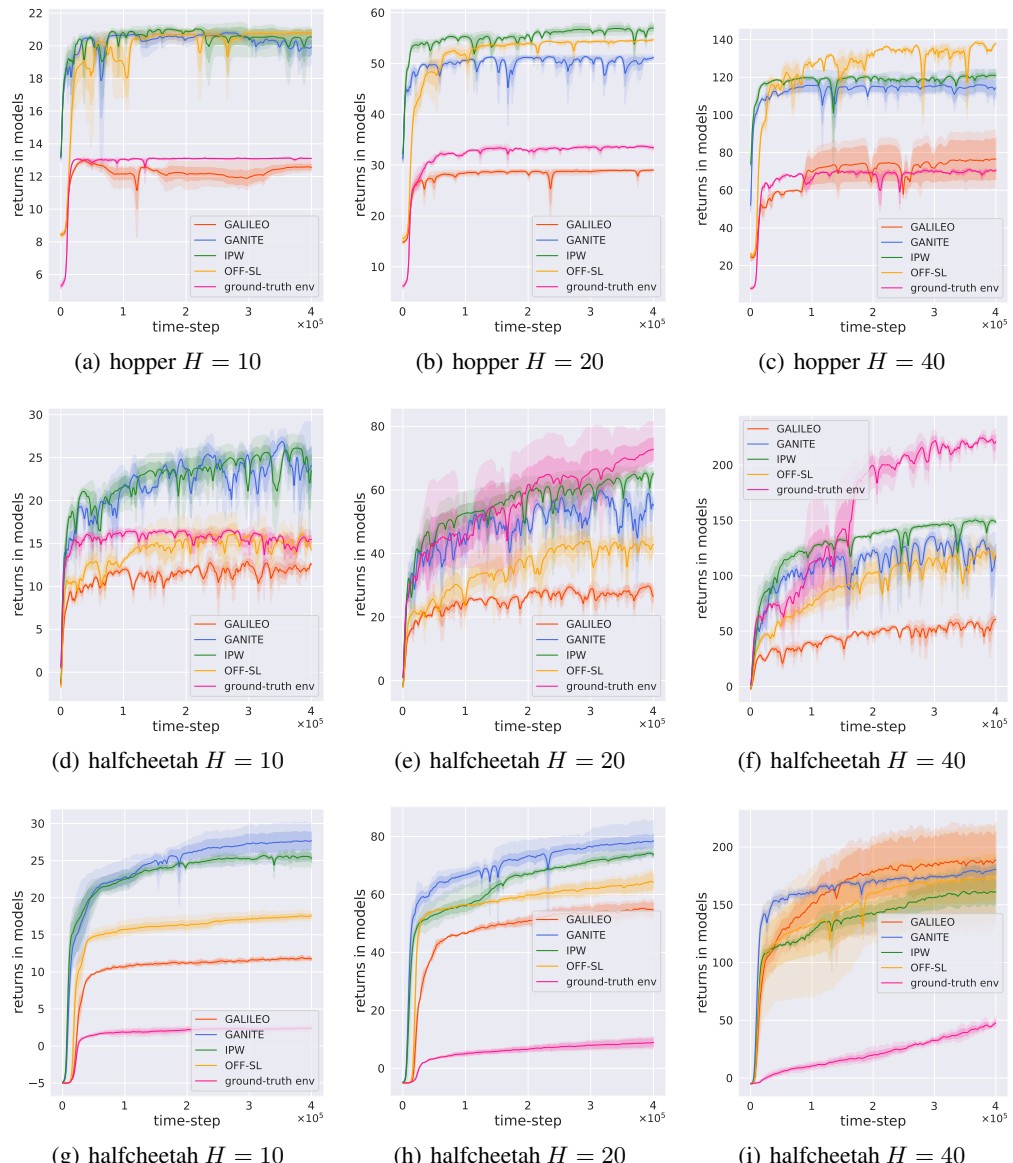

Figure 16: Illustration of offline policy learning curves of the MuJoCo Tasks. The X-axis record the steps of the environment model update, and the Y-axis is the corresponding returns in the **dynamics models**. The solid curves are the mean reward and the shadow is the standard error of three seeds.

**Absolute Error** The absolute error is defined as the difference between the value and estimated value of a policy:

$$\text{AbsErr} = |V^\pi - \hat{V}^\pi|, \tag{28}$$

where $V^\pi$ is the true value of the policy and $\hat{V}^\pi$ is the estimated value of the policy.

**Rank correlation** Rank correlation measures the correlation between the ordinal rankings of the value estimates and the true values, which can be written as:

$$\text{RankCorr} = \frac{\text{Cov}(V^\pi_{1:N}, \hat{V}^\pi_{1:N})}{\sigma(V^\pi_{1:N})\sigma(\hat{V}^\pi_{1:N})}, \tag{29}$$

where $1:N$ denotes the indices of the evaluated policies.

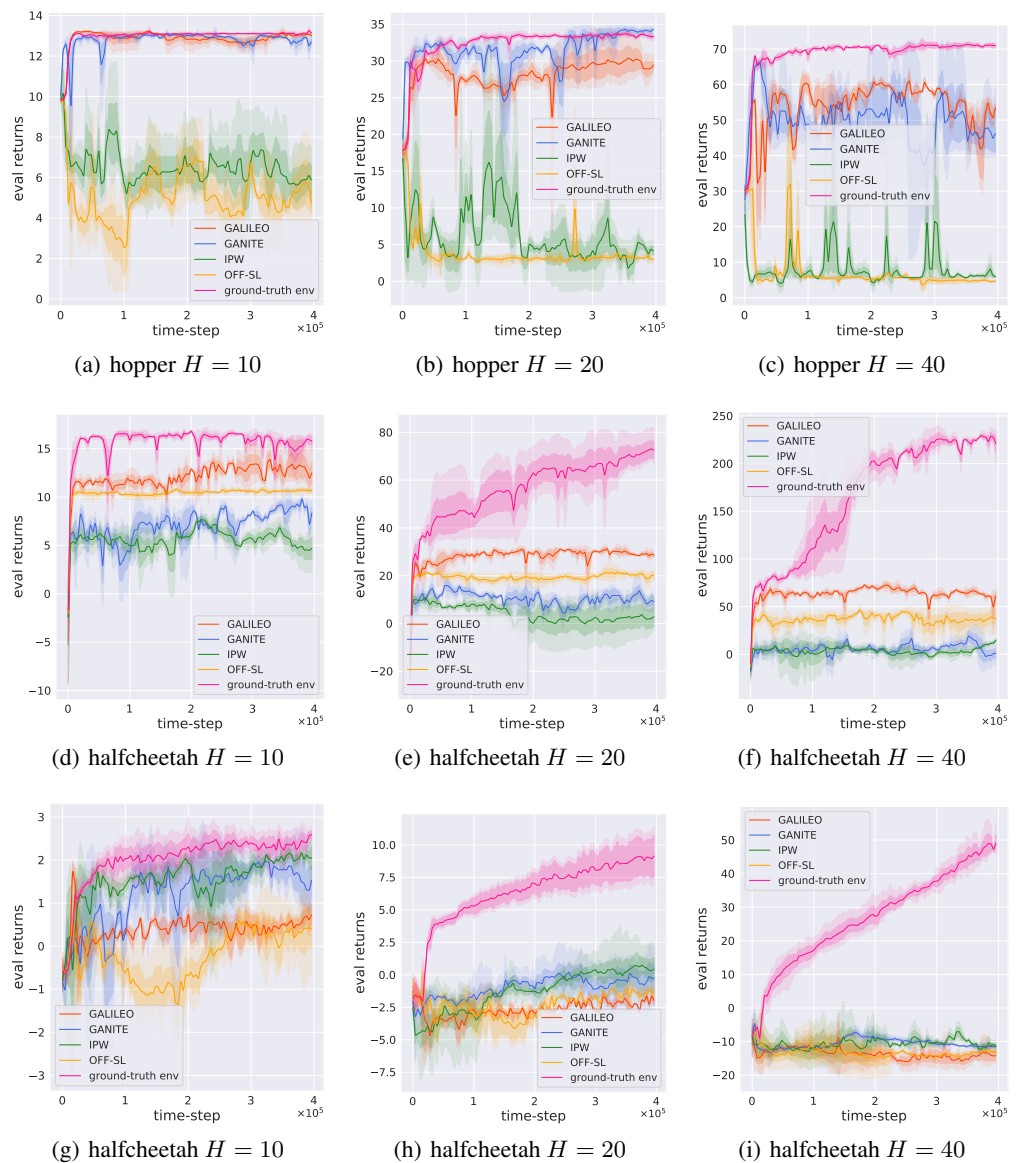

Figure 17: Illustration of offline policy learning curves of the MuJoCo Tasks. The X-axis record the steps of the environment model update, and the Y-axis is the corresponding returns in the **ground-truth environments**. The solid curves are the mean reward and the shadow is the standard error of three seeds.

**Regret@k** Regret@k is the difference between the value of the best policy in the entire set, and the value of the best policy in the top-k set (where the top-k set is chosen by estimated values). It can be defined as:

$$\text{Regret @k} = \max_{i \in 1:N} V_i^\pi - \max_{j \in \text{topk}(1:N)} V_j^\pi, \tag{30}$$

where topk$(1 : N)$ denotes the indices of the top K policies as measured by estimated values $\hat{V}^\pi$.

**Algorithm 4** Off-policy Evaluation with GALILEO model

---

**Require:** GALILEO model $(M_\theta)$, evaluated policy $\pi$, number of rollouts $N$. set of initial states $\mathcal{S}_0$, discount factor $\gamma$, horizon length $H$.

**for** $i = 1$ **to** $N$ **do**
    $R_i = 0$
    Sample initial state $s_0 \sim \mathcal{S}_0$
    Initialize $\tau_{-1} = \mathbf{0}$
    **for** $t = 0$ **to** $H - 1$ **do**
        $a_t \sim \pi(\cdot|s_t)$
        $s_{t+1}, r_t \sim M_\theta(\cdot|s_t, a_t)$
        $R_i = R_i + \gamma^t r_t$
    **end for**
**end for**
**return** $\frac{1}{N} \sum_{i=1}^{N} R_i$

---

### H.6.3 Detailed Results

The results of OPE on three tasks are in Tab. 8. Firstly, we can see that for all recorded rank-correlation scores, GALILEO is significantly better than all of the baseline methods, with at least 25 % improvements. As for regret@1, although GALILEO cannot reach the best performance among all of the tasks, it is always one of the top-2 methods among the three tasks, which demonstrates the stability of GALILEO. Finally, GALILEO has the smallest value gaps except for Halfcheetah. However, for all of the top-4 methods, the value gaps in halfcheetah are around 1,200. Thus we believe that their performances on value gaps are roughly at the same level.

Table 8: Results of OPE on DOPE benchmark. $\pm$ is the standard deviation. We bold the top-2 scores for each metric. The results are tested on 3 random seeds. The results of the baselines are from [18]. Note that the rank correlation is "NAN" for HalfCheetah because the scores are not given in [18].

| TASK | HalfCheetah | | |
|---|---|---|---|
| METRIC | Absolute value gap | Rank correlation | Regret@1 |
| GALILEO | $1280 \pm 83$ | $\mathbf{0.313 \pm 0.09}$ | $\mathbf{0.12 \pm 0.02}$ |
| Best DICE | $1382 \pm 130$ | NAN | $0.82 \pm 0.29$ |
| VPM | $1374 \pm 153$ | NAN | $0.33 \pm 0.19$ |
| FQE (L2) | $\mathbf{1211 \pm 130}$ | NAN | $0.38 \pm 0.13$ |
| IS | $\mathbf{1217 \pm 123}$ | NAN | $\mathbf{0.05 \pm 0.05}$ |
| Doubly Rubost | $1222 \pm 134$ | NAN | $0.37 \pm 0.13$ |

| TASK | Walker2d | | |
|---|---|---|---|
| METRIC | Absolute value gap | Rank correlation | Regret@1 |
| GALILEO | $\mathbf{176 \pm 52}$ | $\mathbf{0.57 \pm 0.08}$ | $\mathbf{0.08 \pm 0.06}$ |
| Best DICE | $\mathbf{273 \pm 31}$ | $0.12 \pm 0.38$ | $0.27 \pm 0.43$ |
| VPM | $426 \pm 60$ | $\mathbf{0.44 \pm 0.21}$ | $\mathbf{0.08 \pm 0.06}$ |
| FQE (L2) | $350 \pm 79$ | $-0.09 \pm 0.36$ | $0.31 \pm 0.10$ |
| IS | $428 \pm 60$ | $-0.25 \pm 0.35$ | $0.70 \pm 0.39$ |
| Doubly Rubost | $368 \pm 74$ | $0.02 \pm 0.37$ | $0.25 \pm 0.09$ |

| TASK | Hopper | | |
|---|---|---|---|
| METRIC | Absolute value gap | Rank correlation | Regret@1 |
| GALILEO | $\mathbf{156 \pm 23}$ | $\mathbf{0.45 \pm 0.1}$ | $\mathbf{0.08 \pm 0.08}$ |
| Best DICE | $\mathbf{215 \pm 41}$ | $\mathbf{0.19 \pm 0.33}$ | $0.18 \pm 0.19$ |
| VPM | $433 \pm 44$ | $0.13 \pm 0.37$ | $\mathbf{0.10 \pm 0.14}$ |
| FQE (L2) | $283 \pm 73$ | $-0.29 \pm 0.33$ | $0.32 \pm 0.32$ |
| IS | $405 \pm 48$ | $-0.55 \pm 0.26$ | $0.32 \pm 0.32$ |
| Doubly Rubost | $307 \pm 73$ | $-0.31 \pm 0.34$ | $0.38 \pm 0.28$ |

## H.7 Detailed Results in the BAT Task

The core challenge of the environment model learning in BAT tasks is similar to the challenge in Fig. 1. Specifically, the behavior policy in BAT tasks is a human-expert policy, which will tend to increase the budget of allowance in the time periods with a lower supply of delivery clerks, otherwise will decrease the budget (Fig. 12 gives an instance of this phenomenon in the real data).

Since there is no oracle environment model for querying, we have to describe the results with other metrics.

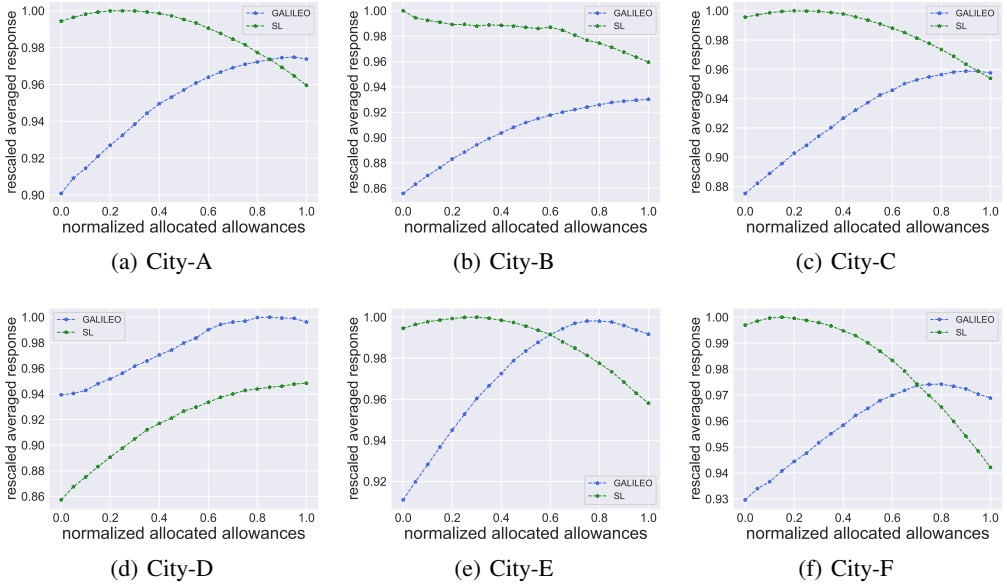

Figure 18: Illustration of the response curves in the 6 cities. Although the ground-truth curves are unknown, through human expert knowledge, *we know that it is expected to be monotonically increasing*.

First, we review whether the tendency of the response curve is consistent. In this application, with a larger budget of allowance, the supply will not be decreased. As can be seen in Fig. 18, the tendency of GALILEO's response is valid in 6 cities but almost all of the models of SL give opposite directions to the response. If we learn a policy through the model of SL, the optimal solution is canceling all of the allowances, which is obviously incorrect in practice.

Second, we conduct randomized controlled trials (RCT) in one of the testing cities. Using the RCT samples, we can evaluate the correctness of the sort order of the model predictions via Area Under the Uplift Curve (AUUC) [6]. To plot AUUC, we first sort the RCT samples based on the predicted treatment effects. Then the cumulative treatment effects are computed by scanning the sorted sample list. If the sort order of the model predictions is better, the sample with larger treatment effects will be computed early. Then

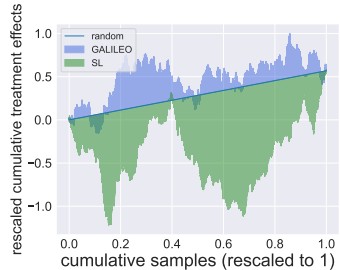

Figure 19: Illustration of the AUCC result for BAT. The model with larger areas above the "random" line makes better predictions in randomized-controlled-trials data [61].

the area of AUUC will be larger than the one via a random sorting strategy. The result of AUUC show GALILEO gives a reasonable sorting to the RCT samples (see Fig. 19).

Finally, we search for the optimal policy via the cross-entropy method planner [21] based on the learned model. We test the online supply improvement in 6 cities. The algorithm compared is a human-expert policy, which is also the behavior policy of the offline datasets. We conduct online

A/B tests for each of the cities. For each test, we randomly split a city into two partitions, one is for deploying the optimal policy learned from the GALILEO model, and the other is as a control group, which keeps the human-expert policy as before. Before the intervention, we collect 10 days' observation data and compute the averaged five-minute order-taken rates as the baselines of the treatment and control group, named $b^t$ and $b^c$ respectively. Then we start intervention and observe the five-minute order-taken rate in the following 14 days for the two groups. The results of the treatment and control groups are $y_i^t$ and $y_i^c$ respectively, where $i$ denotes the $i$-th day of the deployment. The percentage points of the supply improvement are computed via difference-in-difference (DID):

$$\frac{\sum_i^T (y_i^t - b^t) - (y_i^c - b^c)}{T} \times 100,$$

where $T$ is the total days of the intervention and $T = 14$ in our experiments.

Table 9: Results on BAT. We use City-X to denote the experiments on different cities. "pp" is an abbreviation of percentage points on the supply improvement.

| target city | City-A | City-B | City-C |
|---|---|---|---|
| supply improvement | +1.63pp | +0.79pp | +0.27pp |
| target city | City-D | City-E | City-F |
| supply improvement | +0.2pp | +0.14pp | +0.41pp |

The results are summarized in Tab. 9. The online experiment is conducted in 14 days and the results show that the policy learned with GALILEO can make better (the supply improvements are from **0.14 to 1.63** percentage points) budget allocation than the behavior policies in **all the testing cities**. We give detailed results which record the supply difference between the treatment group and the control group in Fig. 20.

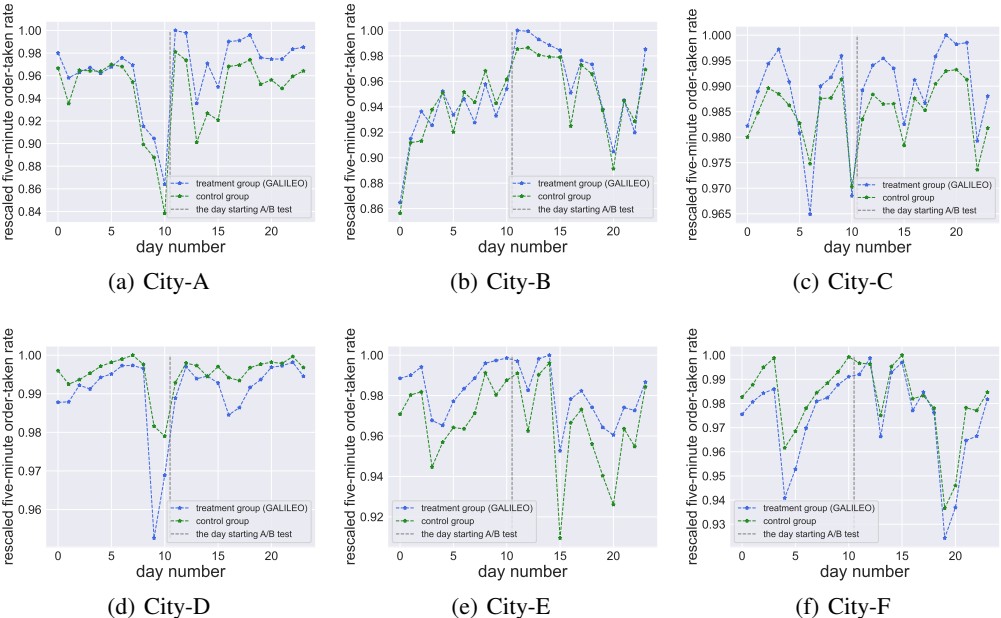

Figure 20: Illustration of the daily responses in the A/B test in the 6 cities.

Table 10: The root mean square errors on MuJoCo tasks. We bold the lowest error for each task. "medium" dataset **is used for training**, while "expert" and "medium-replay" datasets **are just used for testing**. $\pm$ follows the standard deviation of three seeds.

| TASK | HalfCheetah | | |
|---|---|---|---|
| DATASET | medium (train) | expert (test) | medium-replay (test) |
| GALILEO | $0.378 \pm 0.003$ | $\mathbf{2.287} \pm 0.005$ | $\mathbf{1.411} \pm 0.037$ |
| OFF-SL | $0.404 \pm 0.001$ | $3.311 \pm 0.055$ | $2.246 \pm 0.016$ |
| IPW | $0.513 \pm 0.033$ | $2.892 \pm 0.050$ | $2.058 \pm 0.021$ |
| SCIGAN | $\mathbf{0.309} \pm 0.002$ | $3.813 \pm 0.133$ | $2.484 \pm 0.040$ |

| TASK | Walker2d | | |
|---|---|---|---|
| DATASET | medium (train) | expert (test) | medium-replay (test) |
| GALILEO | $0.49 \pm 0.001$ | $\mathbf{1.514} \pm 0.002$ | $\mathbf{0.968} \pm 0.004$ |
| OFF-SL | $0.467 \pm 0.004$ | $1.825 \pm 0.061$ | $1.239 \pm 0.004$ |
| IPW | $0.564 \pm 0.001$ | $1.826 \pm 0.025$ | $1.282 \pm 0.007$ |
| SCIGAN | $\mathbf{0.438} \pm 0.001$ | $1.825 \pm 0.031$ | $1.196 \pm 0.005$ |

| TASK | Hopper | | |
|---|---|---|---|
| DATASET | medium (train) | expert (test) | medium-replay (test) |
| GALILEO | $0.037 \pm 0.002$ | $\mathbf{0.322} \pm 0.036$ | $\mathbf{0.408} \pm 0.003$ |
| OFF-SL | $\mathbf{0.034} \pm 0.001$ | $0.464 \pm 0.021$ | $0.574 \pm 0.008$ |
| IPW | $0.039 \pm 0.001$ | $0.533 \pm 0.00$ | $0.671 \pm 0.001$ |
| SCIGAN | $0.039 \pm 0.002$ | $0.628 \pm 0.050$ | $0.742 \pm 0.019$ |

Table 11: $\sqrt{MISE}$ results on GNFC. We bold the lowest error for each task. $\pm$ is the standard deviation of three random seeds.

| | e1_p1 | e0.2_p1 | e0.05_p1 |
|---|---|---|---|
| GALILEO | $5.17 \pm 0.06$ | $\mathbf{4.73} \pm \mathbf{0.13}$ | $\mathbf{4.70} \pm \mathbf{0.02}$ |
| SL | $\mathbf{5.15} \pm \mathbf{0.23}$ | $4.73 \pm 0.31$ | $23.64 \pm 4.86$ |
| IPW | $5.22 \pm 0.09$ | $5.50 \pm 0.01$ | $5.02 \pm 0.07$ |
| SCIGAN | $7.05 \pm 0.52$ | $6.58 \pm 0.58$ | $18.55 \pm 3.50$ |

| | e1_p0.2 | e0.2_p0.2 | e0.05_p0.2 |
|---|---|---|---|
| GALILEO | $\mathbf{5.03} \pm \mathbf{0.09}$ | $\mathbf{4.72} \pm \mathbf{0.05}$ | $\mathbf{4.87} \pm \mathbf{0.15}$ |
| SL | $5.21 \pm 0.63$ | $6.74 \pm 0.15$ | $33.52 \pm 1.32$ |
| IPW | $5.27 \pm 0.05$ | $5.69 \pm 0.00$ | $20.23 \pm 0.45$ |
| SCIGAN | $16.07 \pm 0.27$ | $12.07 \pm 1.93$ | $19.27 \pm 10.72$ |

| | e1_p0.05 | e0.2_p0.05 | e0.05_p0.05 |
|---|---|---|---|
| GALILEO | $\mathbf{5.23} \pm \mathbf{0.41}$ | $\mathbf{5.01} \pm \mathbf{0.08}$ | $\mathbf{6.17} \pm \mathbf{0.33}$ |
| SL | $5.89 \pm 0.88$ | $14.25 \pm 3.48$ | $37.50 \pm 2.29$ |
| IPW | $5.21 \pm 0.01$ | $5.52 \pm 0.44$ | $31.95 \pm 0.05$ |
| SCIGAN | $11.50 \pm 7.76$ | $13.05 \pm 4.19$ | $25.74 \pm 8.30$ |

Table 12: $\sqrt{MISE}$ results on TCGA. We bold the lowest error for each task. $\pm$ is the standard deviation of three random seeds.

| | t0_bias_2.0 | t0_bias_20.0 | t0_bias_50.0 |
|---|---|---|---|
| GALILEO | **0.34 ± 0.05** | **0.67 ± 0.13** | **2.04 ± 0.12** |
| SL | 0.38 ± 0.13 | 1.50 ± 0.31 | 3.06 ± 0.65 |
| IPW | 6.57 ± 1.16 | 6.88 ± 0.30 | 5.84 ± 0.71 |
| SCIGAN | 0.74 ± 0.05 | 2.74 ± 0.35 | 3.19 ± 0.09 |
| | t1_bias_2.0 | t1_bias_6.0 | t1_bias_8.0 |
| GALILEO | **0.43 ± 0.05** | **0.25 ± 0.02** | **0.21 ± 0.04** |
| SL | 0.47 ± 0.05 | 1.33 ± 0.97 | 1.18 ± 0.73 |
| IPW | 3.67 ± 2.37 | 0.54 ± 0.13 | 2.69 ± 1.17 |
| SCIGAN | 0.45 ± 0.25 | 1.08 ± 1.04 | 1.01 ± 0.77 |
| | t2_bias_2.0 | t2_bias_6.0 | t2_bias_8.0 |
| GALILEO | 1.46 ± 0.09 | **0.85 ± 0.04** | **0.46 ± 0.01** |
| SL | 0.81 ± 0.14 | 3.74 ± 2.04 | 3.59 ± 0.14 |
| IPW | 2.94 ± 1.59 | 1.24 ± 0.01 | 0.99 ± 0.06 |
| SCIGAN | **0.73 ± 0.15** | 1.20 ± 0.53 | 2.13 ± 1.75 |

Table 13: $\sqrt{MMSE}$ results on GNFC. We bold the lowest error for each task. $\pm$ is the standard deviation of three random seeds.

| | e1_p1 | e0.2_p1 | e0.05_p1 |
|---|---|---|---|
| GALILEO | **3.86 ± 0.03** | **3.99 ± 0.01** | **4.07 ± 0.03** |
| SL | 5.73 ± 0.33 | 5.80 ± 0.28 | 18.78 ± 3.13 |
| IPW | 4.02 ± 0.05 | 4.15 ± 0.12 | 22.66 ± 0.33 |
| SCIGAN | 8.84 ± 0.54 | 12.62 ± 2.17 | 24.21 ± 5.20 |
| | e1_p0.2 | e0.2_p0.2 | e0.05_p0.2 |
| GALILEO | **4.13 ± 0.10** | **4.11 ± 0.15** | **4.21 ± 0.15** |
| SL | 5.87 ± 0.43 | 7.44 ± 1.13 | 29.13 ± 3.44 |
| IPW | 4.12 ± 0.02 | 6.12 ± 0.48 | 30.96 ± 0.17 |
| SCIGAN | 12.87 ± 3.02 | 14.59 ± 2.13 | 24.57 ± 3.00 |
| | e1_p0.05 | e0.2_p0.05 | e0.05_p0.05 |
| GALILEO | **4.39 ± 0.20** | **4.34 ± 0.20** | **5.26 ± 0.29** |
| SL | 6.12 ± 0.43 | 14.88 ± 4.41 | 30.81 ± 1.69 |
| IPW | 13.60 ± 7.83 | 26.27 ± 2.67 | 32.55 ± 0.12 |
| SCIGAN | 9.19 ± 1.04 | 15.08 ± 1.26 | 17.52 ± 0.02 |

Table 14: $\sqrt{MMSE}$ results on TCGA. We bold the lowest error for each task. $\pm$ is the standard deviation of three random seeds.

|  | t0_bias_2.0 | t0_bias_20.0 | t0_bias_50.0 |
| --- | --- | --- | --- |
| GALILEO | $\mathbf{1.56 \pm 0.04}$ | $\mathbf{1.96 \pm 0.53}$ | $\mathbf{3.16 \pm 0.13}$ |
| SL | $1.92 \pm 0.67$ | $2.31 \pm 0.19$ | $5.11 \pm 0.66$ |
| IPW | $7.42 \pm 0.46$ | $5.36 \pm 0.96$ | $5.38 \pm 1.24$ |
| SCIGAN | $2.11 \pm 0.47$ | $5.23 \pm 0.27$ | $5.59 \pm 1.02$ |
|  | t1_bias_2.0 | t1_bias_6.0 | t1_bias_8.0 |
| GALILEO | $1.43 \pm 0.06$ | $1.09 \pm 0.05$ | $\mathbf{1.36 \pm 0.36}$ |
| SL | $\mathbf{1.12 \pm 0.15}$ | $3.65 \pm 1.91$ | $3.96 \pm 1.81$ |
| IPW | $1.14 \pm 0.11$ | $\mathbf{0.90 \pm 0.09}$ | $2.04 \pm 0.99$ |
| SCIGAN | $3.32 \pm 0.88$ | $4.74 \pm 2.12$ | $5.17 \pm 2.42$ |
|  | t2_bias_2.0 | t2_bias_6.0 | t2_bias_8.0 |
| GALILEO | $3.77 \pm 0.35$ | $3.99 \pm 0.40$ | $\mathbf{2.08 \pm 0.60}$ |
| SL | $\mathbf{2.70 \pm 0.67}$ | $8.33 \pm 5.05$ | $9.70 \pm 3.12$ |
| IPW | $2.92 \pm 0.15$ | $3.90 \pm 0.17$ | $4.47 \pm 2.16$ |
| SCIGAN | $3.82 \pm 2.12$ | $\mathbf{1.83 \pm 1.49}$ | $3.62 \pm 4.9$ |

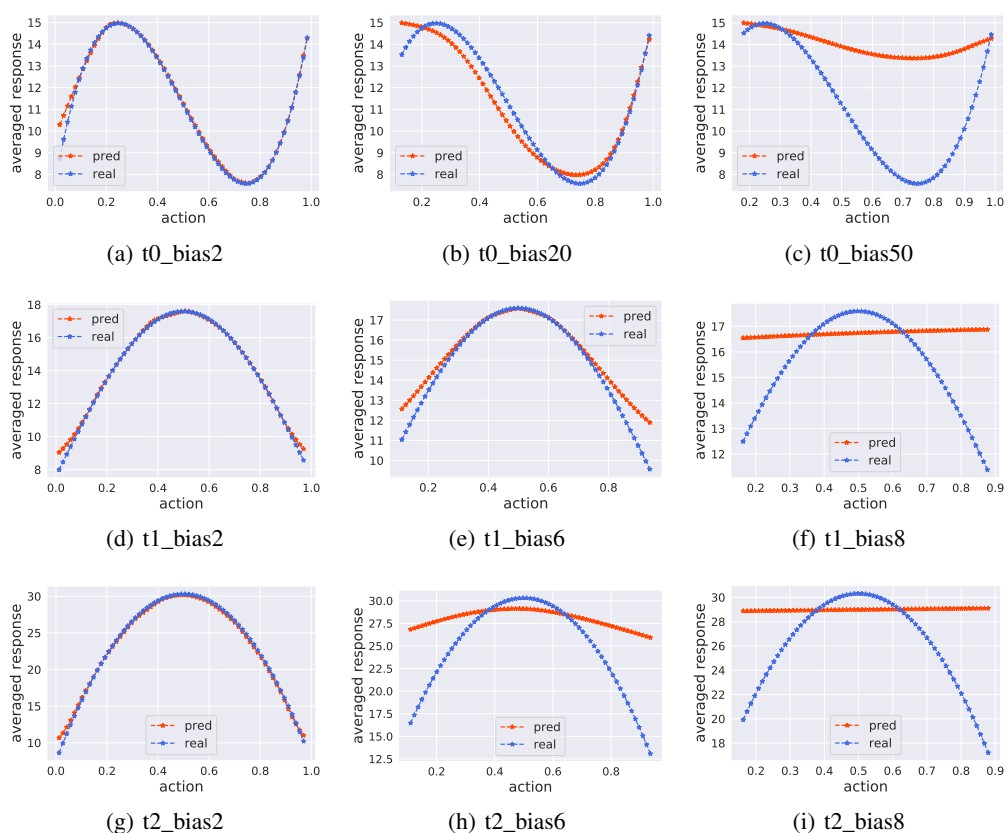

(a) t0_bias2     (b) t0_bias20     (c) t0_bias50

(d) t1_bias2     (e) t1_bias6     (f) t1_bias8

(g) t2_bias2     (h) t2_bias6     (i) t2_bias8

Figure 21: Illustration of the averaged response curves of Supervised Learning (SL) in TCGA.

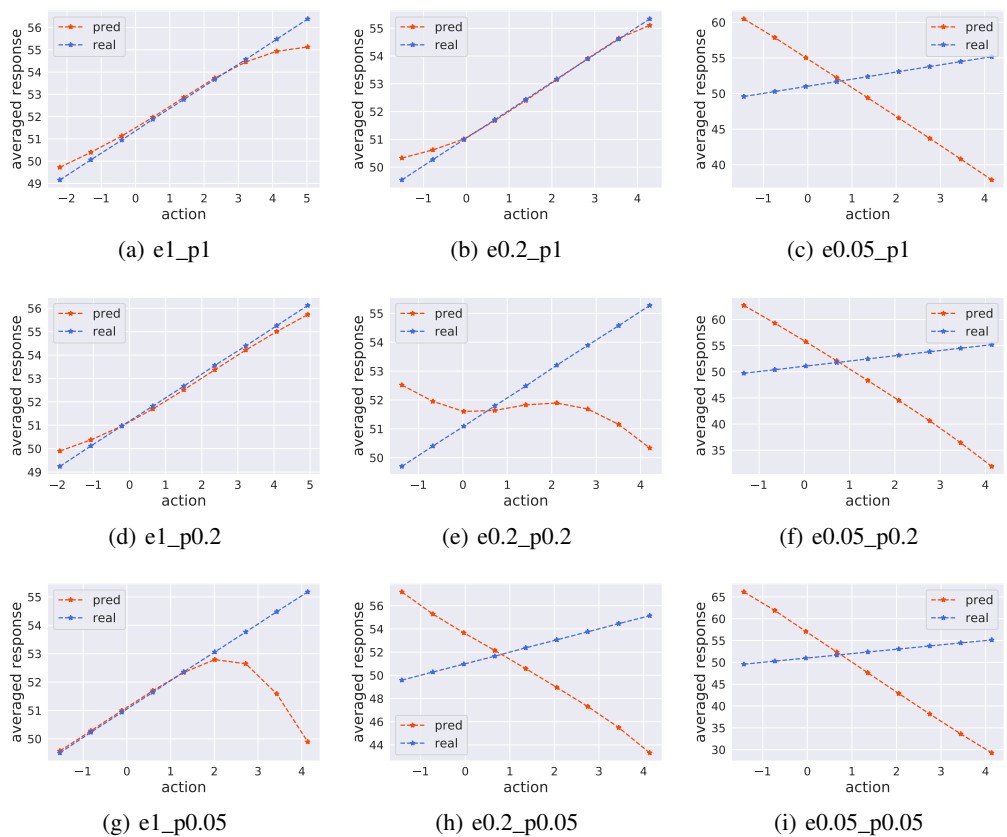

Figure 22: Illustration of the averaged response curves of Supervised Learning (SL) in GNFC.

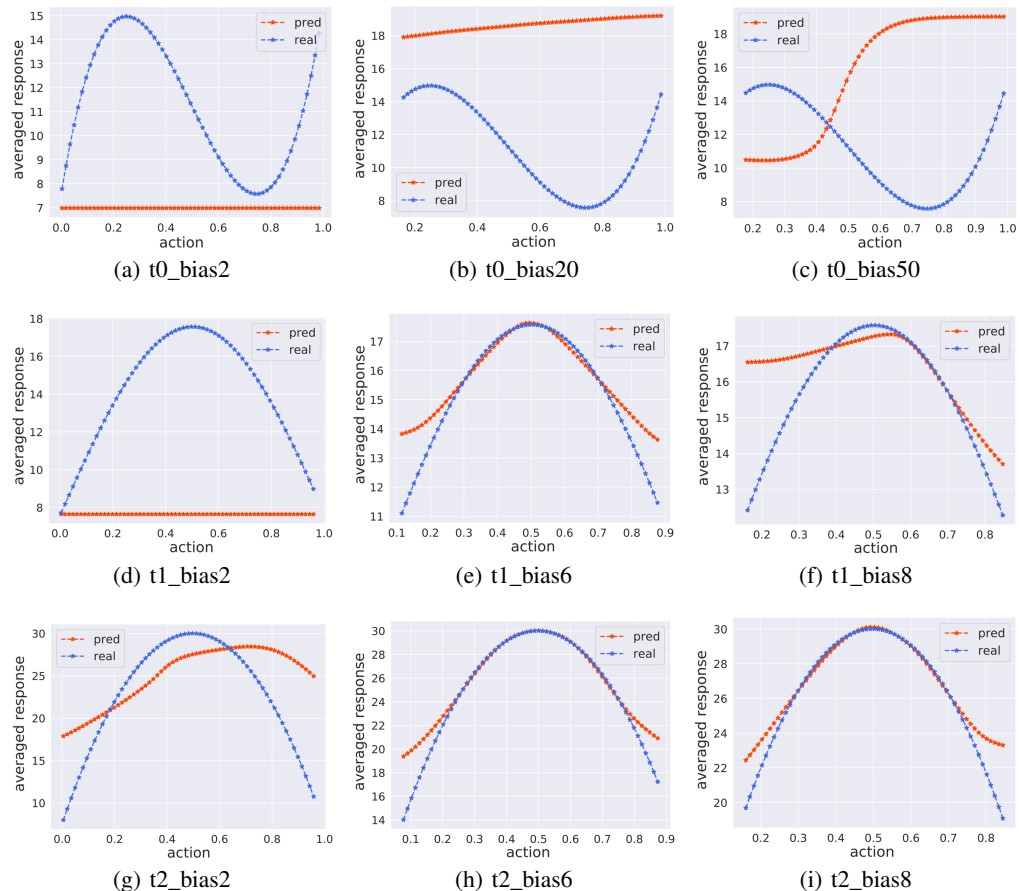

Figure 23: Illustration of the averaged response curves of Inverse Propensity Weighting (IPW) in TCGA.

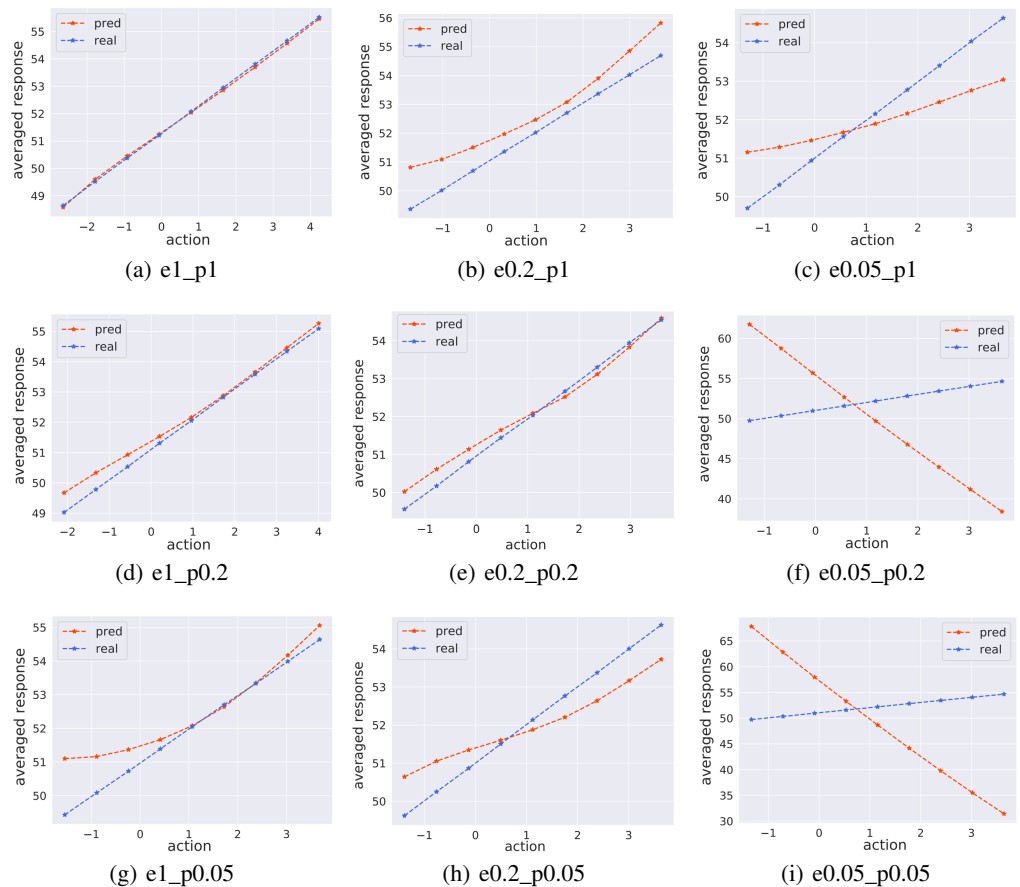

Figure 24: Illustration of the averaged response curves of Inverse Propensity Weighting (IPW) in GNFC.

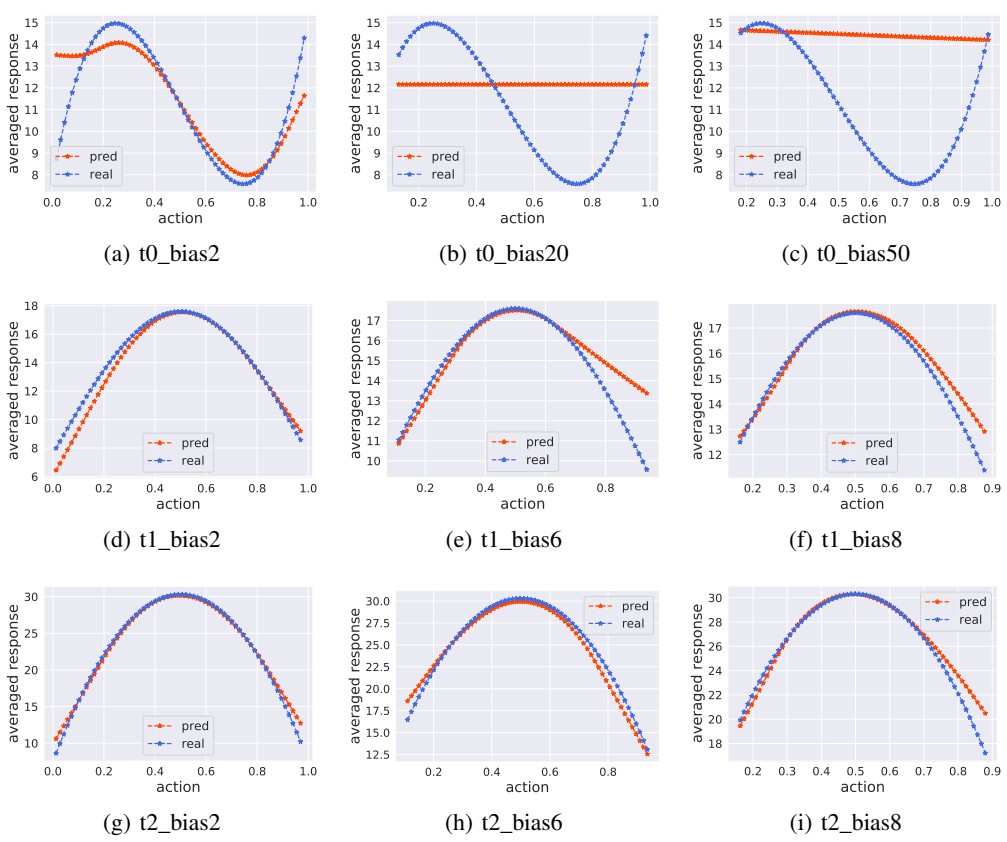

Figure 25: Illustration of the averaged response curves of SCIGAN in TCGA.

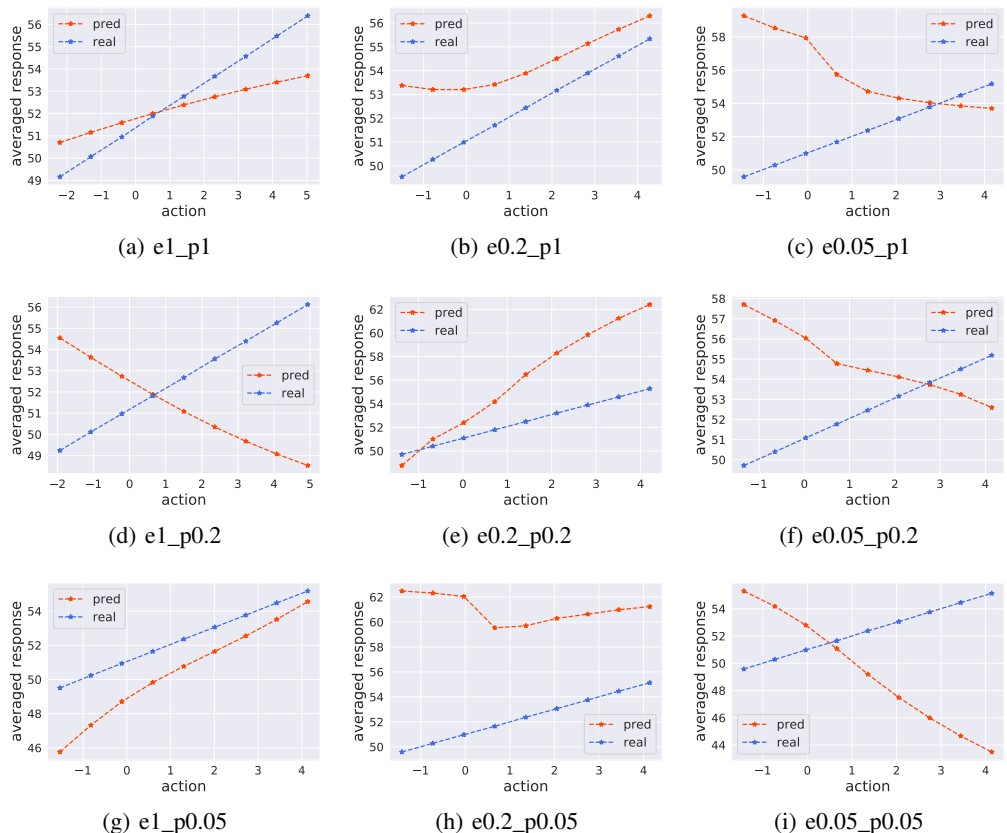

Figure 26: Illustration of the averaged response curves of SCIGAN in GNFC.

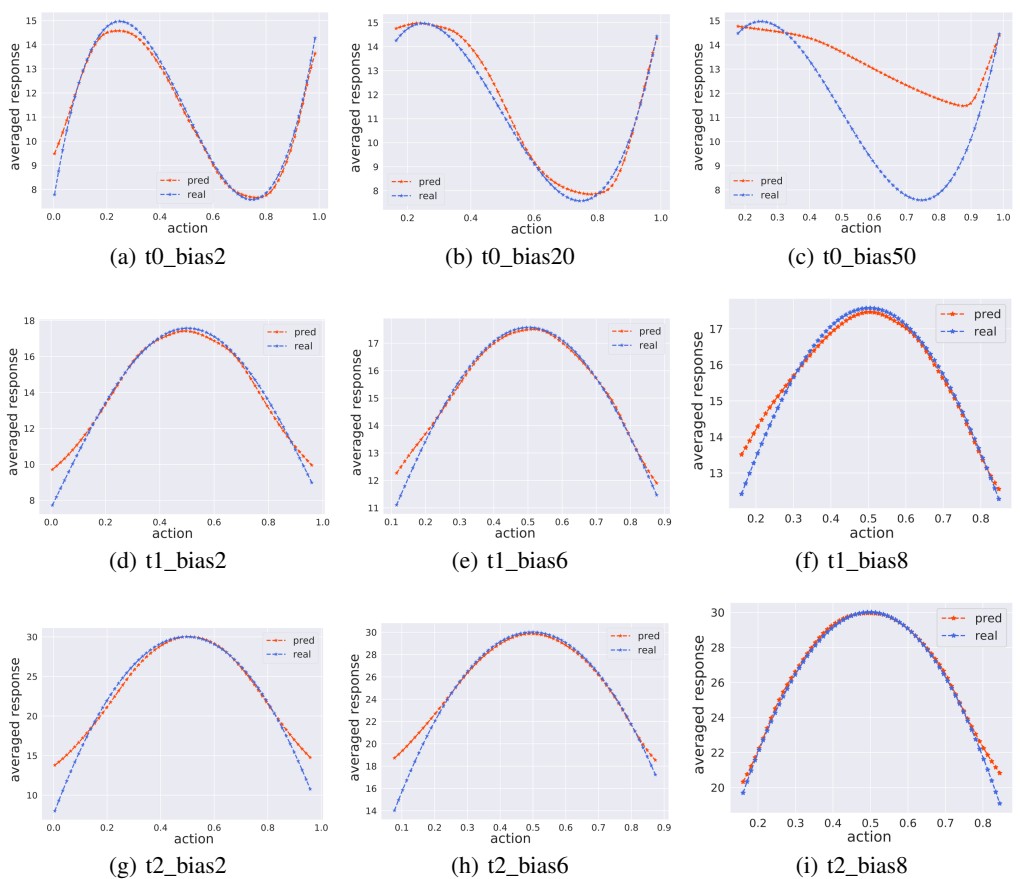

Figure 27: Illustration of the averaged response curves of GALILEO in TCGA.

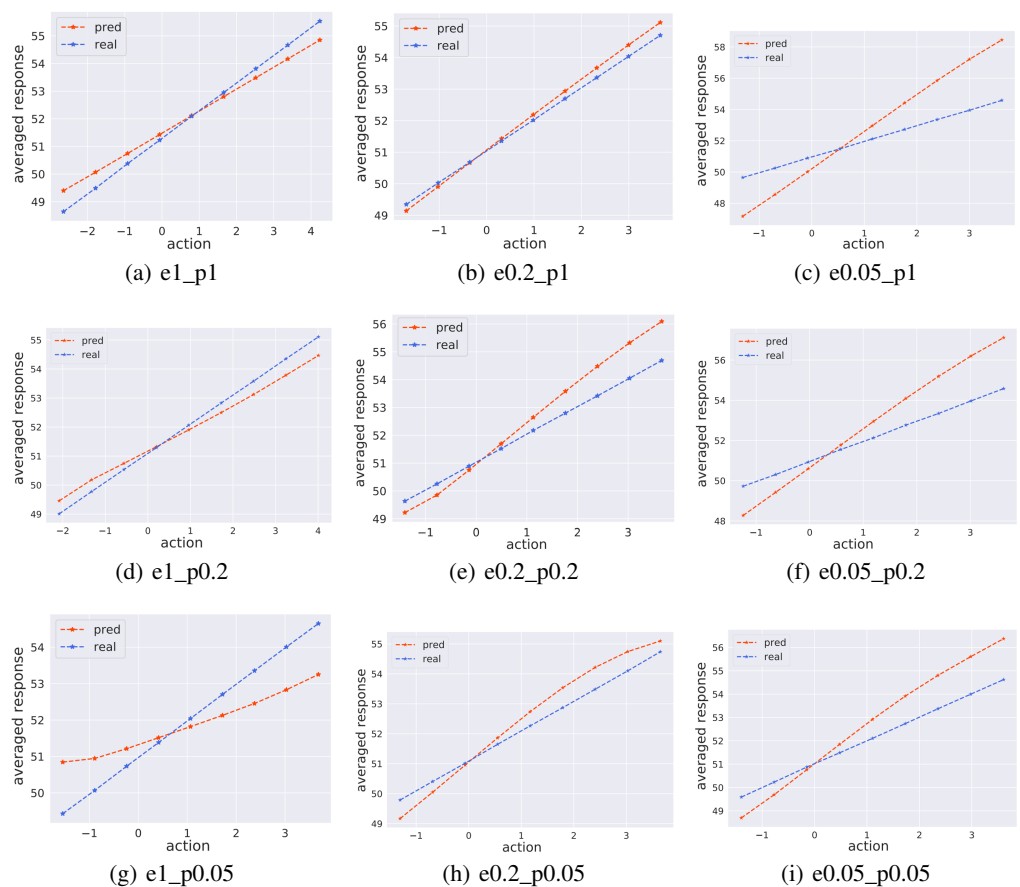

Figure 28: Illustration of the averaged response curves of GALILEO in GNFC.

