# OpenReview forum: "Adversarial Counterfactual Environment Model Learning"
_NeurIPS.cc/2023/Conference — NeurIPS 2023 spotlight_

### Official Review · Reviewer_qQGz · 2023-07-03

**Soundness:** 3 good
**Presentation:** 3 good
**Contribution:** 3 good
**Rating:** 8
**Confidence:** 3

**Summary:**

An accurate environment dynamics model is crucial for various downstream tasks, such as counterfactual prediction, off-policy evaluation, and offline reinforcement learning. Currently, these models were learned through empirical risk minimization by step-wise fitting of historical transition data. However, we first show that, particularly in the sequential decision-making setting, this approach may catastrophically fail to predict counterfactual action effects due to the selection bias of behavior policies during data collection. To tackle this problem, the authors introduce a novel model-learning objective called adversarial weighted empirical risk minimization (AWRM). AWRM incorporates an adversarial policy that exploits the model to generate a data distribution that weakens the model's prediction accuracy, and subsequently, the model is learned under this adversarial data distribution. They implement a practical algorithm, GALILEO, for AWRM and evaluate it on two synthetic tasks, three continuous-control tasks, and a real-world application. The experiments demonstrate that GALILEO can accurately predict counterfactual actions and improve various downstream tasks, including offline policy evaluation and improvement, as well as online decision-making.

**Strengths:**

1. This paper addresses the problem of accurate environment dynamics model learning, which exhibits wide impacts in many downstream tasks, like counterfactual prediction, off-policy evaluation, and offline reinforcement learning. This is a very important and meaningful research topic.

2. This is the first research on faithful dynamics model learning in sequential decision-making settings like RL, which demonstrates the novelty of this paper.

3. The analysis to the challenges brought by the conventional empirical risk minimization method is deep and insightful. Especially, the authors use a vivid example to illustrate them. Based on this, the transition to the propose of adversarial weighted empirical risk minimization objective is smooth, which strongly supports the following the Generative Adversarial Offline Counterfactual environment model learning (GALILEO) method.

3. The experiments of this work is sufficient and persuasive. The authors conduct experiments in two synthetic tasks, three continuous-control tasks, and a real-world application. They first verify that GALILEO can make accurate predictions on counterfactual data queried by other policies. Then, they demonstrate that the model learned by GALILEO is helpful to several downstream tasks.

**Weaknesses:**

1. It can be better if the authors can add several baselines to better validate the superiority of the proposed GALILEO method.

2. The description of the method part is deep and comprehensive. But the authors can consider making it easier to understand if possible.

3. Some typos need to be fixed in the future version, like the subtitle of Section. 5.3.

**Questions:**

1. I am little bit curious to the details of metrics and downstream tasks used in the experiments. For example, the AUUC, value gap, regret@1, and rank correlation metrics and the details behind the off-policy evaluation experiments. Could you please consider adding some corresponding explanations in the future version?

**Limitations:**

See the Weaknesses and Questions above.

---

> ### Author Rebuttal · Authors · 2023-08-10
>
> We express our gratitude for your constructive feedback and the time you dedicated to evaluating our paper. Your insightful remarks help us in refining our work and emphasizing its importance in the reinforcement learning domain.
>
> **Clarify Experiments and Metrics:**
>
> We understand the concerns regarding the metrics and downstream tasks utilized in our experiments. While the main body of the paper had constraints regarding space, we have provided a comprehensive explanation of these metrics and tasks in Appendix H.6.1 and Appendix H.7. To make this information more accessible, we will emphasize this section prominently in the main text and provide a direct link ensuring that readers can easily navigate to it.
>
> **Baselines Justifications:**
>
> Thank you for raising the point about the selection of baselines. Our choices were deliberate and comprehensive to verify the algorithm:
>
> - **SL**: A representation of a standard algorithm for ERM, which is common-used for model learning in current offline model-based RL algorithms.
> - **IPW**: Chosen for its standing as a benchmark in counterfactual modeling for WERM through IPS, it helps elucidate how AWRM objective compares and advances beyond existing techniques.
> - **SCIGAN**: SCIGAN is perceived as subsets of GALILEO for optimizing AWRM, using SCIGAN provides a more distinct reference to assess the effectiveness of our approach. It is also a good baseline since its rigorous testing on TCGA benchmarks [1].
>
> In response, we will provide a more detailed explanation of our baseline choices at the beginning of Section 5 to ensure absolute clarity for our readers.
>
> We believe the baseline selection is representative and comprehensive, and we hope the above clarification can solve your concerns. We commit to introduce GAIL as an additional baseline. It's important to note, however, that the conventional GAIL method focuses on policy learning rather than environment model learning. Given the inherent instability of GAIL-style algorithms, we cannot guarantee optimal results within the constrained timeframe of the rebuttal and discussion period.
>
> **Method Description:**
>
>  We genuinely appreciate your feedback regarding the method's description. Our intention was to provide an exhaustive understanding, which, in hindsight, might have resulted in a dense exposition. In light of your observations, we are inclined to simplify certain sections for better clarity.
>
>
>
> In conclusion, we sincerely thank you for helping us improve our paper's quality. We believe that addressing these points will make the paper more accessible and insightful for the community. We are optimistic that these changes will elevate our work and make it a valuable contribution to the field.

---

> > ### Comment · Reviewer_qQGz · 2023-08-21
> >
> > Thanks for your explanations. I suggest you to mention these contents in the main text to help readers better understand this paper. My previous concerns have all been addressed.

---

### Official Review · Reviewer_wiCk · 2023-07-07

**Soundness:** 3 good
**Presentation:** 3 good
**Contribution:** 3 good
**Rating:** 6
**Confidence:** 2

**Summary:**

The paper presents a novel method for improving the accuracy of environment dynamics models for counterfactual prediction, off-policy evaluation, and offline reinforcement learning. Currently, these models learn via empirical risk minimization (ERM), which the authors show can lead to failures in counterfactual action prediction due to selection bias during data collection. To address this, the authors introduce adversarial weighted empirical risk minimization (AWRM), where an adversarial policy weakens the model's prediction accuracy to encourage improvement. They implement this approach via an algorithm named GALILEO, which is evaluated on synthetic tasks, continuous-control tasks, and a real-world application. Results show that GALILEO can accurately predict counterfactual actions and improve several downstream tasks.

**Strengths:**

1. The concept of an adversarial weighted empirical risk minimization (AWRM) is a novel idea that brings together ideas from adversarial training and reinforcement learning. The use of an adversarial policy to manipulate the model's data distribution and improve its learning process is an innovative approach.
2. The paper provides a theoretical foundation for AWRM and shows its implementation through the GALILEO algorithm. The authors also conduct a variety of tests to assess GALILEO's performance, including synthetic tasks, continuous-control tasks, and a real-world delivery platform.
3. The paper is generally well-written and the use of figures to illustrate key concepts also enhances the clarity of the paper.

**Weaknesses:**

1. The authors didn't cover the preliminaries and related works adequately. I'm not too familiar with counterfactual modelling techniques, and think the authors didn't present enough to situate their work.
2. Justifications for choosing the three baselines are missing. Why not choose the more recent GAIL based methods?
3. Typo: line 320 Downstream
4. Formatting issue: The upper margins in page 2, 7, 8 and 9 seem too small.

**Questions:**

Please see above. Could you please better situate your work and justify the choice of baseline methods?

**Limitations:**

The authors didn't address the limitations nor broader societal impacts in the paper. But I didn't see any ethical concerns.

---

> ### Author Rebuttal · Authors · 2023-08-10
>
> We'd like to genuinely express our gratitude for your thoughtful feedback and the time you've invested in reviewing our work. We've taken your concerns to heart and have attempted to address them as follows.
>
> **1. Contextualizing the Work:**
>
> We  acknowledge the feedback on the need for a more detailed presentation of the preliminaries and related works. While we did provide information on counterfactual modeling techniques in Appendix F, including previous techniques based on WERM through IPS, and structural causal models for counterfactual inference.
>
> We understand that our inclusion of counterfactual modeling techniques in Appendix F might not have been sufficiently prominent. To address this, we commit to elaborating on these concepts within the main sections of the paper (potentially in Section 3) to ensure a more comprehensive and upfront presentation for all readers.
>
> **2. Baseline Justifications:**
>
> Thanks for the comment. Our choice of baselines were indeed deliberate:
>
> - **SL**: A representation of a standard algorithm for ERM, which is common-used for model learning in current offline model-based RL algorithms.
> - **IPW**: Chosen for its standing as a benchmark in counterfactual modeling for WERM through IPS, it helps elucidate how AWRM objective compares and advances beyond existing techniques.
> - **SCIGAN**:  An adversarial model learning technique that can be regarded as a partial implementation of GALILEO for optimizing AWRM. While both GAIL and SCIGAN can be seen as partial implementations of GALILEO, the results from SCIGAN gave us a clearer benchmark against which we could measure our algorithm's success.
>
> We opted for SCIGAN over GAIL because it is well-tuned in [1] and has been tested on TCGA benchmarks. We appreciate the point raised about the potential inclusion of GAIL-based methods. We will further clarify our choice of baselines in the paper, ensuring readers understand the rationale behind our selections in the start of Section 5.
>
> We believe the baseline selection is representative and comprehensive, and we hope the above clarification can solve your concerns. We commit to introduce GAIL as an additional baseline. It's important to note, however, that the conventional GAIL method focuses on policy learning rather than environment model learning. Given the inherent instability of GAIL-style algorithms, we cannot guarantee optimal results within the constrained timeframe of the rebuttal and discussion period.
>
> **Technical Corrections**
>
> We acknowledge the typographical error and formatting issue. We assure you that these will be rectified in our revised submission, and we'll double-check the document to preempt any similar issues.
>
>
>
> In wrapping up, we reiterate our gratitude for your invaluable feedback. We believe these insights will undeniably enhance our paper's clarity and depth. Should you have further queries or need additional clarifications on our revisions, we are open to discussions and more than willing to engage.
>
> [1] Estimating the effects of continuous-valued interventions using generative adversarial networks.

---

> > ### Comment · Reviewer_wiCk · 2023-08-15
> >
> > Thanks for the response. Most of my concerns are addressed. I've modified my rating accordingly. Cheers.

---

> > > ### Author Response · Authors · 2023-08-18
> > >
> > > We greatly appreciate your constructive feedback and are glad to hear that our response addressed most of your concerns. Thank you for your time and consideration.

---

### Official Review · Reviewer_h4bm · 2023-07-07

**Soundness:** 3 good
**Presentation:** 3 good
**Contribution:** 3 good
**Rating:** 6
**Confidence:** 3

**Summary:**

This paper considers the problem of environment modeling. An adversarial method is proposed that the adversarial counterparts is trained to exploit the model to generate a data distribution that weakens the model’s prediction accuracy, and then the model is trained under the adversarial data distribution with a weighted empirical risk minimization objective. Experiments are conducted on synthetic, control and real-world tasks.

**Strengths:**

The paper is clearly written. The illustration examples are clear and well motivate the problem.
Applying IPS and the surrogate optimization step is interesting.
I also agree that conservative or pessimistic offline model-based RL methods often try to limit policy exploration, which might make it hard to obtain an accurate environment model.


**Weaknesses:**

Complexity and stability of the adversarial method might be concerned, where two discriminators should be learned.


**Questions:**

1. It is known that training GAN-like methods suffer from instability and often the hyperparameters should be carefully tuned. The proposed method involving training two discriminators and I am worried about the stability of the optimization step in Eq. (6) if it is easy to rise collapse problem.

2. It is mentioned that in HalfCheetah, all policies just keep the cheetah standing. I think similar problems might also exist for other tasks, while they are not shown up since reward curves or numbers are compared more often without accessing how well the policies really do in these tasks. Despite this, could any other intuitive information on the performance of the environment modeling be conducted considering a specific task, instead of focusing on the rewards only? For example, in halfcheetah, which transitions are more likely to have a larger discrepancy?

Minor:

Title of Section 5.3, ‘Ddownstream’ -> ‘downstream’



**Limitations:**

Yes.

---

> ### Author Rebuttal · Authors · 2023-08-10
>
> We would like to express our gratitude for your thoughtful feedback on our submission.  We've taken your concerns to heart and have attempted to address them in the following manner.
>
> **Addressing Stability Concerns of GALILEO:**
>
> 1. **Instability of training two discriminators:** The reviewer are right in observing that the GALILEO, much like other GAN-style algorithms, can be challenging to tune. Primarily, the challenge lies in striking a balance between the learning rate of the discriminator and the generator, a known issue in the GAN framework. However, we found that the two discriminators setup did not introduce further complications in this respect. In our practice, we just ensure the uniformity between the two discriminators regarding learning rates, frequency, and network structure, then the GALILEO algorithm can learn the model stably.
> 2. **Ablation and Implementation Techniques:** We appreciate your concerns about the stability and effectiveness of the introduced techniques. We have discussed crucial techniques for the GALILEO implementation in Appendix E. Further, ablation studies centered on these techniques can be found in Appendix H.3.
> 3. **Code Open-sourcing:** To foster transparency and facilitate reproducibility, we will open-source our code, which would share exhaustive details on the algorithm.
> 4. **Add to Limitation:** We agree that the instability is an important issue of GAN-style algorithms, we will mention this limitation in Section 6. This is also in our future work to investigate.
>
> **Algorithm Performance Beyond Reward Metrics**
>
> In light of your comments, we decided to provide a more comprehensive view of GALILEO's efficacy by delving into its sample efficiency and policy behavior for downstream tasks. We will supplemented our results with trajectory visualizations for asymptotic policies obtained from various models. Interestingly, we found that policies trained via IPW and SL either remain standing or fall backward in all three environments, whereas those from GALILEO and SCIGAN demonstrate forward movement in Walker2d and Hopper, and keep standing in HalfCheetah.
>
> In the revised version, we will add these visualization results to Appendix H.
>
> **Minor Corrections:**
>
> We acknowledge the typographical error highlighted by you in Section 5.3 and will rectify it in the final version of the paper.
>
>
>
> In conclusion, we appreciate the time and effort you've invested in reviewing our work. We've endeavored to address the concerns raised, and we hope our explanations and additions provide clarity. We're optimistic that the enhancements and clarifications enhance the overall quality and impact of our contribution.

---

### Official Review · Reviewer_So7i · 2023-07-08

**Soundness:** 3 good
**Presentation:** 2 fair
**Contribution:** 2 fair
**Rating:** 5
**Confidence:** 2

**Summary:**

The paper proposes a model-learning approach for counterfactual prediction (CP), off-policy evaluation (OPE), and offline reinforcement learning (ORL). The authors introduce the adversarial weighted empirical risk minimization (AWRM) objective to facilitate learning models that accurately evaluate target policies. Additionally, they present the GALILEO algorithm, a generative adversarial training method that approximates the data distribution induced by the optimal adversarial policy.


**Strengths:**

+ The paper effectively addresses the problem of learning accurate models for CP, OPE, and ORL, which is particularly significant in domains with costly data collection.
+ The authors provide a comprehensive discussion on the impact of selection bias on CP, which serves as motivation for their objective, AWRM, and the GALILEO algorithm.

**Weaknesses:**

- However, it is unclear how novel is the weighted version of empirical risk minimization (ERM), compared to prior research. The main contribution lies in the adversarial aspect. Therefore, I recommend revising the introduction to emphasize and motivate the adversarial part.

- The derivations appear reasonable. However, including a brief discussion about Algorithm 1 could enhance the paper's readability.

- While the experimental results are generally ok, one notable limitation is that the authors only consider three tasks from the D4RL and DOPE benchmarks. In addition, the proposed approach does not outperform other methods in one of these tasks (HalfCheetah). To convincingly demonstrate the efficacy of their algorithm, the paper should include more tasks in their evaluation. Additionally, the constraint of limiting the time horizon to {10, 20, 40} is very strong and lacks proper motivation.

- Furthermore, the proposed algorithm lacks comparison with state-of-the-art algorithms for the D4RL benchmark, such as the one mentioned in https://openreview.net/pdf?id=VYYf6S67pQc.

**Questions:**

In Table 1, it is stated that the returns are normalized to lie between 0 and 100; however, there are negative values in the HalfCheetah column. Is this a typo?

**Limitations:**

yes

---

> ### Author Rebuttal · Authors · 2023-08-10
>
> Thank you for the time and effort you have put into reviewing our paper. We appreciate your feedback and would like to address the concerns you've raised as follows:
>
> 1. **Experimental Evaluation**:
>
>    1. *Limiting the Time Horizon to {10, 20, 40}*:
>
>       As detailed in Line 328, our motivation for this choice,  was to *rigorously* examine the models' capabilities. By excluding certain tricks that constrain policy exploration and risky region learning which are commonly done in offline model-based RL algorithms like MOPO, we can fully exploit the learned models using standard RL algorithm. However, we observed a large compounding error in the 1,000-step rollout in this setting, making all algorithms fails in learning a reasonable policy. To better verify the effects of models  on policy improvement, we opted for smaller horizons of {10, 20, 40}. We found that after learning in 40-horizon setting, there are significant gaps between the policy learned in real environment and dynamics model, thus we do not further scale up the horizons.
>
>    2. *Choice of Tasks from the D4RL and DOPE Benchmarks. To convincingly demonstrate the efficacy of their algorithm, the paper should include more tasks in their evaluation.*
>
>       As mentioned in Line 263, we opted for the three `medium` datasets in D4RL and DOPE explicitly because these are the only datasets within the benchmark exhibiting selection bias. Other datasets in the benchmark were amassed without an evident selection bias due to the utilization of mixed policies during data collection.
>
>       In addition, we have also incorporated two synthetic benchmarks comprising 18 tasks and a real-world application to demonstrate the effectiveness of our proposed method. **We believe these comprehensive experiments significantly validate our approach, and we hope they aren’t overlooked.**
>
>    3. *Performance on the HalfCheetah Task*:
>
>       As elucidated in Lines 334 to 341, we found certain challenges unique to the HalfCheetah dataset that led to the observed results. All the policies derived, including ours, had the cheetah either standing stationary or moving backward, which means that all policies actually fail in complete the task, even though IPW reaches a bit better performance.
>
>    4. *Comparison with State-of-the-Art Algorithms:*
>
>       We recognize the importance of benchmarking against state-of-the-art methods. Due to the distinct settings in our experiments using D4RL, direct comparisons are challenging. However, we acknowledge the related work highlighted by the reviewer and commit to incorporating it into Appendix F.
>
>    We understand the author's concerns about missing evaluation in standard offlineRL benchmarks. **We would like to emphasize that our experiment is to evaluate the effects of selection bias, and the ability of learned models' predictions in counterfactual actions, instead of purely pursuing the SOTA policy performance in offlineRL benchmarks.** We will revised the paper in the experiment section to clarify this point.
>
> 2. **Concern with Table 1**:  In Table 1, only the column `avg. norm.` uses the normalized return, all other scores is the raw returns.
>
> 3. **Regarding the Novelty of the WERM and AWRM**: Thanks for the suggestion. We agree that there may have been some confusion regarding the emphasis on the weighted version of ERM (WERM). Our primary contribution is the adversarial objective AWRM. The introduction of the weighted version of ERM served mainly as a scaffold to lead into the discussion of AWRM, ensuring a smoother narrative flow.
>
>    We will revise the introduction section to stress the importance and novelty of the adversarial aspect of our contribution.
>
> 4. **Discussion about Algorithm 1 could enhance the paper's readability:** Thanks for the valuable suggestion, we will revise the last paragraph of Section 4.3 to improve the readability of the implementation  of algorithm 1.
>
> In closing, we are grateful for your insights and will undertake necessary revisions to address the mentioned concerns. We believe that our contributions offer a valuable perspective in the domain of reinforcement learning and hope that the clarifications provided here assuage any reservations.

---

> > ### Comment · Reviewer_So7i · 2023-08-14
> >
> > Thank you for the comments.
> > Some of my concerns have been addressed and I have increased my score.

---

> > > ### Author Response · Authors · 2023-08-15
> > >
> > > We're grateful for your insightful feedback on our work. We will revise the paper following your suggestions.

---

### Official Review · Reviewer_E1XE · 2023-07-24

**Soundness:** 4 excellent
**Presentation:** 3 good
**Contribution:** 4 excellent
**Rating:** 8
**Confidence:** 5

**Summary:**

This paper introduces an adversarial training approach to model learning that improves performance for counterfactual data that may differ widely from the data used to train the model. This is particularly relevant when training from offline data and expecting the model to generalize when deployed later. This paper extensively presents an adversarially weighted empirical risk minimization objective drawing inspiration from inverse propensity scoring/weighting. A practical algorithm, GALILEO, is introduced and extensively tested across synthetic, continuous control, and real-world data. The paper clearly lays out the advantages of unbiased and accurate counterfactual models in a wide array of use cases in RL. The authors do this to contrast to major limitations of a majority of model-learning approaches that use supervised learning to perform empirical risk minimization.

This is a complete and well written paper. The development of the proposed method are clearly justified with sufficient grounding in the formal exposition of the equations. The derivation of AWRM was easy to follow with the structure put in place by the authors.

The included experiments clearly lay out the intended contribution of the proposed approach, that GALILEO provides a more accurate and counterfactually correct model. Impressively, this improved model is shown to have clear benefits for downstream performance in tasks beyond the “pre-training” task.

**Strengths:**

The paper clearly lays out the advantages of unbiased and accurate counterfactual models in a wide array of use cases in RL. The authors do this to contrast to major limitations of a majority of model-learning approaches that use supervised learning to perform empirical risk minimization.

This is a complete and well written paper. The development of the proposed method are clearly justified with sufficient grounding in the formal exposition of the equations. The derivation of AWRM was easy to follow with the structure put in place by the authors.

The included experiments clearly lay out the intended contribution of the proposed approach, that GALILEO provides a more accurate and counterfactually correct model. Impressively, this improved model is shown to have clear benefits for downstream performance in tasks beyond the “pre-training” task.

**Weaknesses:**

I’m not sure I agree that having real-world offline data being biased is a catastrophic problem. Obviously for model-learning it creates major difficulties but perhaps we can focus on identifying better approaches to using expert data than expecting an RL agent to “figure it out” when there aren’t reliable demonstrations of counterfactuals. There’s likely a good reason why alternative or “exploratory” behaviors are not represented in the data? See Fatemi, et al (2021; NeurIPS) and Killian, et al (2023; TMLR) for a formulation of how we may more adequately think about risk and decision making in such environments.

The spacing between paragraphs and definitions+equations is really tight. This makes the paper difficult to read. I understand that this was likely a response to the NeurIPS template and page limitations but it should be fixed.

While the downstream performance of MBRL methods using GALILEO models is promising. I wish that more analysis was done in the learning dynamics + performance in these downstream tasks regarding sample efficiency and deviations from the behavioral data / optimal policies in these environments. Do the GALILEO agents have predetermined action sequences that they exploit early on or are they flexible to the change in domain/task?

>Fatemi, Mehdi, et al. "Medical dead-ends and learning to identify high-risk states and treatments." Advances in Neural Information Processing Systems 34 (2021): 4856-4870

>Killian, Taylor W., Sonali Parbhoo, and Marzyeh Ghassemi. "Risk Sensitive Dead-end Identification in Safety-Critical Offline Reinforcement Learning." Transactions on Machine Learning Research (2022).


**Questions:**

For the MuJoCo experiments it is unclear between Section 5.1 and 5.2 what the set-up is. It appears that the models are trained with the “medium” datasets but are evaluated on “expert” and “medium-replay” datasets. Is this correct?

For Figure 5b and 5c, what is meant by “update steps” on the horizontal axis? Is this an evaluation of model performance on the test data while training on the training data? This should be made more clear. I was at first inclined to believe that the models were being fine-tuned on the test data…

Title for Section 5.3 has typo: “Ddownstream”

I was curious why Causal Curiosity (Sontakke, et al 2021ICML) wasn’t used as a baseline? The paper is included among the References but there is no mention of this work in the Related Work section. I would imagine that it would be a relevant baseline to a counterfactually driven model learning approach like GALILEO.

> Sontakke, Sumedh A., et al. "Causal curiosity: Rl agents discovering self-supervised experiments for causal representation learning." International conference on machine learning. PMLR, 2021.


**Limitations:**

As stated by the authors, there are several simplications to the modeling process which may be a cause for the deviation in GALILEO performance when applied to downstream tasks.

Additionally, as mentioned in the “Weaknesses” section, there is an assumption that counterfactual modeling (on action sets outside the support of the dataset) is admissible. This may eliminate the use of GALILEO among safety critical domains, which would be a majority of real-world settings where model-learning could be useful.

---

> ### Author Rebuttal · Authors · 2023-08-10
>
> Thank you for the  detailed feedback on our paper. Your constructive comments are greatly appreciated. We would like to address the concerns and questions you raised as follows:
>
> 1. **Concern about Offline Data Bias and other potential solution in Fatemi, et al [1] and Killian, et al [2]**
>
>    We genuinely appreciate the meaningful references you shared.  We believe that  these two solutions are suitable for different scenarios. They [1,2] focus on dead-end discovery, intend to prevent agents from navigating towards potential terminal states, making them highly apt for risk-sensitive scenarios with explicit definitions, such as medical contexts. However, in some other tasks, there isn't a clear dead-end region. The profits or rewards of the tasks are continuous to the action space. To develop an effective policy, we inevitably have to handle the negative influence of biased data on dynamics model predictions, catering to a different set of applications and scenarios between these works.
>
>    Take the task BAT we've explored as an example,  in such tasks, for any order, raising the allowance will increases its acceptance intention. The optimization focus lies in devising a budget allocation policy to maximize the system's overall order intent. In this scenario, no order is truly a "dead end"; there are only differences in the speed of acceptance. To develop an effective policy, we have to learn an accurate dynamics model to find an accurately  allocate budgets among different targets.
>
> In the revised version, we will add these related studies to Appendix F.
>
> 2. **Learning Dynamics and Performance Analysis:**
>
>    Thank you for the suggestion. To provide a more comprehensive view, we will include supplemental results in the revision, including
>
>    (1) The return curve of the algorithms in dynamics models and the testing environments;
>
>    (2) Visualizations of the trajectories of asymptotic policies learned by different models.
>
>    We summarize our finding as follows:
>
>    1. The RL algorithm requires roughly 4e5 samples to find the optimal policies in the dynamics models learned by these algorithms. However,  after policy learning in the dynamics models within about 0.5e5 samples, the policies' performances evaluated the deployment environments will be stable.
>    2. Interestingly, policies trained via IPW and SL either remain standing or fall backward in all three environments, whereas those from GALILEO and SCIGAN demonstrate forward movement in Walker2d and Hopper, and keep standing in HalfCheetah.
>
>
> 3. **Omission of Causal Curiosity in the Comparison:**
>
>  We did mention this work in Appendix F, recognizing its value in another thread of applying causal inference techniques to RL. In these studies, the researchers consider that the transition function is relevant to some hidden noise variables, where the concept of causal factors in [3] is a special instance of the hidden noise variable. These studies focus on reconstructing the representation of the noise variable, or discovering and estimating the effects of the noise variables.   Our studies and the previous studies using IPS focused on handling the unbiased causal effect estimation problem of actions in the offline dataset under behavior policies collected with selection bias. In this branch of studies, we only consider the environment model learning problem in the fully observed setting thus the **hidden noise variable does not exist**.
>
> In the revised version, we will split the references into the  main-body part and the appendix part. We then will leave this citation to the appendix reference.
>
> 4. **Minor Typos and tight space**
>
>    Thank you for highlighting this. We commit to correcting these in the revised version for a more reader-friendly presentation.
>
> 5. **Questions Regarding MuJoCo Experiments:**
>
>    - **Q1: do the GALILEO agents have predetermined action sequences that they exploit early on or are they flexible to the change in domain/task?**
>
>      No. All agents don't employ any predetermined action sequences prior to deployment.
>
>    - **Q2: It appears that the models are trained with the “medium” datasets but are evaluated on “expert” and “medium-replay” datasets. Is this correct?**
>
>      The understanding are correct that models are trained on "medium" datasets and assessed on "expert" and "medium-replay" datasets. We will clarify this in the revision.
>
>    - **Q3: For Figure 5b and 5c, what is meant by “update steps” on the horizontal axis? Is this an evaluation of model performance on the test data while training on the training data? This should be made more clear. I was at first inclined to believe that the models were being fine-tuned on the test data…**
>
>      The observation about "update steps" is accurate. We load all datasets, train the model in the medium dataset, and periodically evaluate performance in other datasets. We will elucidate this further in the revised manuscript.
>
> 6. **Limitation on Safety-critical Domains:**
>
>    We agree with the observation that in highly safety-critical domains where no risks are allowed, the mentioned works [1,2] might be more appropriate. GALILEO, on the other hand, can be applicable in contexts where safety is essential but certain types of risks are permissible. We believe this distinction clarifies GALILEO's positioning and application scope in the field. We will leave this discussion in Section 6.
>
>
>
> In conclusion, your thoughtful feedback has guided us in identifying areas for enhancement and clarification. We believe that these revisions will significantly improve our paper and more effectively convey the contributions and applicability of GALILEO.
>
>
>
> [1] Medical dead-ends and learning to identify high-risk states and treatments.
>
> [2] Risk Sensitive Dead-end Identification in Safety-Critical Offline Reinforcement Learning.
>
> [3] Causal curiosity: Rl agents discovering self-supervised experiments for causal representation learning.

---

> > ### Comment · Reviewer_E1XE · 2023-08-14
> > **Thanks for the responses**
> >
> > After reviewing the author response, I am satisfied and my concerns have been addressed. I have no further questions.
> >
> > However, one point of clarification about the dead-end discovery approaches. From my understanding of the work, the notion of "dead-end" is unknown a priori and the formulation of how the value functions are trained help to identify possible dead-end regions in the state-action space. True, these methods do depend on the definition of a clear negative outcome that would ideally be avoided but this does not necessarily mean that the suboptimal decisions or regions of the state space are also known.
> >
> > My suggestion of these works is not to point out important weakness in the submitted paper but is to perhaps raise some point of discussion about the claim that biased real-world datasets present irrecoverable limitations. Perhaps, as motivated in part by the insights derived for AWRM, dealing with complex data sources requires alternative approaches to derive helpful insights.
> >
> > Altogether, I'm pleased with the current state of this paper and do feel that it would be of high interest to the MBRL community. It should certainly be published.

---

> > > ### Author Response · Authors · 2023-08-15
> > >
> > > Sincerely thank you for your appreciation of our work. We would like to further clarify the point made in the previous discussion about the studies of dead-end discovery.  What we like to say is that dead-end discovery is more suitable for applications where avoiding reaching the terminal function is crucial to completing the tasks, e.g., health care. In these applications, we can use the formulation of the value function, e.g., $Q^*_D$ [Fatemi, et al], to discover the risky regions to help the policy training directly, instead of learning the dynamics models.   However, in some applications, it is not very important to avoid reaching the terminal state.   For example, in the BAT task,  none of our intervention actions in the process of fulfilling orders will really lead the deliverymen to be terminal or ``dead'', where the only differences are in the acceptance time and delivery time; what we really care about is allocating the budget to reduce the system's overall order-taken time. In such applications, we have to make accurate predictions on the effects of actions on the order-taken rate for finding an effective policy.

---

### Author Rebuttal · Authors · 2023-08-10

We would like to extend our heartfelt appreciation to the esteemed reviewers for their invaluable feedback and insightful comments. Their constructive input has undoubtedly enhanced the quality of our manuscript. The common strengths highlighted across the reviews are encouraging, and we are grateful for the recognition of our efforts:

**Clarity of Presentation**: Multiple reviewers (E1XE, h4bm, wiCk, qQGz) emphasized that the paper is well-articulated, with clear illustrations and examples. The structure, as highlighted by E1XE, made the derivation of AWRM easy to follow, while the use of figures as pointed out by wiCk added to the paper's clarity.

**Novelty and Significance of Methodology**: The introduction of the adversarial weighted empirical risk minimization (AWRM) was recognized as innovative and novel by reviewers So7i, wiCk, and qQGz. The method effectively addresses significant challenges in the domain, including the problem of selection bias as mentioned by So7i.

**Comprehensive Experiments**: Reviewers E1XE, wiCk, and qQGz lauded the comprehensive experimental evaluation provided, noting the wide range of tasks, including synthetic, continuous control, and real-world applications. Particularly, the experiments showcased that GALILEO can accurately predict counterfactual actions and significantly enhance downstream tasks.

**Depth of Analysis**: The in-depth analysis of the challenges posed by conventional empirical risk minimization methods and the subsequent development of our proposed method was highlighted by qQGz as being insightful and backed by strong rationale.

We sincerely thank the reviewers for recognizing these strengths, and we have diligently addressed other feedback and suggestions to further refine our work.

---

### Decision · Program_Chairs · 2023-09-21

**Decision:**

Accept (spotlight)

**Comment:**

All of the reviewers agree that the paper makes significant and novel contributions to learning from data with selection biases. This is a well-motivated problem for off-policy bandit and reinforcement learning and counterfactual prediction in general. During the discussion phase, the authors clarified their choice of baselines, and detailed how they trained the discriminators to address the reviewers' concerns about adversarial training stability. These clarifications substantially strengthen the paper's exposition and should be included in the revision.